# Linear Dynamics meets Linear MDPs:
# Closed-Form Optimal Policies via Reinforcement Learning

## Abstract

Many applications—including power systems, robotics, and economics—involve a dynamical system interacting with a stochastic and hard-to-model environment. We adopt a reinforcement learning approach to control such systems. Specifically, we consider a deterministic, discrete-time, linear, time-invariant dynamical system coupled with a feature-based linear Markov process with an unknown transition kernel. The objective is to learn a control policy that minimizes a quadratic cost over the system state, the Markov process, and the control input. Leveraging both components of the system, we derive an explicit parametric form for the optimal state-action value function and the corresponding optimal policy. Our model is distinct in combining aspects of both classical Linear Quadratic Regulator (LQR) and linear Markov decision process (MDP) frameworks. This combination retains the implementation simplicity of LQR, while allowing for sophisticated stochastic modeling afforded by linear MDPs, without estimating the transition probabilities, thereby enabling direct policy improvement. For the nominal setting, where the linear system dynamics are known, we use tools from control theory to provide theoretical guarantees on the stability of the system under the learned policy and provide a sample complexity analysis for its convergence to the optimal policy. We further extend our framework to systems with Gaussian process noise and to systems with unknown linear dynamics. We illustrate our results via numerical examples for the nominal, noisy, and unknown dynamics settings to demonstrate the effectiveness of our approach in learning the optimal control policy under partially known stochastic dynamics.

## 1 Introduction

In many applications, a well-modeled agent must interact with and make decisions in stochastic and hard-to-model environments, with the aim to optimize a certain cost that is affected by the agent's objective and the environment. A prominent example arises in power systems, where a controllable energy storage device evolves under known physical dynamics, yet must respond to uncertain net load demand driven by exogenous factors such as variability in generation, consumer behavior, and weather conditions, all of which are unaffected by the device's control actions. Similar challenges appear for autonomous systems operating in unknown stochastic environments, or economic systems influenced by latent market factors. In such settings, designing optimal control strategies requires accounting for both the predictable evolution of the system and the stochastic nature of the surrounding environment. Effectively controlling such systems requires models that capture both the deterministic evolution of the agent's state and the stochastic evolution of the environment. To address this challenge, in this work, we model the agent (e.g., battery energy storage system, self-driving car) with deterministic linear dynamics derived from first principles, while we model the environment (e.g., net load demand, traffic) as a linear Markov process.

Classical control theory offers elegant solutions for systems with entirely known dynamics, such as the Linear Quadratic Regulator (LQR) optimal control problem, which yields a closed-form optimal policy via Riccati equations (Anderson & Moore, 2007). On the other hand, reinforcement learning (RL) approaches have developed data-driven techniques for decision-making in unknown environments, including model-based approaches. One compelling model in this setting is that of linear Markov decision processes (linear MDPs) that leverage feature-based representations to approximate the transition kernel (Bradtke & Barto, 1996;

Francisco & Ribeiro, 2007; Sutton & Barto, 2018). The linearity of the Markov kernel coupled with the non-linearity of feature functions results in a rich but tractable model.

Yet, even such a tractable framework does not distinguish between the system dynamics and environment, viewing them as a single entity driven by the same dynamics. To this end, we propose a RL framework that combines both the LQR and linear MDP paradigms: a deterministic, discrete-time, linear time-invariant (LTI) system coupled with a stochastic environment modeled as a feature-based linear Markov process with unknown transition kernel that is unaffected by the control actions. Our objective is to design a controller that minimizes a quadratic cost over the joint system and environment states and the control actions. We first consider a nominal setting in which the LTI dynamics and the quadratic cost weights are known a priori, while only the environment's transition kernel is unknown. We also consider extensions of the nominal setting to systems with Gaussian process noise and to LTI systems with unknown dynamics.

*Our approach*: By combining the structure of LQR and linear MDPs, we derive parametrized closed-form expressions for the optimal state-action value function and the corresponding optimal policy, capturing both LTI dynamics and latent stochastic effects of the environment within a unified model. This hybrid model preserves the simplicity of LQR policies while incorporating the expressive stochastic modeling of linear MDPs. Based on this structure, we develop a least-squares value iteration (LSVI) algorithm to learn the value function parameters from online data in an episodic fashion. The closed-form expression of the policy that optimizes the state-action value function makes it amenable to efficiently perform the policy update directly using the updated parameters at the end of each episode. Furthermore, because the unknown transition kernel is unaffected by the control actions, our LSVI algorithm does not require exploration in the nominal setting. For the nominal setting, we establish closed-loop stability guarantees under the learned policy and show that our LSVI achieves a regret bound of $\widetilde{\mathcal{O}}\left(T\sqrt{dL}\right)$ with high probability, where $d$, $T$, and $L$ denote the dimension of the feature-space of the linear Markov model, the time horizon of each episode, and the number of episodes, respectively. We further extend our framework to systems with Gaussian process noise and to systems with unknown dynamics. These extensions are illustrated numerically. However, stability and convergence analysis for these settings are left for future work.

## 1.1 Related work

Our work lies at the intersection of optimal control and reinforcement learning, where we bridge ideas from the LQR optimal control problem and linear MDPs. Below, we review prior work that has been done in each area and highlight how our approach uniquely integrates them.

**Linear Quadratic Regulator (LQR):** The classical LQR problem admits closed-form optimal control policies for linear systems. Traditional methods assume full knowledge of the system dynamics and cost, enabling the computation of optimal policies via Riccati equations (Anderson & Moore, 2007). Recent work has studied the LQR problem in data-driven settings. Direct data driven approaches have been studied in (De Persis & Tesi, 2020; Dörfler et al., 2022; Celi et al., 2023), where the optimal policy is learned directly from offline data generated by the open-loop system. Indirect data-driven approaches, explored in (Aangenent et al., 2005; da Silva et al., 2018; Dean et al., 2020), first identify a model of the system dynamics from data then solve the LQR problem using the identified model. Other works have studied the LQR problem in online learning setting (Fazel et al., 2018; Mohammadi et al., 2019; Bu et al., 2019; Fatkhullin & Polyak, 2021; Bradtke et al., 1994), where the optimal policy is learned online using policy gradient methods.

**Reinforcement learning with function approximation:** In many reinforcement learning (RL) problems, the state or action spaces are too large (or continuous) to allow for tabular representations of value functions or policies (Kober et al., 2013; Mnih et al., 2013; Silver et al., 2016). To address this, function approximation techniques are employed to generalize from observed states and actions to unseen ones, enabling scalability and improved sample efficiency. Among the function approximation models, linear function approximation is particularly appealing due to its computational simplicity, theoretical tractability, and its ability to support efficient learning algorithms. Early approaches such as temporal difference learning, Q-learning, and least-squares temporal difference (LSTD) algorithms with linear value function approximation were explored in works like (Bradtke & Barto, 1996; Francisco & Ribeiro, 2007; Sutton & Barto, 2018).

While these methods laid important foundations, they often lacked sample efficiency guarantees and relied on heuristic exploration. Recent studies have introduced sample-efficient algorithms for linear MDPs, where the transition kernel is assumed to be a linear function of known features and unknown parameters (Yang & Wang, 2019; 2020; Jin et al., 2020). In (Jin et al., 2020), the authors developed a sample-efficient reinforcement learning algorithm for linear MDPs with a finite action space and a potentially infinite state space. Their model represents the transition kernel as a linear combination of known features with unknown probability measures, and assumes the reward function is linear in the same features with unknown parameters. In (Yang & Wang, 2020), the authors proposed a sample-efficient reinforcement learning algorithm under a linear MDP setting with possible infinite state and action space. In their framework, they assume the reward is known; further, their model introduces an additional structural assumption compared to (Jin et al., 2020), by parameterizing the transition kernel with a low-dimensional unknown matrix. This assumption reduces the learning problem to estimating this matrix, thereby significantly lowering the overall learning complexity. In our framework, we model the stochastic environment as a feature-based linear Markov Process. Similar to (Jin et al., 2020), we represent the transition kernel as a linear combination of known features with unknown probability measures. However, we assume a known quadratic cost that is independent of the features and consistent with the LQR framework, enabling efficient policy computation. This choice of the cost is more realistic and aligns with common formulations in engineering applications. Furthermore, our model avoids explicit parametric assumptions on the transition kernel made in (Yang & Wang, 2020), while allowing infinite state and action spaces. Additionally, our approach bypasses full model estimation by learning the value function directly through least-squares, benefiting from control-theoretic structure to ensure stability as well as computational and sample efficiency. Finally, our framework does not require exploration, since the environment is exogenous and is unaffected by the control inputs. Beyond linear MDPs, several works propose generalized model classes for sample-efficient RL including Bellman rank class, (Jiang et al., 2017), linear Bellman-complete classes (Munos, 2005; Zanette et al., 2020), witness rank (Sun et al., 2019), and bilinear class (Du et al., 2021).

## 1.2 Contributions

We list our contributions below.

- We propose a RL framework that unifies the classical LQR optimal control problem with linear MDPs. This hybrid model captures both the deterministic dynamics of physical systems and the stochastic evolution of exogenous environments. To the best of our knowledge, this integration has not been addressed in the existing literature.

- We derive a parametric form for the optimal state-action value function that decouples the agent's dynamics from the environment's stochasticity. This yields a closed-form policy that exhibits the simplicity of the LQR while inheriting the rich modeling capabilities linear MDPs.

- We propose a least-squares value iteration (LSVI) algorithm that learns the optimal policy by directly estimating the value function parameters. The LQR structure of our problem allows the control policy to be explicitly expressed in terms of the learned parameters, without requiring optimization of the value function at each step as in (Jin et al., 2020), thus simplifying the algorithm's computational complexity.

- For the nominal setting, in which the linear system dynamics are known, we establish stability guarantees for the closed-loop system under the learned policy. These guarantees are given in terms of input-to-state stability, extending beyond standard sample-efficiency results in RL literature, where it is typically assumed that the reward is bounded.

- For the nominal setting, we derive a regret bound for our LSVI algorithm, showing that it achieves a rate $\widetilde{\mathcal{O}}\left(T\sqrt{dL}\right)$ with high probability, where $d$, $T$, and $L$ denote the dimension of the feature-space of the linear Markov model, the time horizon of each episodes, and the number of episodes, respectively.

- Going beyond the nominal setting, we extend our framework to systems with Gaussian process noise and to systems with unknown linear dynamics.

- We provide numerical examples to demonstrate the effectiveness of our framework in the nominal, noisy, and unknown system dynamics settings. In the nominal setting, we highlight the convergence and verify the closed-loop stability of the learned policy.

## 2 Problem formulation

Consider an agent obeying the discrete-time, linear, time-invariant dynamics over a finite time horizon

$$x_{t+1} = Ax_t + Bu_t, \quad t \in \{0, 1, \ldots, T-1\}, \tag{1}$$

where $x_t \in \mathcal{X} = \mathbb{R}^n$ denotes the state and $u_t \in \mathcal{U} = \mathbb{R}^m$ the input with $x_0 \sim \mathcal{N}(0, \Sigma_x)$ with $\Sigma_x \succ 0$. We assume the linear dynamical system, defined via the matrix pair $(A, B)$, is controllable[1]. We consider an environment evolving according to the discrete-time Markov Process

$$s_{t+1}|s_t \sim \mathbb{P}_t\left(s_{t+1}|s_t\right), \quad t \in \{0, \ldots, T-1\}, \tag{2}$$

where $s \in \mathcal{S} \subset \mathbb{R}^p$ denotes the state of the Markov Process and $\mathbb{P}_t\left(s'|s\right)$ denotes the transition probability from state $s$ to $s'$, with $s_0 \sim \mu_0$ for some distribution $\mu_0 \in \Delta(\mathcal{S})$, where $\Delta(\mathcal{S})$ denotes the set of distributions over $\mathcal{S}$. We assume that the matrices $A$ and $B$ in eq. (1) are known, while the transition probability, $\mathbb{P}_t$, in eq. (2) is unknown. The agent follows a control policy $\pi_t : \mathcal{X} \times \mathcal{S} \to \mathcal{U}$, where $u_t = \pi_t\left(x_t, s_t\right)$ is the action that the agent takes at state $x_t$ and $s_t$ at time $t$, for $t \geq 0$. The objective is to find an optimal control policy, $\boldsymbol{\pi} = (\pi_0, \ldots, \pi_T)$, that optimizes the following control task

$$
\begin{aligned}
\underset{\boldsymbol{\pi}}{\text{minimize}} \quad & \mathbb{E}\left[\sum_{t=0}^{T} c\left(x_t, s_t, u_t\right)\right], \\
\text{subject to} \quad & x_{t+1} = Ax_t + Bu_t, \\
& s_{t+1} \sim \mathbb{P}_t\left(s_{t+1}|s_t\right), \\
& u_t = \pi_t\left(x_t, s_t\right),
\end{aligned}
\tag{3}
$$

where $c : \mathcal{X} \times \mathcal{S} \times \mathcal{U} \to \mathbb{R}_{\geq 0}$ is the cost evaluated at $x_t$, $s_t$, and $u_t$ for $t \geq 0$ with $u_T = \pi_T(x_T, s_T) = 0$. We restrict our search in eq. (3) to the class of deterministic policies. We show later in Section 3.1 that the optimizer of eq. (3) is indeed deterministic. We introduce the following assumptions on the transition probability in eq. (2) and the cost in eq. (3).

**Assumption 2.1.** *(**Linear Markov Process**) Let $\phi : \mathcal{S} \to \mathbb{R}^d$ be a known feature vector and $\mu_t \in \mathbb{R}^d$ a vector of $d$ unknown signed measures over $\mathcal{S}$. For $s', s \in \mathcal{S}$, we have*

$$\mathbb{P}_t\left(s'|s\right) = \phi(s)^\mathsf{T} \mu_t(s'), \tag{4}$$

*We assume $\|\phi(s)\| \leq 1/\sqrt{d}$ and $\|s\| \leq \delta_s$ for all $s \in \mathcal{S}$, $\mathbb{E}\left[\phi(s_t)\phi(s_t)^\mathsf{T}\right] \succ 0$, and $\|\mu_t\| \leq 1$ for all $t$.*

**Assumption 2.2.** *(**Quadratic cost**) For $x \in \mathcal{X}$, $s \in \mathcal{S}$, and $u \in \mathcal{U}$, we have*

$$
c(x, s, u) = \begin{bmatrix} x \\ s \\ u \end{bmatrix}^\mathsf{T} \underbrace{\begin{bmatrix} W & F & D \\ F^\mathsf{T} & M & H \\ D^\mathsf{T} & H^\mathsf{T} & R \end{bmatrix}}_{C} \begin{bmatrix} x \\ s \\ u \end{bmatrix}, \tag{5}
$$

*where $C \succeq 0$ is known and $R \succ 0$. Further, we assume the pair $(A, W^{1/2})$ is observable.*

Assumption 2.1 is inspired by the linear MDP framework introduced in (Bradtke & Barto, 1996; Francisco & Ribeiro, 2007; Jin et al., 2020). However, unlike the original definition, our model assumes that the stochastic process governs only the exogenous state and is unaffected by control input. This assumption is motivated by the fact that, in our target applications, the environment is not influenced by control actions. Moreover, it

---

[1]When the system is controllable, it implies that there exist an input sequence, $u$, that can drive the system from its initial state, $x_0$ to any final state, $x_t$, within finite time horizon (see (Ogata, 2010, Section 9.8)).

simplifies the expression of the optimal policy, as the optimal policy requires minimizing a quadratic function in the input $u$ (from Assumption 2.2), rather than the nonlinear (possibly non-convex) function $\phi$. We define the value function $V_t^\pi : \mathcal{X} \times \mathcal{S} \to \mathbb{R}$ as the expected cumulative cost incurred under policy $\pi$ starting from state $x_t$ and $s_t$ at time $t \geq 0$, given by

$$V_t^\pi(x, s) \triangleq \mathbb{E}\left[\sum_{i=t}^{T} c\left(x_i, s_i, \pi_i\left(x_i, s_i\right)\right) \Big| x_t = x, s_t = s\right].$$

Further, we define the state-action value function $Q_t^\pi : \mathcal{X} \times \mathcal{S} \times \mathcal{U} \to \mathbb{R}$ as the expected cumulative cost under policy $\pi$ starting from state $x_t$, $s_t$, and action $u_t$ at time $t \geq 0$, given by

$$Q_t^\pi(x, s, u) \triangleq c(x, s, u) + \mathbb{E}\left[\sum_{i=t+1}^{T} c(x_i, s_i, \pi_t\left(x_i, s_i\right)) \Big| x_t = x, s_t = s, u_t = u\right].$$

To learn the optimal policy, we focus on estimating the state-action value function $Q_t^\pi$, since it directly guides policy improvement through greedy action selection. In particular, by learning an appropriate parametric approximation of the state-action value function, $Q_t^\pi$, we can infer an optimal policy without explicitly learning the transition probability measures, $\mu$, in eq. (4). This approach leverages the structure of the system and cost, allowing us to bypass the need for full system identification and instead focus on value function approximation within the RL framework.

**Remark 1.** *(On the knowledge of $A$ and $B$ in eq. (1)) In many control applications—such as robotics and power systems—the plant dynamics (i.e., $A$ and $B$) can be easily derived from first principles or can be accurately identified through standard system identification techniques prior to deployment. Our framework leverages this knowledge to focus on learning the stochastic environment component, which simplifies the computational complexity of the policy update step, and enables stability-aware control without requiring aggressive exploration. Nonetheless, we extend our framework to settings in which $A$ and $B$ are unknown in Section 5. We also extend it to systems with Gaussian process noise in Section 4.*

## 3 Main Results

We leverage the linear structures of the system in eq. (1), the transition model in eq. (4), and the quadratic structure of the cost in eq. (5) to derive a parametric expression for the state-action value function that is linear in the feature map, $\phi$, along with a parametric expression for the optimal greedy policy. We introduce a least-squares value iteration algorithm to learn the parameters of the state-action value function, and therefore learn the optimal policy. We provide stability guarantees for the closed-loop system under the learned policy and a convergence analysis yielding a high-probability regret bound.

### 3.1 State-action value function approximation

Let the optimal value function at time $t$ and evaluated at $x \in \mathcal{X}$ and $s \in \mathcal{S}$ under the optimal policy, $\pi_t^*$, be denoted by $V_t^*(x, s)$. Following the Bellman optimality equation, we can write the optimal state-action value function at time $t$ and evaluated at $x \in \mathcal{X}$, $s \in \mathcal{S}$, and $u \in \mathcal{U}$ under $\pi_t^*$ as

$$Q_t^*(x, s, u) = c(x, s, u) + \mathbb{E}_{s' \sim \mathbb{P}_t(s'|s)}\left\{V_{t+1}^*(Ax + Bu, s')|s\right\}.$$

The next result provides an explicit parametric form for the state-action value function $Q_t$.

**Theorem 3.1.** *(Q-function representation) Consider the dynamics in eq. (1) and the Markov Process in eq. (2). Let Assumption 2.1 and Assumption 2.2 be satisfied. Then, for any $x \in \mathcal{X}$, $s \in \mathcal{S}$, and $u \in \mathcal{U}$, and under $\pi_t^*$ for $t \geq 0$, there exists $\overline{h}_{i,t+1} \in \mathbb{R}^n$ and $\overline{q}_{i,t+1} \in \mathbb{R}$ such that*

$$Q_t^*(x, s, u) = c(x, s, u) + (Ax + Bu)^\mathsf{T} G_{t+1}(Ax + Bu) + \sum_{i=1}^{d} \phi_i(s)\Big(2(Ax + Bu)^\mathsf{T}\overline{h}_{i,t+1} + \overline{q}_{i,t+1}\Big), \qquad (6)$$

where $G$ solves the discrete-time algebraic Riccati equation

$$G_t = A^{\mathsf{T}}G_{t+1}A + W - (A^{\mathsf{T}}G_{t+1}B + D)(R + B^{\mathsf{T}}G_{t+1}B)^{-1}(B^{\mathsf{T}}G_{t+1}A + D^{\mathsf{T}}), \tag{7}$$

with $G_T = W$.

A proof of Theorem 3.1 is in Appendix A. Several comments are in order. First, by leveraging the linearity of the system in eq. (1) and the Markov process in eq. (4), along with the quadratic structure of the cost in eq. (5), the state-action value function in eq. (6) exhibits a structure that decouples the linear system state $x$ and action $u$ from the exogenous state $s$. Second, the derived expression of the state-action value function in eq. (6) is linear in the feature map $\phi$ and the weight parameters $\overline{h}_{i,t}$ and $\overline{q}_{i,t}$. Third, the weight parameters $\overline{h}_{i,t}$ and $\overline{q}_{i,t}$ depend on the unknown transition probability $\mathbb{P}(\cdot|s)$ in eq. (4), and therefore, learning the state-action value function boils down to learning these weights, thereby bypassing the need to explicitly learn the probability measures, $\mu$, in eq. (4). The optimal policy is found by minimizing the $Q$ function over the input $u$. Since by Theorem 3.1, $Q$ is quadratic in $u$, this optimal policy can be found in closed form, as shown in the following corollary, which expresses the optimal policy in terms of feedback gains and weight parameters.

**Corollary 3.2.** *(Optimal policy representation) For any $x \in \mathcal{X}$, $s \in \mathcal{S}$, and $t \in \{0, 1, \cdots, T-1\}$*

$$u_t^*(x, s) = \pi_t^*(x, s) = K_{x,t}x + K_{s,t}s + K_{h,t}\sum_{i=1}^{d}\phi_i(s)\overline{h}_{i,t+1},$$

*where $\overline{h}_{i,t+1}$ is as in Theorem 3.1 and*

$$
\begin{aligned}
K_{x,t} &= -\left(R + B^{\mathsf{T}}G_{t+1}B\right)^{-1}\left(B^{\mathsf{T}}G_{t+1}A + D^{\mathsf{T}}\right), \\
K_{s,t} &= -\left(R + B^{\mathsf{T}}G_{t+1}B\right)^{-1}H^{\mathsf{T}}, \\
K_{h,t} &= -\left(R + B^{\mathsf{T}}G_{t+1}B\right)^{-1}B^{\mathsf{T}},
\end{aligned}
\tag{8}
$$

*and $G_{t+1}$ satisfies eq. (7).*

---

**Algorithm 1** Least-Squares Value Iteration

---

1: Given: $L$, $R_\theta$, $\lambda$
2: **for** episode $\ell = 1, \cdots, L$ **do**
3: $\quad x_0^\ell \overset{\text{i.i.d}}{\sim} \mathcal{N}(0, \Sigma_x)$ with $\Sigma_x \succ 0$
4: $\quad s_0^\ell \overset{\text{i.i.d}}{\sim} \mu_0$ such that $\mathbb{E}\left[\phi(s_0)\phi(s_0)^{\mathsf{T}}\right] \succ 0$
5: $\quad$ **for** step $t = T-1, \cdots, 0$ **do**
6: $\qquad \Lambda_t^\ell \leftarrow \sum_{i=1}^{\ell-1}Y(x_t^i, u_t^i)^{\mathsf{T}}\phi(s_t^i)\phi(s_t^i)^{\mathsf{T}}Y(x_t^i, u_t^i) + \lambda I_{dn+d}$
7: $\qquad \theta_{t+1}^\ell \leftarrow (\Lambda_t^\ell)^{-1}\sum_{i=1}^{\ell-1}Y(x_t^i, u_t^i)^{\mathsf{T}}\phi(s_t^i)\epsilon_{t+1}(x_{t+1}^i, s_{t+1}^i)$
8: $\qquad$ **if** $\|\theta_{t+1}^\ell\| > R_\theta$ **then**
9: $\qquad\quad \theta_{t+1}^\ell \leftarrow \frac{R_\theta}{\|\theta_{t+1}^\ell\|}\theta_{t+1}^\ell$
10: $\qquad$ **end if**
11: $\quad$ **end for**
12: $\quad$ **for** step $t = 0, \cdots, T-1$ **do**
13: $\qquad u_t^\ell \leftarrow K_{x,t}x_t^\ell + K_{s,t}s_t^\ell + K_{h,t}(\phi(s_t^\ell)^{\mathsf{T}} \otimes Z)\theta_{t+1}^\ell$
14: $\qquad$ Take action $u_t^\ell$
15: $\qquad$ Observe $x_{t+1}^\ell$ and $s_{t+1}^\ell$
16: $\quad$ **end for**
17: **end for**

---

### 3.2 Learning weight parameters of the value function via least-squares value iteration

In this subsection, we learn the weight parameters, $\overline{h}$ and $\overline{q}$, that parameterize the state-action value function in Theorem 3.1. To this aim, we propose a least-squares value iteration algorithm (Algorithm 1) that is inspired by (Jin et al., 2020). Before we lay out the steps of our algorithm, we introduce the following notations. At each time step, $t$, we concatenate the parameters $\overline{h}_{i,t}$ and $\overline{q}_{i,t}$ for $i \in \{1, \cdots, d\}$ as

$$\theta_t = \begin{bmatrix} \theta_{1,t}^\mathsf{T} & \cdots & \theta_{d,t}^\mathsf{T} \end{bmatrix}^\mathsf{T}, \quad \text{where} \quad \theta_{i,t} = \begin{bmatrix} \overline{h}_{i,t}^\mathsf{T} & \overline{q}_{i,t}^\mathsf{T} \end{bmatrix}^\mathsf{T}. \tag{9}$$

Using the notation in eq. (9), we rewrite the $Q$-function in Theorem 3.1 and the policy in Corollary 3.2 as

$$Q_t(x, s, u) = c(x, s, u) + (Ax + Bu)^\mathsf{T} G_{t+1} (Ax + Bu) + \phi(s)^\mathsf{T} Y(x, u)\theta_{t+1},$$
$$u_t(x, s) = K_{x,t}x + K_{s,t}s + K_{h,t}\left(\phi(s)^\mathsf{T} \otimes Z\right)\theta_{t+1}, \tag{10}$$

where $Y(x, u) = I_d \otimes [2(Ax + Bu)^\mathsf{T}, 1]$ and $Z = [I_n, 0_{n \times 1}]$. Now we lay out the steps of our least-squares value iteration algorithm (Alg. 1). Our algorithm consists of an outer loop over $L$ episodes, where each episode consists of two loops: 1) backward-in-time weight update loop (lines 5-11) and 2) forward roll-out and data collection loop (lines 12-16). During the first pass of episode $\ell$ (lines 5–11), we treat the data collected in the previous $\ell - 1$ episodes as a fixed dataset

$$\mathcal{D}_{\ell-1} := \left\{ (x_t^i, s_t^i, u_t^i, x_{t+1}^i, s_{t+1}^i) : i < \ell,\ 0 \le t < T \right\}. \tag{11}$$

At each time step $t$, $\theta$ minimizes a regularized least-squares loss—the squared error between the parametric state-action value function in eq. (10) and the Bellman target (immediate cost plus the estimated value of the next state). Solving this problem on past trajectory data yields an accurate value-function approximation and enables closed-form greedy policy updates without estimating the transition probabilities. The regularized least-squares regression is stated as (see Appendix B for details)

$$\theta_{t+1}^\ell = \underset{\theta \in \mathbb{R}^{d(n+1)}}{\arg\min} \sum_{i=1}^{\ell-1} \left( \phi(s_t^i)^\mathsf{T} Y(x_t^i, u_t^i)\theta - \epsilon_{t+1}^\ell(x_{t+1}^i, s_{t+1}^i) \right)^2 + \lambda \|\theta\|^2,$$

where

$$\epsilon_{t+1}^\ell(x, s) = 2(x)^\mathsf{T} h_{t+1}^\ell(s) + q_{t+1}^\ell(s), \tag{12}$$
$$h_{t+1}^\ell(s) = \left(A^\mathsf{T} + K_{x,t}^\mathsf{T} B^\mathsf{T}\right)(\phi(s)^\mathsf{T} \otimes Z)\theta_{t+2}^\ell + \left(F + K_{x,t}^\mathsf{T} H^\mathsf{T}\right) s, \tag{13}$$
$$q_{t+1}^\ell(s) = \left(\phi(s)^\mathsf{T} \otimes \overline{Z}\right)\theta_{t+2}^\ell + s^\mathsf{T}(M + HK_{s,t})s + \theta_{t+2}^{\ell\ \mathsf{T}}\left(\phi(s) \otimes Z^\mathsf{T}\right) BK_{h,t}(\phi(s)^\mathsf{T} \otimes Z)\theta_{t+2}^\ell$$
$$+ 2s^\mathsf{T} HK_{h,t}\left(\phi(s)^\mathsf{T} \otimes Z\right)\theta_{t+2}^\ell, \tag{14}$$

with $\overline{Z} = [0_{1 \times n}, 1]$. Unlike prior work (e.g., (Jin et al., 2020)), we leverage the structure of our model to derive a closed-form expression for the Bellman target in terms of previously learned parameters, thereby avoiding an inner optimization over the action space at each time step (often required in discrete action space settings). In fact, $\epsilon_{t+1}^\ell(x, s)$ is obtained directly from this closed-form Bellman target (see Appendix B). The closed-form parameter update is given by

$$\Lambda_t^\ell = \sum_{i=1}^{\ell-1} Y(x_t^i, u_t^i)^\mathsf{T} \phi(s_t^i)\phi(s_t^i)^\mathsf{T} Y(x_t^i, u_t^i) + \lambda I_{d(n+1)},$$
$$\theta_{t+1}^\ell = \left(\Lambda_t^\ell\right)^{-1} \sum_{i=1}^{\ell-1} Y\left(x_t^i, u_t^i\right)^\mathsf{T} \phi\left(s_t^i\right) \epsilon_{t+1}^\ell\left(x_{t+1}^i, s_{t+1}^i\right), \tag{15}$$

which recover lines 6 and 7 of Alg. 1. For $\ell = 1$, we set $\theta_{t+1}^\ell = 0$ and $\Lambda_t^\ell = \lambda I_{d(n+1)}$ for $t \in \{0, \cdots, T-1\}$. The regularizer term $\lambda I_{d(n+1)}$ ensures numerical stability, the projection step in lines 8-10 makes sure that

the norm of the learned parameters is uniformly bounded for $t \in \{0, \cdots, T-1\}$ and $\ell \in \{1, \cdots, L\}$. In the second pass (lines 12–16) the newly computed parameters $\theta_{t+1}^\ell$ are plugged into the policy eq. (10),

$$u_t^\ell(x_t^\ell, s_t^\ell) = K_{x,t} x_t^\ell + K_{s,t} s_t^\ell + K_{h,t}\left(\phi(s_t^\ell)^\top \otimes Z\right)\theta_{t+1}^\ell,$$

generating a new trajectory $(\{x_t^\ell, s_t^\ell, u_t^\ell\}_{t=0}^T)$. These samples are appended to the collected data eq. (11), and will be used in the next episode's backward update. Notice that Alg. 1 does not require exploration as in (Jin et al., 2020), which we discuss in the following remark.

**Remark 2.** *(Role of exploration) In classical reinforcement learning, exploration (e.g., using $\varepsilon$-greedy or optimism-based methods) is necessary to sufficiently explore the environment and estimate unknown transition dynamics. However, our framework does not require exploration. This is because the stochastic component of the environment is modeled as an exogenous Markov process that evolves independently of the control inputs (see Assumption 2.1), and the system dynamics, $A$ and $B$, are known. Our algorithm estimates the value function parameters, $\theta$, via a least-squares procedure using observed trajectories without the need to infer the transition probabilities explicitly. Thus, the optimal policy can be computed in closed form by minimizing a known quadratic function of the input.*

**Remark 3.** *(Choice of $R_\theta$) The projection radius $R_\theta$ in Alg. 1 ensures that the learned parameters at each episode and time step remain within a ball of radius $R_\theta$, ensuring numerical stability. Moreover, it plays a crucial role in the theoretical analysis (i.e., stability and regret bound). In practice, $R_\theta$ should be chosen large enough to contain the true parameters, $\theta^*$, but not excessively large to keep the constants in the stability and regret bounds moderate. In Appendix C, we derive an upper bound on $\|\theta^*\|$; if $R_\theta$ is larger than this bound, then the ball of radius $R_\theta$ is guaranteed to contain $\theta^*$. In particular, we show that $\theta^*$ is contained in this ball if $R_\theta \geq c_\theta \sqrt{d}$, where $c_\theta > 0$ depends on known problem parameters, e.g., the system matrices, cost weights, the feature map, and the bound on $s$.*

### 3.3 Input-to-State Stability

It is critical to ensure that the learned policy stabilizes the closed-loop system in each episode, particularly in settings where the environment evolves independently of the control actions and safety is a concern. To this end, we establish an input-to-state stability (ISS) bound for the system under the learned policy, which we present in the following result.

**Theorem 3.3.** *(Input-to-state stability) Consider system eq. (1), let $u$ be the output of Algorithm 1 at episode $\ell$. Let $\|\theta_t^\ell\| \leq R_\theta$, $\|K_{s,t}\| \leq \overline{K}_s$, and $\|K_{h,t}\| \leq \overline{K}_h$ for $t \in \{0, \cdots, T-1\}$. Let $x_0^\ell$ be the initial state in episode $\ell$. Then, under Assumptions 2.1 and 2.2,*

$$||x_t^\ell|| \leq \alpha \rho^t \|x_0^\ell\| + \frac{\alpha \|B\|}{1-\rho}\left(\overline{K}_s \delta_s + \frac{\overline{K}_h R_\theta}{\sqrt{d}}\right), \tag{16}$$

*for $t \in \{0, \cdots, T-1\}$, where $\alpha > 0$ and $0 < \rho < 1$ are constants.*

A proof of Theorem 3.3 is deferred to Appendix D. Several comments are in order. First, Theorem 3.3 implies that the state trajectory at each episode $\ell \in \{1, \cdots, L\}$ remains bounded in terms of the initial condition, the system dynamics, the control gains in Corollary 3.2, and $R_\theta$. Second, the first term on the right-hand side of eq. (16), which depends on the initial state, decays exponentially with time, while the second term is independent of time and the number of episodes. This latter term depends on the system matrices, feedback gains, the bound on $s$, $R_\theta$ from Algorithm 1, and the dimension of the feature map, $d$. This result leverages the known system dynamics and the structure of the policy, extending traditional stability notions in control to learning-based policies in partially known settings.

### 3.4 Regret analysis

We define the regret $\mathcal{R}(L)$ as the difference between the total cost incurred by the learned policy and that of the optimal policy over $L$ episodes. Mathematically, for $L$ episodes, the regret is defined as

$$\mathcal{R}(L) = \sum_{\ell=1}^{L}\left(V_0^\ell(x_0^\ell, s_0^\ell) - V_0^*(x_0^\ell, s_0^\ell)\right). \tag{17}$$

where $V_0^\ell(x_0^\ell, s_0^\ell)$ denotes the value evaluated at the initial states $x_0^\ell$ and $s_0^\ell$ under the policy learned at episode $\ell$, and $V_0^*(x_0^\ell, s_0^\ell)$ is the value of the optimal policy evaluated at the initial states $x_0^\ell$ and $s_0^\ell$. We derive a bound on the regret in the following result.

**Theorem 3.4. *(Regret bound)*** *Let Assumptions 2.1 and 2.2 be satisfied. Let $\|Y(x_t^\ell, u_t^\ell)^\mathsf{T}\phi(s_t^\ell)\| \le \delta_\psi$, for $t \in \{0, \cdots, T\}$ and $\ell \in \{1, \cdots, L\}$. Let $\beta = \log\left(1 + \frac{L\delta_\psi^2}{\lambda}\right)$ with $\lambda > 0$. Let $\delta \in [0, 1/3]$. Then, with probability at least $1 - 3\delta$*

$$\mathcal{R}(L) \le \sigma\sqrt{2TL\log(1/\delta)} + \delta_\psi T\left(\frac{1}{\sqrt{\lambda}} + \frac{4\sqrt{L}}{\sqrt{\gamma}}\right)\left(\sigma\sqrt{2dn\beta + 2\log(\frac{1}{\delta})} + (R_\theta + 2\delta_v)\sqrt{\lambda}\right)$$

*where $\sigma > 0$, $\gamma > 0$ and $\delta_v > 0$ are constants that do not depend on $L$ and $T$, and do not scale with $d$. Further, $\delta_\psi$ scales with $\mathcal{O}(1/\sqrt{d})$ and $R_\theta$ scales with $\mathcal{O}(\sqrt{d})$.*

A proof of Theorem 3.4 is presented in Appendix E. Theorem 3.4 provides a probabilistic upper bound on the cumulative regret. Several comments are in order. First, the leading term of the bound scales as $\mathcal{O}\left(T\sqrt{dL\log(L)}\right)$ or $\widetilde{\mathcal{O}}\left(T\sqrt{dL}\right)$, which matches, in terms of the number of episodes, the rate reported in (Jin et al., 2020). Second, our bound grows linearly in $T$ and $\sqrt{d}$, in contrast to the $T^2$ and $d\sqrt{d}$ factors in (Jin et al., 2020), respectively. Third, the constants $\sigma$ and $\delta_v$ are independent of $L$ and $T$, and they do not scale with $d$ and they depend only on the system matrices in eq. (1), the cost weight matrices in eq. (5), the bound on the state, $x_t$, in eq. (16), and the bound on the exogenous state $s_t$ in eq. (4). In fact, they arise from the uniform upper bound on the value function, $V_t$ (see Appendix E for the explicit formulas). Fourth, the constant $\gamma$ satisfies $\mathbb{E}\left[Y(x_0, u_0)^\mathsf{T}\phi(s_0)\phi(s_0)^\mathsf{T}Y(x_0, u_0)\right] \succeq \gamma I_{d(n+1)}$, which holds because the initial states, $x_0$ and $s_0$ are drawn independently in each episode. Finally, the bound in Theorem 3.4 suggests, when the initial states, $x_0$ and $s_0$, are fixed for all episodes, Alg. 1 can learn an $\varepsilon-$optimal policy, $\pi$, that satisfies $V_0^\pi(x_0, s_0) - V_0^*(x_0, s_0) \le \varepsilon$ after $L = \widetilde{\mathcal{O}}\left(\frac{dT^2}{\varepsilon^2}\right)$ episodes.

# 4 Extension to Systems with Process Noise

In this section, we add Gaussian process noise to the system in eq. (1). In particular, we consider the following discrete-time, linear, time-invariant dynamics over a finite time horizon

$$x_{t+1} = Ax_t + Bu_t + w_t, \quad t \in \{0, 1, \ldots, T-1\}, \tag{18}$$

where $x_t \in \mathcal{X} = \mathbb{R}^n$ denotes the state, $u_t \in \mathcal{U} = \mathbb{R}^m$ the input with $x_0 \sim \mathcal{N}(0, \Sigma_x)$ with $\Sigma_x \succ 0$, and $w_t \in \mathbb{R}^n$ the process noise, where $w \overset{\text{i.i.d.}}{\sim} \mathcal{N}(0, \Sigma_w)$ and $\Sigma_w \succ 0$.

## 4.1 State-action value function approximation

Following the Bellman optimality equation, we can write the optimal state-action value function at time $t$ and evaluated at $x \in \mathcal{X}$, $s \in \mathcal{S}$, and $u \in \mathcal{U}$ under $\pi_t^*$ with $w_t \overset{\text{i.i.d.}}{\sim} \mathcal{N}(0, \Sigma_w)$ as

$$Q_t^*(x, s, u) = c(x, s, u) + \mathbb{E}_w\left\{\mathbb{E}_{s' \sim \mathbb{P}_t(s'|s)}\left\{V_{t+1}^*(Ax + Bu + w, s')|s\right\}\right\}.$$

The next result provides an explicit parametric form for the state-action value function $Q_t$ and $V_t$.

**Theorem 4.1. *(Value-function representation)*** *Consider the dynamics in eq. (18) and the Markov Process in eq. (2). Let Assumption 2.1 and Assumption 2.2 be satisfied. Then, for any $x \in \mathcal{X}$, $s \in \mathcal{S}$, and $u \in \mathcal{U}$, and under $\pi_t^*$ for $t \ge 0$, there exists $\overline{h}_{i,t+1} \in \mathbb{R}^n$ and $\overline{q}_{i,t+1} \in \mathbb{R}$ such that*

$$Q_t^*(x, s, u) = c(x, s, u) + (Ax + Bu)^\mathsf{T}G_{t+1}(Ax + Bu) + \sum_{i=1}^d \phi_i(s)\left(2(Ax + Bu)^\mathsf{T}\overline{h}_{i,t+1} + \overline{q}_{i,t+1}\right) + \sum_{i=t+1}^T \mathrm{tr}[G_i \Sigma_w].$$

$$\tag{19}$$

*Further,*

$$V_t^*(x, s) = x^\mathsf{T} G_t x + 2h_t^\mathsf{T}(s)x + q_t(s) + \sum_{i=t+1}^{T} \mathrm{tr}[G_i \Sigma_w], \tag{20}$$

*where $G_t \in \mathbb{R}^{n \times n} \succ 0$, $h_t(\cdot) \in \mathbb{R}^n$, and $q_t(\cdot) \in \mathbb{R}$ are as in Theorem 3.1.*

*Proof.* The proof follows in a similar manner as the proof of Theorem 3.1 and noting that $\mathbb{E}_w[w_t] = 0$ and $\mathbb{E}_w\left[w_t w_t^\mathsf{T}\right] = \Sigma_w$, for $t \in \{0, 1, \cdots, T-1\}$. □

### 4.2 Learning weight parameters of the value function via least-squares value iteration

In this subsection, we extend the least-squares value iteration procedure in Section 3.2 to the setting with process noise in eq. (18). The key observation is that the parametric structure of the optimal $Q$-function remains the same, up to an additive constant term. In particular, from Theorem 4.1, for any $x \in \mathcal{X}$, $s \in \mathcal{S}$, $u \in \mathcal{U}$ and $t \in \{0, \ldots, T-1\}$, we can write

$$Q_t(x, s, u) = c(x, s, u) + (Ax + Bu)^\top G_{t+1}(Ax + Bu) + \phi(s)^\top Y(x, u)\theta_{t+1} + \sum_{i=t+1}^{T} \mathrm{tr}[G_i \Sigma_w],$$

$$u_t(x, s) = K_{x,t} x + K_{s,t} s + K_{h,t}\left(\phi(s)^\mathsf{T} \otimes Z\right)\theta_{t+1}, \tag{21}$$

where $\theta_{t+1}$ and $Y(x, u) = I_d \otimes \left[2(Ax + Bu)^\top, 1\right]$ are the same as in Section 3.2. Notice that the additional term $\mathrm{tr}(G_{t+1}\Sigma_w)$ does not depend on the input, and therefore does not affect the minimization over $u$. We follow the same steps as Algorithm 1. For episode $\ell$, as in Section 3.2, we treat the data collected in the previous $\ell - 1$ episodes as a fixed dataset $\mathcal{D}^{\ell-1}$. At each time step $t$, the LSVI update fits the parametrized $Q$-function in eq. (21) to a Bellman target. Thus, $\theta$ is obtained by solving the following regularized least-squares regression problem (see Appendix F for details):

$$\theta_{t+1}^\ell = \arg\min_\theta \sum_{j=1}^{\ell-1} \left( (Ax_t^j + Bu_t^j)^\mathsf{T} G_{t+1}(Ax_t^j + Bu_t^j) + \phi(s_t^j)^\mathsf{T} Y(x_t^j, u_t^j)\theta + \sum_{i=t+1}^{T} \mathrm{tr}[G_i \Sigma_w]\right.$$

$$\left. - (x_{t+1}^j)^\mathsf{T} G_{t+1} x_{t+1}^j - 2\left(h_{t+1}^\ell(s_{t+1}^j)\right)^\mathsf{T} x_{t+1}^j - q_{t+1}^\ell(s_{t+1}^j) - \sum_{i=t+2}^{T} \mathrm{tr}[G_i \Sigma_w]\right)^2 + \lambda\|\theta\|_2^2, \tag{22}$$

where $h_{t+1}^\ell(\cdot)$ and $q_{t+1}^\ell(\cdot)$ are defined as in eq. (13) and eq. (14), respectively. The closed-forrm parameter update is therefore given by

$$\theta_{t+1}^\ell = \Lambda_t^{-1} \sum_{j=1}^{\ell-1} Y^\mathsf{T}(x^j, u^j)\phi(s^j)\epsilon^\ell(x_t^i, x_{t+1}^i, s_{t+1}^i, u_t^i),$$

$$\Lambda_t = \sum_{j=1}^{\ell-1} Y(x^j, u^j)^\mathsf{T} \phi(s^j)\phi(s^j)^\mathsf{T} Y(x^j, u^j) + \lambda I_{d(n+1)}, \tag{23}$$

$$\epsilon^\ell(x_t^i, x_{t+1}^i, s_{t+1}^i, u_t^i) = (x_{t+1}^i)^\mathsf{T} G_{t+1} x_{t+1}^i + 2\left(h_{t+1}^\ell(s_{t+1}^i)\right)^\mathsf{T} x_{t+1}^i + q_{t+1}^\ell(s_{t+1}^i)$$

$$- (Ax_t^i + Bu_t^i)^\mathsf{T} G_{t+1}(Ax_t^i + Bu_t^i) - \mathrm{tr}(G_{t+1}\Sigma_w).$$

Thus, the only modification to Algorithm 1 is in Step 7, where we compute $\epsilon_{t+1}^\ell$ as in eq. (23).

## 5 Extension to Unknown System Dynamics

In this section, we relax the assumption that $A$ and $B$ in eq. (1) are known. We begin by re-writing the $Q$-function in eq. (6) as

$$Q_t^*(x, s, u) = c(x, s, u) + \begin{bmatrix} x \\ u \end{bmatrix}^\mathsf{T} \underbrace{\begin{bmatrix} A^\mathsf{T} G_{t+1} A & A^\mathsf{T} G_{t+1} B \\ B^\mathsf{T} G_{t+1} A & B^\mathsf{T} G_{t+1} B \end{bmatrix}}_{P_{t+1}} \underbrace{\begin{bmatrix} x \\ u \end{bmatrix}}_{z} + 2 \sum_{i=1}^d \phi_i(s) \begin{bmatrix} x \\ u \end{bmatrix}^\mathsf{T} \underbrace{\begin{bmatrix} A^\mathsf{T} \\ B^\mathsf{T} \end{bmatrix} \overline{h}_{i,t+1}}_{\widetilde{h}_{i,t+1}} + \sum_{i=1}^d \phi_i(s) \overline{q}_{i,t+1}.$$
(24)

Equation (24) preserves the same structural form as eq. (6). The difference is that in eq. (6) we assumed $A$ and $B$ were known, so only $\overline{h}_{t+1}$ and $\overline{q}_{t+1}$ were treated as unknowns. In contrast, when the dynamics are unknown, the matrices $A$ and $B$ become implicitly embedded inside the quantities $P_{t+1}$ and $\tilde{h}_{t+1}$. Thus, the roles previously played by $\overline{h}_{t+1}$ and $\overline{q}_{t+1}$ in eq. (6) are now taken over by the enlarged set of parameters, $P_{t+1}$, $\tilde{h}_{t+1}$, and $\overline{q}_{t+1}$, all of which must be learned from data. Similarly, we re-write the policy in Corollary 3.2 as

$$u_t^*(x, s) = K_{x,t} x + K_{s,t} s + \widetilde{K}_{h,t} \sum_{i=1}^d \phi_i(s) \widetilde{h}_{i,21,t+1}.$$
(25)

where $\widetilde{h}_{i,21} \in \mathbb{R}^m$ is the $(2, 1)$-block of $\widetilde{h}_i \in \mathbb{R}^{n+m}$ in eq. (24), and

$$\begin{aligned} K_{x,t} &= -(R + P_{22,t+1})^{-1} \left( D^\mathsf{T} + P_{21,t+1} \right), \\ K_{s,t} &= -(R + P_{22,t+1})^{-1} H^\mathsf{T}, \\ \widetilde{K}_{h,t} &= -(R + P_{22,t+1})^{-1}, \end{aligned}$$
(26)

where $P_{21} \in \mathbb{R}^{m \times n}$ and $P_{22} \in \mathbb{R}^{m \times m}$ are the $(2, 1)$-block and the $(2, 2)$-block of the matrix $P$ in eq. (24), respectively. The details of learning the parameters, $P_t$, $\tilde{h}_t$, and $\overline{q}_t$, for $t \in \{0, \dots, T\}$, together with the corresponding least-squares value iteration algorithm extending Algorithm 1 to the case where $A$ and $B$ are unknown, are presented in Appendix G with resulting method summarized in Algorithm 2.

**Remark 4.** *(The need for exploration) In contrast to Algorithm 1, a key aspect of Algorithm 2 is that the input $u_t$ in Line 22 involves an additional excitation term. This term is necessary to sufficiently excite the system so that the collected data are informative enough to learn the unknown system matrices $A$ and $B$.*

**Remark 5.** *(Stability and convergence) Unlike Algorithm 1, for Algorithm 2 neither closed-loop stability nor convergence is guaranteed in general, especially when the underlying system is unstable. Since the system matrices $A$ and $B$ are unknown and must be learned online, the exploratory inputs and estimation errors may affect both stabilization and parameter convergence during the learning process. Therefore, further analysis is needed to establish conditions under which Algorithm 2 guarantees stability and convergence.*

## 6 Illustration of Results

### 6.1 Known System Dynamics

We consider a discrete-time, linear, time-invariant system

$$x_{t+1} = \underbrace{\begin{bmatrix} 1.8 & 1.2 \\ 0 & 1.19 \end{bmatrix}}_{A} x_t + \underbrace{\begin{bmatrix} 0 \\ 1 \end{bmatrix}}_{B} u_t,$$
(27)

and a stochastic environment evolving according to a feature-based linear Markov process with

$$s_{t+1} | s_t \sim \underbrace{\begin{bmatrix} \frac{f_1(s_t)}{f_1(s_t) + f_1(s_t)} & \frac{f_2(s_t)}{f_1(s_t) + f_1(s_t)} \end{bmatrix}}_{\phi(s_t)^T} \underbrace{\begin{bmatrix} \mathcal{N}(s_{t+1}; 7, 1) \\ \mathcal{N}(s_{t+1}; -1, 1.5) \end{bmatrix}}_{\mu_{t+1}(s_{t+1})},$$
(28)

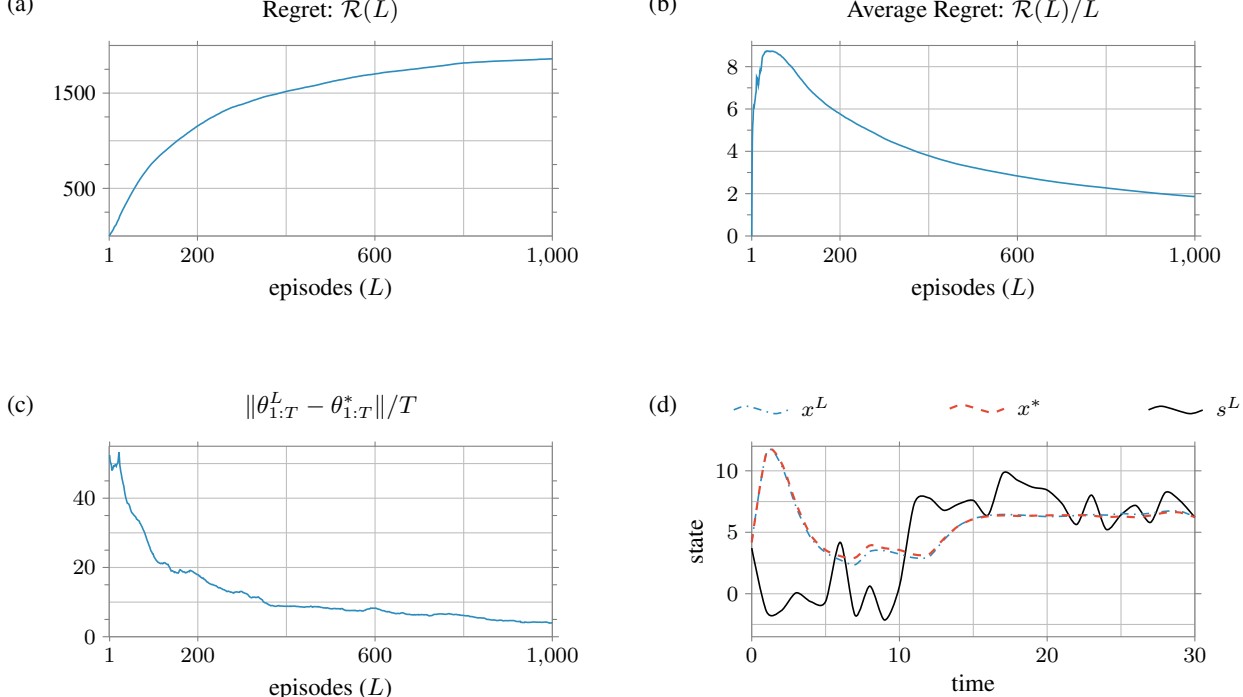

Figure 1: This figure shows the numerical results for the setting in Section 6.1. Panel (a) shows the regret as a function of $L$, scaling as $\widetilde{\mathcal{O}}(\sqrt{L})$ in line with Theorem 3.4. Panel (b) shows the average regret as a function of $L$, and we observe that it converges as $L$ increases. Panel (c) shows the norm of the estimation error between the learned and the true parameters, averaged over the episode horizon $T$, as a function of $L$. We observe that the estimation error decreases with $L$, indicating that the learned policy converges to the optimal one. Panel (d) shows the state trajectory generated by the system in eq. (27) under the learned policy at episode $L = 1000$ (dot-dashed blue line) and under the optimal policy (dashed red line). It also shows the exogenous state trajectory generated by the linear Markov process in eq. (28) (solid black line). We observe that the trajectory under the learned policy closely matches that of the optimal policy, and both track the mean of the exogenous state.

where $f_1(s_t) = \exp\left(\frac{-(s_t - \nu_1)^2}{2\rho_1^2}\right)$ and $f_2(s_t) = \exp\left(\frac{-(s_t - \nu_2)^2}{2\rho_2^2}\right)$, where $\nu_1 = 7$, $\nu_2 = -1$, $\rho_1 = 5$, and $\rho_2 = 3$, with $\|s_t\| \leq \delta_s = 15$ for all $t$. We define the cost function to capture the tracking error between the first state of eq. (27) and the exogenous state, $s$, and is expressed as

$$c(x, s, u) = (Cx - s)^\mathsf{T} M (Cx - s) + u^\mathsf{T} R u = x^\mathsf{T} \underbrace{C^\mathsf{T} M C}_{W} x + s^\mathsf{T} M s + u^\mathsf{T} R u - 2s \underbrace{M C}_{F^\mathsf{T}} x, \qquad (29)$$

where $C = [1, 0]$, $M = 1$, and $R = 1$. First, we use the matrices $A$ and $B$, along with the cost weight matrices, to compute the feedback gains in Corollary 3.2. Then, we apply Algorithm 1 to learn the parameters, $\theta$, using $L = 1000$ episodes, each with horizon $T = 30$. We set $\lambda = 2$ and $R_\theta = 500$ (see Remark 3). At each episode, we sample $x_0 \overset{\text{i.i.d.}}{\sim} \mathcal{N}(0, 4)$ and $s_0 \overset{\text{i.i.d.}}{\sim} \mathcal{N}(0, 4)$, which are independent of each other. Using knowledge of the true distributions in eq. (28), we compute the true parameters, $\theta_t^*$ for $t \in \{1, \cdots, T\}$, via the results in Appendix C.1, which we then use to compute the true optimal policy $\pi_t^*$ as in Corollary 3.2. Finally, we use the true parameters, we compute the regret in eq. (17). We present our numerical results in Figure 1. The regret $\mathcal{R}(L)$ as a function of the number of episodes $L$ is shown in Fig. 1(a). We observe that the regret scales as $\widetilde{\mathcal{O}}(\sqrt{L})$, which is consistent with our results in Theorem 3.4. The average regret, $\mathcal{R}(L)/L$ as a function of $L$ is shown in Figure 1(b) and is observed to converge as $L$ increases, indicating convergence of our algorithm. Fig 1(c) presents the norm of the estimation error between the learned and the true parameters, averaged

over the episode horizon $T$, as a function of $L$. This is expressed as $\|\theta_{1:T}^L - \theta_{1:T}^*\|/T$, where $\theta_{1:T}^L \in \mathbb{R}^{d(n+1) \times T}$ is a matrix whose columns corresponds to the parameters $\theta_t^L$ at each time step $t$, and similarly for $\theta_{1:T}^*$. We observe that the estimation error decreases with $L$, indicating that the learned policy gradually converges to the optimal one. Finally, we apply both the learned policy at episode $L = 1000$ and the optimal policy to the system in eq. (27), and compare their corresponding closed-loop state trajectories, as shown in Figure 1(d), alongside the trajectory of the exogenous state $s$. We observe that the trajectory under the learned policy closely matches that of the optimal policy, and both effectively track the mean of the exogenous state.

## 6.2 Known System Dynamics with Process Noise

In this subsection, we consider the same setting as in Section 6.1, except that the system in eq. (27) is affected by additive process noise. In particular, we consider

$$x_{t+1} = Ax_t + Bu_t + w_t, \qquad t \in \{0, 1, \ldots, T-1\}, \tag{30}$$

where $w_t \overset{\text{i.i.d.}}{\sim} \mathcal{N}(0, \Sigma_w)$ with $\Sigma_w \succ 0$. We use the same system matrices $A$ and $B$ as in eq. (27), the same stochastic environment in eq. (28), and the same quadratic tracking cost in eq. (29). We set $\Sigma_w = I_2$. We first use the known matrices $A$ and $B$, together with the cost weight matrices, to compute the feedback gains in Corollary 3.2. Then, we implement Algorithm 1 with the modified Bellman target in eq. (23) to learn the parameters $\theta$. We use $L = 1000$ episodes, each with horizon $T = 30$, and set $\lambda = 2$ and $R_\theta = 500$. At each episode, we sample $x_0 \overset{\text{i.i.d.}}{\sim} \mathcal{N}(0, 25)$ and $s_0 \overset{\text{i.i.d.}}{\sim} \mathcal{N}(0, 4)$, independently of each other. Since the additive term $\sum_{i=t+1}^{T} \text{tr}[G_i \Sigma_w]$ in Theorem 4.1 does not depend on the control input, the optimal policy retains the same closed-form structure as in Corollary 3.2, while $\theta$ is learned as in eq. (23). Using knowledge of the true distributions in eq. (28), we compute the true parameters, $\theta_t^*$ for $t \in \{1, \cdots, T\}$, via the results in Appendix C.1, which we then use to compute the true optimal policy $\pi_t^*$ as in Corollary 3.2. Finally, we use the true parameters, we compute the regret in eq. (17).

We present our numerical results in Figure 2. The results show that Algorithm 1 continues to learn an effective control policy in the presence of process noise. In particular, the regret $\mathcal{R}(L)$ shown in Figure 2(a) grows sublinearly with the number of episodes $L$. The average regret shown in Figure 2(b) decreases as $L$ increases. Figure 2(c) presents the norm of the estimation error between the learned and the true parameters, averaged over the episode horizon $T$, as a function of $L$. This is expressed as $\|\theta_{1:T}^L - \theta_{1:T}^*\|/T$, where $\theta_{1:T}^L \in \mathbb{R}^{d(n+1) \times T}$ is a matrix whose columns corresponds to the parameters $\theta_t^L$ at each time step $t$, and similarly for $\theta_{1:T}^*$. We observe that the estimation error decreases with $L$, indicating that the learned policy gradually converges to the optimal one. Finally, we apply both the learned policy at episode $L = 1000$ and the optimal policy to the system in eq. (27), and compare their corresponding closed-loop state trajectories, as shown in Figure 2(d), alongside the trajectory of the exogenous state $s$. We observe that the trajectory under the learned policy closely matches that of the optimal policy, and both effectively track the mean of the exogenous state.

## 6.3 Unknown System Dynamics

In this subsection, we consider the same setting as in Section 6.1, except that the system matrices $A$ and $B$ in eq. (27) are assumed to be unknown. Consequently, we need to learn the parameters $P_t$, $\tilde{h}_t$, and $\bar{q}_t$ in eq. (24) jointly from data. These parameters can be combined into a single vector $\theta$, as described in eq. (137). We use the following stable linear system (see Remark 5)

$$x_{t+1} = \underbrace{\begin{bmatrix} 0.4 & 1.2 \\ 0 & 0.19 \end{bmatrix}}_{A} x_t + \underbrace{\begin{bmatrix} 0 \\ 1 \end{bmatrix}}_{B} u_t, \tag{31}$$

and the same stochastic environment and quadratic tracking cost as in eq. (28) and eq. (29), respectively. We implement Algorithm 2 with $L = 1500$ episodes and horizon $T = 30$. We set the regularization parameter to $\lambda = 2$ and the projection radius to $R_\theta = 800$. At each episode, we sample $x_0 \overset{\text{i.i.d.}}{\sim} \mathcal{N}(0, 25)$ and $s_0 \overset{\text{i.i.d.}}{\sim} \mathcal{N}(0, 25)$, independent of each other. We use an exploration term $\eta_t \overset{\text{i.i.d.}}{\sim} \mathcal{N}(0, 25)$ for $t \in \{0, \cdots, T\}$. Similarly as in

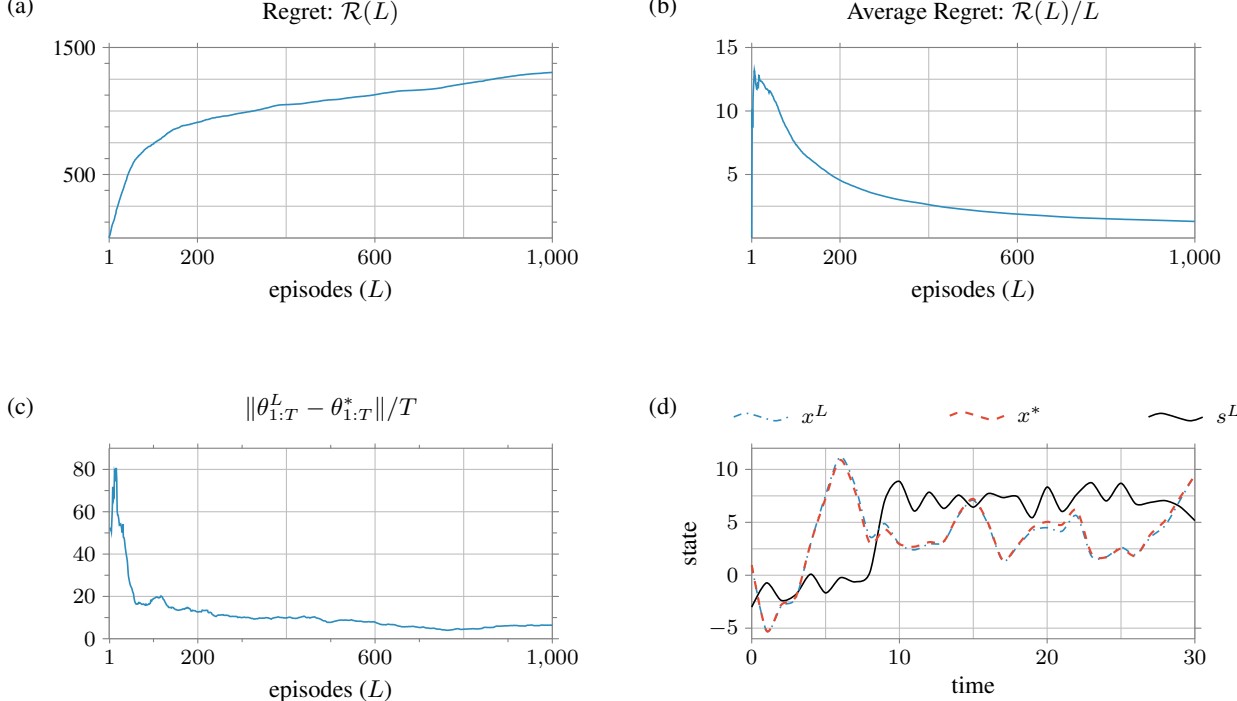

Figure 2: This figure shows the numerical results for the setting in Section 6.2. Panel (a) shows the regret as a function of $L$, scaling sublinearly with $L$. Panel (b) shows the average regret as a function of $L$, and we observe that it converges as $L$ increases. Panel (c) shows the norm of the estimation error between the learned and the true parameters, averaged over the episode horizon $T$, as a function of $L$. We observe that the estimation error decreases with $L$, indicating that the learned policy converges to the optimal one. Panel (d) shows the state trajectory generated by the system in eq. (27) under the learned policy at episode $L = 1000$ (dot-dashed blue line) and under the optimal policy (dashed red line). It also shows the exogenous state trajectory generated by the linear Markov process in eq. (28) (solid black line). We observe that the trajectory under the learned policy closely matches that of the optimal policy, and both track the mean of the exogenous state.

Section 6.1, we use the knowledge of dynamics in eq. (31) and the true distributions in eq. (28) to compute the true parameters, $\theta_t^*$ for $t \in \{1, \cdots, T\}$, via the results in Appendix C.1. These are then used to compute the true optimal policy $\pi_t^*$ as in Corollary 3.2. Finally, we use the true parameters, we compute the regret in eq. (17).

We present our numerical results in Figure 3. The results show that Algorithm 2 is able to learn an effective control policy even when the system matrices $A$ and $B$ are unknown. In particular, the regret $\mathcal{R}(L)$ shown in Figure 3(a) grows sublinearly with the number of episodes $L$. The average regret shown in Figure 3(b) decreases as $L$ increases. Figure 3(c) presents the norm of the estimation error between the learned and the true parameters, averaged over the episode horizon $T$, as a function of $L$. This is expressed as $\|\theta_{1:T}^L - \theta_{1:T}^*\|/T$, where $\theta_{1:T}^L \in \mathbb{R}^{n_\theta \times T}$ is a matrix whose columns correspond to the learned parameters $\theta_t^L$ at each time step $t$ with $n_\theta = ((n+m+1)d + (n+m)(n+m+1))/2 = 14$, and similarly for $\theta_{1:T}^*$. We observe that the estimation error decreases with $L$, indicating that the learned parameters gradually converge to their true values, and consequently that the learned policy approaches the optimal one. Finally, we apply both the learned policy at episode $L = 1500$ and the optimal policy to the system in eq. (31), and compare their corresponding closed-loop state trajectories, as shown in Figure 3(d), alongside the trajectory of the exogenous state $s$. We observe that the trajectory under the learned policy closely matches that of the optimal policy, showing that Algorithm 2 successfully recovers near-optimal closed-loop behavior despite the unknown system dynamics.

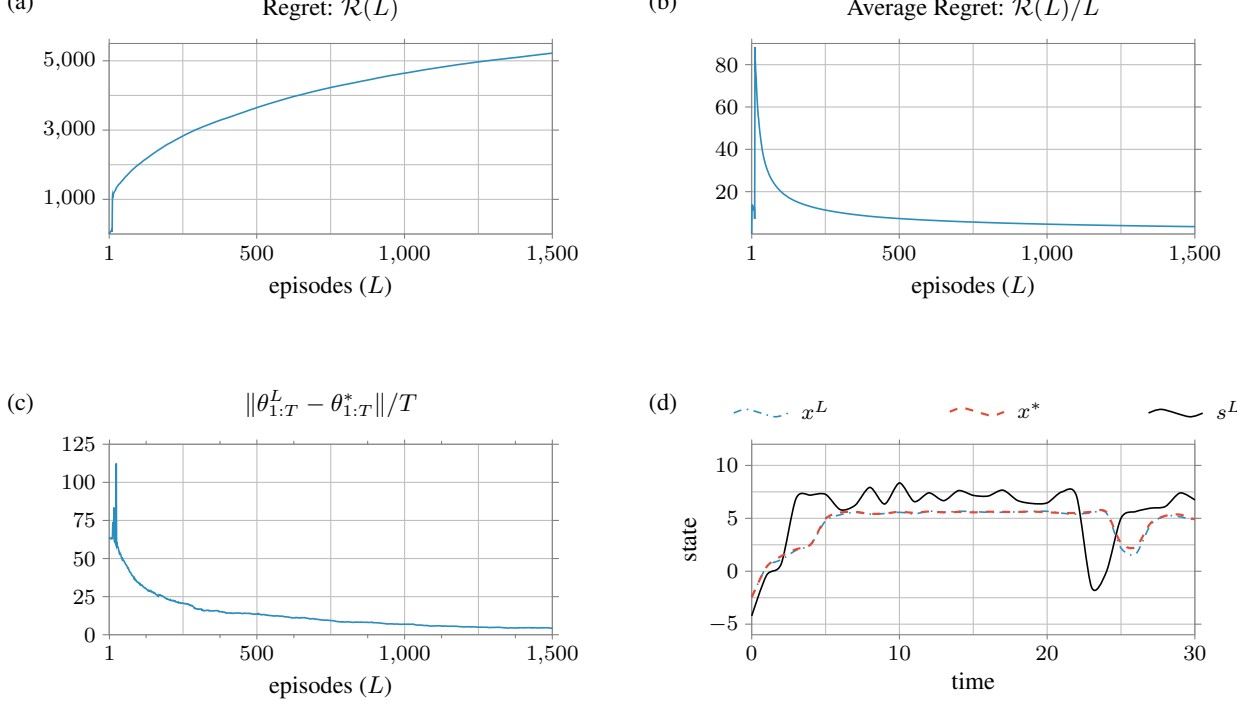

Figure 3: This figure shows the numerical results for the setting in Section 6.3. Panel (a) shows the regret as a function of $L$, scaling sublinearly with $L$. Panel (b) shows the average regret as a function of $L$, and we observe that it converges as $L$ increases. Panel (c) shows the norm of the estimation error between the learned and the true parameters, averaged over the episode horizon $T$, as a function of $L$. We observe that the estimation error decreases with $L$, indicating that the learned policy converges to the optimal one. Panel (d) shows the state trajectory generated by the system in eq. (31) under the learned policy at episode $L = 1500$ (dot-dashed blue line) and under the optimal policy (dashed red line). It also shows the exogenous state trajectory generated by the linear Markov process in eq. (28) (solid black line). We observe that the trajectory under the learned policy closely matches that of the optimal policy, and both track the mean of the exogenous state.

We have also observed that the regret may not converge if the open loop system is unstable (i.e., $A$ has eigenvalues with magnitude greater than 1). Additional enhancements to Algorithm 2 are needed to ensure closed-loop stability. We leave it for future exploration.

## 7   Conclusion

In this work, we proposed a reinforcement learning framework that unifies linear control systems and feature-based linear Markov models, capturing both deterministic system dynamics and stochastic environmental effects. By leveraging this structure, we derived closed-form expressions for the optimal value function and policy, and introduced a least-squares value iteration algorithm that learns the optimal control policy without requiring explicit model identification or exploration. We provided theoretical guarantees on stability and convergence, and demonstrated the effectiveness of our approach through numerical simulations. We also extended the framework to systems with process noise and to the case of unknown system dynamics. Future directions include establishing stability and convergence guarantees for the these extended settings, as well as developing extensions beyond linear system structure.

**Broader Impact Statement**

This work develops a reinforcement learning framework for controlling linear dynamical systems that interact with stochastic environments, with potential applications in areas such as power systems, robotics, and other cyber-physical systems. The proposed framework may have a positive impact by enabling more adaptive and computationally efficient control under uncertainty, while also incorporating stability considerations that are important in safety-critical applications.

At the same time, like many learning-based control methods, this work could have negative consequences if deployed without sufficient validation and safety safeguards. In particular, the proposed approach is developed under specific modeling assumptions, and its theoretical guarantees are established for the nominal setting. Although we also consider extensions involving Gaussian process noise and unknown system dynamics, these settings are illustrated numerically and are not yet supported by corresponding stability and convergence guarantees. As a result, applying our approach directly to real-world safety-critical systems without additional verification, robustness analysis, and domain-specific safeguards may lead to unsafe or unreliable behavior.

We therefore view this work as a step toward more principled learning-based control, while emphasizing that practical deployment should be accompanied by careful system identification, validation, monitoring, and human oversight.

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

# A  Proof of Theorem 3.1

We first present the following result in which we provide an expression for the optimal greedy policy, $\pi_t^*$, and the optimal value function, $V_t^*(x, s)$ under the greedy policy.

**Theorem A.1.** *(optimal policy and value function) Consider the dynamics in eq. (1) and the Markov Process in eq. (2). Let Assumption 2.1 be satisfied. Then, for any $x \in \mathcal{X}$, $s \in \mathcal{S}$, $s' \sim \mathbb{P}(.|s)$, and $t \geq 0$*

$$u^*(x, s, t) = \underbrace{K_{x,t}x + K_{h,t}\mathbb{E}\left[h_{t+1}(s')|s\right] + K_{s,t}s}_{\pi_t^*(x,s)}, \tag{32}$$

*where*

$$
\begin{aligned}
K_{x,t} &= -\left(R + B^{\mathsf{T}}G_{t+1}B\right)^{-1}\left(B^{\mathsf{T}}G_{t+1}A + D^{\mathsf{T}}\right), \\
K_{h,t} &= -\left(R + B^{\mathsf{T}}G_{t+1}B\right)^{-1}B^{\mathsf{T}}, \\
K_{s,t} &= -\left(R + B^{\mathsf{T}}G_{t+1}B\right)^{-1}H^{\mathsf{T}}.
\end{aligned}
\tag{33}
$$

*Further,*

$$V_t^*(x, s) = x^{\mathsf{T}}G_t x + 2h_t^{\mathsf{T}}(s)x + q_t(s), \tag{34}$$

*where $G_t \in \mathbb{R}^{n \times n} \succ 0$, $h_t(\cdot) \in \mathbb{R}^n$, and $q_t(\cdot) \in \mathbb{R}$ satisfy*

$$G_t = A^{\mathsf{T}}G_{t+1}A + W - (A^{\mathsf{T}}G_{t+1}B + D)(R + B^{\mathsf{T}}G_{t+1}B)^{-1}(B^{\mathsf{T}}G_{t+1}A + D^{\mathsf{T}}), \tag{35}$$

$$h_t(s_t) = \left(A^{\mathsf{T}} + K_{x,t}^{\mathsf{T}}B^{\mathsf{T}}\right)\mathbb{E}\left[h_{t+1}(s_{t+1})|s_t\right] + \left(F + K_{x,t}^{\mathsf{T}}H^{\mathsf{T}}\right)s_t, \tag{36}$$

$$
\begin{aligned}
q_t(s_t) &= \mathbb{E}\left[q_{t+1}(s_{t+1})|s_t\right] + s_t^{\mathsf{T}}(M + HK_{s,t})s_t + \mathbb{E}\left[h_{t+1}^{\mathsf{T}}(s_{t+1})|s_t\right]BK_{h,t}\mathbb{E}\left[h_{t+1}(s_{t+1})|s_t\right] \\
&\quad + 2s_t^{\mathsf{T}}HK_{h,t}\mathbb{E}\left[h_{t+1}(s_{t+1})|s_t\right],
\end{aligned}
\tag{37}
$$

*with $G_T = W$, and $h_T(s_T) = Fs_T$ and $q_T(s_T) = s_T^{\mathsf{T}}Ms_T$.*

*Proof.* We prove our claim by induction. For notational convenience, we drop the time index from the states and inputs inside the expressions and arguments of $c(\cdot)$, $V_t^*(\cdot)$, and $Q_t^*(\cdot)$ for $t \geq 0$, where we use $x$, $s$, $u$, $x'$, and $s'$ to denote $x_t$, $s_t$, $u_t$, $x_{t+1}$, and $s_{t+1}$, respectively. At $t = T - 1$,

$$
\begin{aligned}
Q_{T-1}^*(x, s, u) &= c(x, s, u) + \mathbb{E}_{s' \sim \mathbb{P}_t(s'|s)}\left[V_T^*(x', s')|x, s, u\right] \\
&= x^{\mathsf{T}}(W + A^{\mathsf{T}}WA)x + u^{\mathsf{T}}(R + B^{\mathsf{T}}WB)u + 2(s^{\mathsf{T}}F^{\mathsf{T}} + \mathbb{E}\left[s'|s\right]^{\mathsf{T}}F^{\mathsf{T}}A)x \\
&\quad + 2\left(x^{\mathsf{T}}\left(D + A^{\mathsf{T}}WB\right) + s^{\mathsf{T}}H + \mathbb{E}\left[s'|s\right]^{\mathsf{T}}F^{\mathsf{T}}B\right)u + s^{\mathsf{T}}Ms + \mathbb{E}\left[s'^{\mathsf{T}}Ms'|s\right],
\end{aligned}
\tag{38}
$$

where we used the fact that $V_T^*(x_T, s_T) = c(x_T, s_T, u_T)$ and $u_T = 0$. Taking the derivative of $Q_{T-1}^*$ with respect to $u$

$$\frac{\partial Q_{T-1}^*(x, s, u)}{\partial u} = 2(R + B^{\mathsf{T}}WB)u + 2(B^{\mathsf{T}}WA + D^{\mathsf{T}})x + 2(B^{\mathsf{T}}F\mathbb{E}\left[s'|s\right] + H^{\mathsf{T}}s).$$

Setting the above derivative to zero and solving for $u$, we get

$$u_{T-1}^* = -(R + B^{\mathsf{T}}WB)^{-1}(B^{\mathsf{T}}WA + D^{\mathsf{T}})x_{T-1} - (R + B^{\mathsf{T}}WB)^{-1}(B^{\mathsf{T}}F\mathbb{E}\left[s_T|s_{T-1}\right] + H^{\mathsf{T}}s_{T-1}), \tag{39}$$

which is the minimizer of eq. (38). We substitute eq. (39) in eq. (38),

$$V_{T-1}^*(x, s) = x^{\mathsf{T}}G_{T-1}x + 2h_{T-1}(s)^{\mathsf{T}}x + q_{T-1}(s), \tag{40}$$

where $G_{T-1}$, $h_{T-1}(s)$, and $q_{T-1}(s)$ are as in Theorem A.1 for $t = T - 1$. Suppose for $t = k + 1$,

$$V_{k+1}^*(x, s) = x^{\mathsf{T}}G_{k+1}x + 2h_{k+1}(s)^{\mathsf{T}}x + q_{k+1}(s),$$

where $G_{k+1}$, $h_{k+1}(s)$, and $q_{k+1}(s)$ are as in Theorem A.1 for $t = k + 1$. Then we have,

$$\begin{aligned}
Q_k^*(x, s, u) &= c(x, s, u) + \mathbb{E}_{s' \sim \mathbb{P}_t(s'|s)} \left[ V_{k+1}^*(x', s')|x, s, u \right] \\
&= x^\mathsf{T}(W + A^\mathsf{T} G_{k+1} A)x + u^\mathsf{T}(R + B^\mathsf{T} G_{k+1} B)u + 2(s^\mathsf{T} F^\mathsf{T} + \mathbb{E}\left[h_{k+1}(s')|s\right]^\mathsf{T} A)x \\
&\quad + 2\left(x^\mathsf{T}(D + A^\mathsf{T} G_{k+1} B) + s^\mathsf{T} H + \mathbb{E}\left[h_{k+1}(s')|s\right]^\mathsf{T} B\right)u + s^\mathsf{T} M s + \mathbb{E}\left[q_{k+1}(s')|s\right],
\end{aligned}$$

(41)

Taking the derivative of $Q_k^*$ with respect to $u$

$$\frac{\partial Q_k^*(x, s, u)}{\partial u} = 2(R + B^\mathsf{T} G_{k+1} B)u + 2(B^\mathsf{T} G_{k+1} A + D^\mathsf{T})x + 2\left(B^\mathsf{T} \mathbb{E}\left[h_{k+1}(s')|s\right] + H^\mathsf{T} s\right).$$

Setting the above derivative to zero and solving for $u$, we get

$$\begin{aligned}
u_k^* = &- (R + B^\mathsf{T} G_{k+1} B)^{-1}(B^\mathsf{T} G_{k+1} A + D^\mathsf{T})x_k - (R + B^\mathsf{T} G_{k+1} B)^{-1} H^\mathsf{T} s_k \\
&- (R + B^\mathsf{T} G_{k+1} B)^{-1} B^\mathsf{T} \mathbb{E}\left[h_{k+1}(s(k+1))|s_k\right],
\end{aligned}$$

(42)

which is the minimizer of eq. (41). We substitute eq. (42) in eq. (41),

$$V_k^*(x, s) = x^\mathsf{T} G_k x + 2h_k(s)^\mathsf{T} x + q_k(s),$$

where $G_k$, $h_k(s)$, and $q_k(s)$ are as in Theorem A.1 for $t = k$. This completes the proof. $\square$

*Proof of Theorem 3.1:* For notational convenience, we drop the time index from the states and inputs inside the expressions and arguments of $c(\cdot)$, $V_t^*(\cdot)$, and $Q_t^*(\cdot)$ for $t \geq 0$, where we use $x$, $s$, $u$, $x'$, and $s'$ to denote $x_t$, $s_t$, $u_t$, $x_{t+1}$, and $s_{t+1}$, respectively. Using Theorem A.1, we write

$$\begin{aligned}
Q_t^*(x, s, u) &= c(x, s, u) + \mathbb{E}_{s' \sim \mathbb{P}_t(s'|s)} \left[ V_{t+1}^*(x', s')|x, s, u \right] \\
&= c(x, s, u) + \int_{\mathcal{S}} V_{t+1}^*(Ax + Bu, s') \mathbb{P}_t(\mathrm{d}s'|s) \\
&\overset{(a)}{=} c(x, s, u) + \int_{\mathcal{S}} V_{t+1}^*(Ax + Bu, s') \phi(s)^\mathsf{T} \mu_t(\mathrm{d}s') \\
&\overset{(b)}{=} c(x, s, u) + (Ax + Bu)^\mathsf{T} G_{t+1}(Ax + Bu) + \int_{\mathcal{S}} (2h_{t+1}(s')^\mathsf{T}(Ax + Bu) + q_{t+1}(s')) \sum_{i=1}^{d} \phi_i(s) \mu_{i,t}(\mathrm{d}s') \\
&= c(x, s, u) + (Ax + Bu)^\mathsf{T} G_{t+1}(Ax + Bu) + \sum_{i=1}^{d} \phi_i(s) \int_{\mathcal{S}} q_{t+1}(s') \mu_{i,t}(\mathrm{d}s') \\
&\quad + 2\sum_{i=1}^{d} \left(\phi_i(s) \int_{\mathcal{S}} h_{t+1}(s')^\mathsf{T} \mu_{i,t}(\mathrm{d}s')\right)(Ax + Bu) \\
&= c(x, s, u) + (Ax + Bu)^\mathsf{T} G_{t+1}(Ax + Bu) + \sum_{i=1}^{d} \phi_i(s) \underbrace{\mathbb{E}_{\mu_{i,t}}\left[q_{t+1}(s')\right]}_{\bar{q}_{i,t+1}} \\
&\quad + 2\sum_{i=1}^{d} \left(\phi_i(s) \underbrace{\mathbb{E}_{\mu_{i,t}}\left[h_{t+1}(s')\right]^\mathsf{T}}_{\bar{h}_{i,t+1}^\mathsf{T}}\right)(Ax + Bu),
\end{aligned}$$

where in step (a) we have used Assumption 2.1, and in step (b) we have used Theorem A.1. $\blacksquare$

## B    Least-squares value iteration

We formulate the regularized least squares regression and derive its solution presented in lines 6-7 of Algorithm 1. We begin by using the notation in eq. (9) to derive the expression of the parametrized state-action value

function, $Q_t$ and the corresponding parametrized optimal greedy policy in eq. (10). Using the expression of $Q_t$ in Theorem 3.1, we can write

$$
\begin{aligned}
Q_t(x,s,u) =& c(x,s,u) + (Ax+Bu)^\mathsf{T} G_{t+1}(Ax+Bu) + \sum_{i=1}^d \phi_i(s)\Big(2(Ax+Bu)^\mathsf{T}\overline{h}_{i,t+1} + \overline{q}_{i,t+1}\Big) \\
=& c(x,s,u) + (Ax+Bu)^\mathsf{T} G_{t+1}(Ax+Bu) + \sum_{i=1}^d \phi_i(s)\underbrace{\left[2(Ax+Bu)^\mathsf{T}\quad 1\right]}_{y(x,u)^\mathsf{T}}\underbrace{\begin{bmatrix}\overline{h}_{i,t+1}\\\overline{q}_{i,t+1}\end{bmatrix}}_{\theta_{i,t+1}} \\
=& c(x,s,u) + (Ax+Bu)^\mathsf{T} G_{t+1}(Ax+Bu) + \sum_{i=1}^d \phi_i(s)y(x,u)^\mathsf{T}\theta_{i,t+1} \\
=& c(x,s,u) + (Ax+Bu)^\mathsf{T} G_{t+1}(Ax+Bu) + \phi(s)^\mathsf{T}\underbrace{\left(I_d \otimes y(x,u)^\mathsf{T}\right)}_{Y(x,u)}\underbrace{\begin{bmatrix}\theta_{1,t+1}\\\vdots\\\theta_{d,t+1}\end{bmatrix}}_{\theta_{t+1}} \\
=& c(x,s,u) + (Ax+Bu)^\mathsf{T} G_{t+1}(Ax+Bu) + \phi(s)^\mathsf{T} Y(x,u)\theta_{t+1}.
\end{aligned}
\tag{43}
$$

Next, using the notation in eq. (9), we can write

$$
\begin{aligned}
\overline{h}_{i,t+1} =& \underbrace{\left[I_n\quad 0_{n\times 1}\right]}_{Z}\underbrace{\begin{bmatrix}\overline{h}_{i,t+1}\\\overline{q}_{i,t+1}\end{bmatrix}}_{\theta_{i,t+1}} = Z\theta_{i,t+1},\quad i\in\{1,\cdots,d\}, \\
\overline{h}_{t+1} =& \begin{bmatrix}Z & 0 & \cdots & 0\\0 & Z & \cdots & 0\\\vdots & \ddots & \ddots & \vdots\\0 & 0 & \cdots & Z\end{bmatrix}\begin{bmatrix}\theta_{1,t+1}\\\vdots\\\theta_{d,t+1}\end{bmatrix} = (I_d\otimes Z)\theta_{t+1}.
\end{aligned}
\tag{44}
$$

Similarly, we can write

$$
\begin{aligned}
\overline{q}_{i,t+1} =& \underbrace{\left[0_{1\times n}\quad 1\right]}_{\overline{Z}}\begin{bmatrix}\overline{h}_{i,t+1}\\\overline{q}_{i,t+1}\end{bmatrix} = \overline{Z}\theta_{i,t+1},,\quad i\in\{1,\cdots,d\}, \\
\overline{q}_{t+1} =& \begin{bmatrix}\overline{Z} & 0 & \cdots & 0\\0 & \overline{Z} & \cdots & 0\\\vdots & \ddots & \ddots & \vdots\\0 & 0 & \cdots & \overline{Z}\end{bmatrix}\begin{bmatrix}\theta_{1,t+1}\\\vdots\\\theta_{d,t+1}\end{bmatrix} = (I_d\otimes\overline{Z})\theta_{t+1}.
\end{aligned}
\tag{45}
$$

Next, from the expression of $u_t$ in Corollary 3.2, we write

$$
\begin{aligned}
u_t(x,s) =& K_{x,t}x + K_{s,t}s + K_{h,t}\sum_{i=1}^d \phi_i(s)\overline{h}_{i,t+1} \\
=& K_{x,t}x + K_{s,t}s + K_{h,t}\left[\phi_1(s)I_n\quad\cdots\quad\phi_d(s)I_n\right]\begin{bmatrix}\overline{h}_{1,t+1}\\\vdots\\\overline{h}_{d,t+1}\end{bmatrix} \\
=& K_{x,t}x + K_{s,t}s + K_{h,t}\left(\phi(s)^\mathsf{T}\otimes I_n\right)\overline{h}_{t+1} \\
\overset{(a)}{=}& K_{x,t}x + K_{s,t}s + \left(\phi(s)^\mathsf{T}\otimes I_n\right)(I_d\otimes Z)\theta_{t+1} \\
=& K_{x,t}x + K_{s,t}s + \left(\phi(s)^\mathsf{T}\otimes Z\right)\theta_{t+1},
\end{aligned}
\tag{46}
$$

where in step (a) we have used eq. (44). We define the Bellman target at time $t$ as

$$g_t(x, s, u) = c(x, s, u) + \min_v \widehat{Q}_{t+1}(x', s', v), \tag{47}$$

where $x'$ and $s'$ denote the states resulting from taking action $u$ in states $x$ and $s$, and $\widehat{Q}_{t+1}(x', s', v)$ is the estimate of the state-action value function at time $t+1$. We re-write eq. (48) as

$$
\begin{aligned}
g_t(x, s, u) =& c(x, s, u) + \widehat{V}_{t+1}^*(x', s') \\
\overset{(b)}{=}& c(x, s, u) + x'^\mathsf{T} G_{t+1} x' + 2\widehat{h}_{t+1}^\mathsf{T}(s')x' + \widehat{q}_{t+1}(s'),
\end{aligned} \tag{48}
$$

where in step (b), we have used Theorem A.1. For notational convenience, let $X_1(t) = A^\mathsf{T} + K_{x,t}^\mathsf{T} B^\mathsf{T}$, $X_2(t) = F + K_{x,t}^\mathsf{T} H^\mathsf{T}$, $Y_1(t) = M + H K_{s,t}$, $Y_2(t) = B K_{h,t}$, and $Y_3(t) = H K_{h,t}$. Using eq. (36) and eq. (37), we re-write $\widehat{h}_{t+1}$ and $\widehat{q}_{t+1}$ in eq. (48) as

$$
\begin{aligned}
\widehat{h}_{t+1}(s_{t+1}) =& X_1(t+1)\mathbb{E}\left[h_{t+2}(s_{t+2})\,|s_{t+1}\right] + X_2(t+1)s_{t+1}, \\
\overset{(c)}{=}& X_1(t+1)\left(\phi(s_{t+1})^\mathsf{T} \otimes Z\right)\widehat{\theta}_{t+2} + X_2(t+1)s_{t+1},
\end{aligned} \tag{49}
$$

$$
\begin{aligned}
\widehat{q}_{t+1}(s_{t+1}) =& \mathbb{E}\left[q_{t+2}(s_{t+2})\,|s_{t+1}\right] + s_{t+1}^\mathsf{T} Y_1(t+1)s_{t+1} + \mathbb{E}\left[h_{t+2}^\mathsf{T}(s_{t+2})|s_{t+1}\right] Y_2(t+1)\mathbb{E}\left[h_{t+2}(s_{t+2})|s_{t+1}\right] \\
& + 2s_{t+1}^\mathsf{T} Y_3(t+1)\mathbb{E}\left[h_{t+2}(s_{t+2})|s_{t+1}\right], \\
\overset{(d)}{=}& \phi(s_{t+1})^\mathsf{T}\left(I_d \otimes \overline{Z}\right)\widehat{\theta}_{t+2} + s_{t+1}^\mathsf{T} Y_1(t+1)s_{t+1} + \widehat{\theta}_{t+2}^\mathsf{T}\left(\phi(s_{t+1}) \otimes Z^\mathsf{T}\right) Y_2(t+1)\left(\phi(s_{t+1})^\mathsf{T} \otimes Z\right)\widehat{\theta}_{t+2} \\
& + 2s_{t+1}^\mathsf{T} Y_3(t+1)\left(\phi(s_{t+1})^\mathsf{T} \otimes Z\right)\widehat{\theta}_{t+2},
\end{aligned} \tag{50}
$$

where in steps (c) and (d) we have used eq. (44) and eq. (45), respectively. The temporal difference (TD) error is written as

$$
\begin{aligned}
\varepsilon_t(x, s, u) =& g_t(x, s, u) - \widehat{Q}_t(x, s, u) \\
=& c(x, s, u) + \min_v \widehat{Q}_{t+1}(x', s', v) - \widehat{Q}_t(x, s, u) \\
\overset{(d)}{=}& c(x, s, u) + x'^\mathsf{T} G_{t+1} x' + 2\widehat{h}_{t+1}^\mathsf{T}(s')x' + \widehat{q}_{t+1}(s') - c(x, s, u) - (Ax + Bu)^\mathsf{T} G_{t+1}(Ax + Bu) \\
& - \phi(s)^\mathsf{T} Y(x, u)\widehat{\theta}_{t+1} \\
=& 2\widehat{h}_{t+1}^\mathsf{T}(s')x' + \widehat{q}_{t+1}(s') - \phi(s)^\mathsf{T} Y(x, u)\widehat{\theta}_{t+1},
\end{aligned} \tag{51}
$$

where in step (d) we have used eq. (43) and eq. (48). The TD error $\varepsilon_t(x, s, u)$ in eq. (51) captures the discrepancy between the Bellman target and the current estimate of the $Q$-function. In Least-Squares Value Iteration (LSVI) in Algorithm 1, we minimize the squared TD error over the dataset, eq. (11), collected up to episode $\ell - 1$, to obtain an updated estimate of the $Q$-function. Specifically, the parameters $\widehat{\theta}_t$ at episode $\ell$ denoted by $\theta_t^\ell$ is obtained by solving the following regularized least-squares problem

$$
\begin{aligned}
\theta_{t+1}^\ell =& \arg\min_\theta \underbrace{\sum_{j=1}^{\ell-1} \varepsilon_t(x^j, s^j, u^j)^2 + \lambda\|\theta\|_2^2}_{J} \\
=& \arg\min_\theta \sum_{j=1}^{\ell-1}\left(\phi(s^j)^\mathsf{T} Y(x^j, u^j)\widehat{\theta}_{t+1} - 2\widehat{h}_{t+1}^\mathsf{T}(s'^j)x'^j - \widehat{q}_{t+1}(s'^j)\right)^2 + \lambda\|\theta\|_2^2.
\end{aligned} \tag{52}
$$

Taking the derivative of eq. (52) with respect to $\theta$, we get

$$
\frac{\partial J}{\partial \theta} = 2\sum_{j=1}^{\ell-1} Y^\mathsf{T}(x^j, u^j)\phi(s^j)\left(\phi(s^j)^\mathsf{T} Y(x^j, u^j)\theta - 2\widehat{h}_{t+1}^\mathsf{T}(s'^j)x' - \widehat{q}_{t+1}(s'^j)\right) + 2\lambda\theta.
$$

Setting the above derivative to zero and solving for $\theta$, we get

$$
\begin{aligned}
\theta_{t+1}^\ell &= \Lambda_t^{-1} \sum_{j=1}^{\ell-1} Y^\mathsf{T}(x^j, u^j) \phi(s^j) \big( 2\widehat{h}_{t+1}^\mathsf{T}(s'^j) x'^j + \widehat{q}_{t+1}(s'^j) \big), \\
\Lambda_t &= \sum_{j=1}^{\ell-1} Y(x^j, u^j)^\mathsf{T} \phi(s^j) \phi(s^j)^\mathsf{T} Y(x^j, u^j) + \lambda I_{d(n+1)}.
\end{aligned}
\tag{53}
$$

In Algorithm 1 we computed $\widehat{h}_{t+1}$ and $\widehat{q}_{t+1}$ at episode $\ell$ as in eq. (49) and eq. (50), respectively. These quantities are obtained using the updated parameter $\theta_{t+2}^\ell$ from the previous iteration of the backward-in-time weight update loop(lines 5-11 in Algorithm 1). In particular, we have $\widehat{h}_{t+1}(\cdot) = h_{t+1}^\ell(\cdot)$ and $\widehat{q}_{t+1}(\cdot) = q_{t+1}^\ell(\cdot)$.

## C   True weights $\overline{h}_t$ and $\overline{q}_t$

Throwout this Appendix, we use the following notation,

$$
\begin{aligned}
X_1(t) &= A^\mathsf{T} + K_{x,t}^\mathsf{T} B^\mathsf{T}, \quad X_2(t) = F + K_{x,t}^\mathsf{T} H^\mathsf{T}, \\
Y_1(t) &= M + H K_{s,t}, \quad Y_2(t) = B K_{h,t}, \quad Y_3(t) = H K_{h,t}, \\
\Phi_t &= \begin{bmatrix} \mathbb{E}_{\mu_{1,t}}\left[\phi(s_t)^\mathsf{T}\right] \\ \vdots \\ \mathbb{E}_{\mu_{d,t}}\left[\phi(s_t)^\mathsf{T}\right] \end{bmatrix}, \text{ and } \overline{m}_t = \begin{bmatrix} \overline{m}_{1,t} \\ \vdots \\ \overline{m}_{d,t} \end{bmatrix} = \begin{bmatrix} \mathbb{E}_{\mu_{1,t}}\left[s_t\right] \\ \vdots \\ \mathbb{E}_{\mu_{d,t}}\left[s_t\right] \end{bmatrix},
\end{aligned}
\tag{54}
$$

for $t \in \{0, \cdots, T\}$. In addition, we define for $t \in \{0, \cdots, T\}$

$$
\|X_1(t)\| \le \overline{X}_1, \quad \|X_2(t)\| \le \overline{X}_2, \quad \|Y_1(t)\| \le \overline{Y}_1, \quad \|Y_2(t)\| \le \overline{Y}_2, \quad \|Y_3(t)\| \le \overline{Y}_3.
\tag{55}
$$

### C.1   Closed-form expressions of $\overline{h}_t$ and $\overline{q}_t$

In this Appendix, we derive closed-form expressions for the true parameters $\overline{h}_t = \mathbb{E}_{\mu_t}[h_t(s_t)]$ and $\overline{q}_t = \mathbb{E}_{\mu_t}[q_t(s_t)]$.

**Theorem C.1.** *(closed-form expressions for the true $\overline{h}_t$ and $\overline{q}_t$) Consider the dynamics in eq. (1) and the Markov Process in eq. (2). Let Assumption 2.1 and Assumption 2.2 be satisfied. Let $\overline{h}_t = \mathbb{E}_{\mu_t}[h_t(s_t)]$ and $\overline{q}_t = \mathbb{E}_{\mu_t}[q_t(s_t)]$ for $t \in \{0, \cdots, T\}$, where $h_t(\cdot)$ and $q_t(\cdot)$ are as in eq. (36) and eq. (37), respectively. Then, for $t \in \{0, \cdots, T-1\}$*

$$
\begin{aligned}
\overline{h}_t &= (\Phi_t \otimes X_1(t))\overline{h}_{t+1} + (I_d \otimes X_2(t))\overline{m}_t, \\
\overline{q}_t &= \Phi_t \overline{q}_{t+1} + \mathbb{E}_{\mu_t}\left[s_t^\mathsf{T} Y_1(t) s_t\right] + \begin{bmatrix} \overline{h}_{t+1}^\mathsf{T}\big(\mathbb{E}_{\mu_{1,t}}\left[\phi(s_t)\phi(s_t)^\mathsf{T}\right] \otimes Y_2(t)\big)\overline{h}_{t+1} \\ \vdots \\ \overline{h}_{t+1}^\mathsf{T}\big(\mathbb{E}_{\mu_{d,t}}\left[\phi(s_t)\phi(s_t)^\mathsf{T}\right] \otimes Y_2(t)\big)\overline{h}_{t+1} \end{bmatrix} + 2\mathbb{E}_{\mu_t}\left[\phi(s_t)^\mathsf{T} \otimes s_t^\mathsf{T} Y_3(t)\right]\overline{h}_{t+1},
\end{aligned}
$$

*with $\overline{h}_T = F\mathbb{E}_{\mu_T}[s_T]$ and $\overline{q}_T = \mathbb{E}_{\mu_T}\left[s_T^\mathsf{T} M s_T\right]$, where $X_1(t), X_2(t), Y_1(t), Y_2(t), Y_3(t), \Phi_t$, and $\overline{m}_t$ are as in eq. (54).*

*Proof.* We re-write equation eq. (36) as

$$
h_t(s_t) = X_1(t)\mathbb{E}\left[h_{t+1}(s_{t+1}) \mid s_t\right] + X_2(t)s_t.
\tag{56}
$$

Taking the expectation of both sides with respect to $\mu_{i,t}$ for each $i \in \{1, \cdots, d\}$, we get

$$
\begin{aligned}
\mathbb{E}_{\mu_{i,t}} \left[ h_t \left( s_t \right) \right] &= X_1(t) \mathbb{E}_{\mu_{i,t}} \left[ \mathbb{E} \left[ h_{t+1} \left( s_{t+1} \right) | s_t \right] \right] + X_2(t) \mathbb{E}_{\mu_{i,t}} \left[ s_t \right] \\
&= X_1(t) \mathbb{E}_{\mu_{i,t}} \left[ \sum_{j=1}^{d} \phi_j(s_t) \mathbb{E}_{\mu_{j,t+1}} \left[ h_{t+1}(s_{t+1}) \right] \right] + X_2(t) \overline{m}_{i,t} \\
&= X_1(t) \sum_{j=1}^{d} \mathbb{E}_{\mu_{i,t}} \left[ \phi_j(s_t) \right] \mathbb{E}_{\mu_{j,t+1}} \left[ h_{t+1}(s_{t+1}) \right] + X_2(t) \overline{m}_{i,t} \\
&= X_1(t) (\mathbb{E}_{\mu_{i,t}} \left[ \phi(s_t)^{\mathsf{T}} \right] \otimes I_n) \overline{h}_{t+1} + X_2(t) \overline{m}_{i,t}.
\end{aligned}
\tag{57}
$$

By noting that

$$
\overline{h}_t = \begin{bmatrix} \mathbb{E}_{\mu_{1,t}} \left[ h_t(s_t) \right] \\ \vdots \\ \mathbb{E}_{\mu_{d,t}} \left[ h_t(s_t) \right] \end{bmatrix},
$$

and denoting

$$
\Phi_t = \begin{bmatrix} \mathbb{E}_{\mu_{1,t}} \left[ \phi(s_t)^{\mathsf{T}} \right] \\ \vdots \\ \mathbb{E}_{\mu_{d,t}} \left[ \phi(s_t)^{\mathsf{T}} \right] \end{bmatrix}, \quad \text{and} \quad \overline{m}_t = \begin{bmatrix} \overline{m}_{1,t} \\ \vdots \\ \overline{m}_{d,t} \end{bmatrix},
$$

we can write

$$
\begin{aligned}
\overline{h}_t &= (I_d \otimes X_1(t))(\Phi_t \otimes I_n) \overline{h}_{t+1} + (I_d \otimes X_2(t)) \overline{m}_t \\
&= (\Phi_t \otimes X_1(t)) \overline{h}_{t+1} + (I_d \otimes X_2(t)) \overline{m}_t.
\end{aligned}
\tag{58}
$$

Next, we re-write equation eq. (37) as

$$
\begin{aligned}
q_t \left( s_t \right) &= \mathbb{E} \left[ q_{t+1} \left( s_{t+1} \right) | s_t \right] + s_t^{\mathsf{T}} Y_1(t) s_t + \mathbb{E} \left[ h_{t+1}^{\mathsf{T}} \left( s_{t+1} \right) | s_t \right] Y_2(t) \mathbb{E} \left[ h_{t+1} \left( s_{t+1} \right) | s_t \right] \\
&\quad + 2 s_t^{\mathsf{T}} Y_3(t) \mathbb{E} \left[ h_{t+1} \left( s_{t+1} \right) | s_t \right].
\end{aligned}
\tag{59}
$$

Taking the expectation of both sides with respect to $\mu_{i,t}$ for each $i \in \{1, \cdots, d\}$, we get

$$
\begin{aligned}
\mathbb{E}_{\mu_{i,t}} \left[ q_t(s_t) \right] &= \mathbb{E}_{\mu_{i,t}} \left[ \mathbb{E} \left[ q_{t+1}(s_{t+1}) | s_t \right] \right] + \mathbb{E}_{\mu_{i,t}} \left[ s_t^{\mathsf{T}} Y_1(t) s_t \right] + \mathbb{E}_{\mu_{i,t}} \left[ \mathbb{E} \left[ h_{t+1}^{\mathsf{T}}(s_{t+1}) | s_t \right] Y_2(t) \mathbb{E} \left[ h_{t+1}(s_{t+1}) | s_t \right] \right] \\
&\quad + 2 \mathbb{E}_{\mu_{i,t}} \left[ s_t^{\mathsf{T}} Y_3(t) \mathbb{E} \left[ h_{t+1}(s_{t+1}) | s_t \right] \right].
\end{aligned}
\tag{60}
$$

We start with,

$$
\begin{aligned}
\mathbb{E}_{\mu_{i,t}} \left[ \mathbb{E} \left[ q_{t+1} \left( s_{t+1} \right) | s_t \right] \right] &= \mathbb{E}_{\mu_{i,t}} \left[ \sum_{j=1}^{d} \phi_j(s_t) \mathbb{E}_{\mu_{j,t+1}} \left[ q_{t+1}(s_{t+1}) \right] \right] \\
&= \mathbb{E}_{\mu_{i,t}} \left[ \sum_{j=1}^{d} \phi_j(s_t) \right] \mathbb{E}_{\mu_{j,t+1}} \left[ q_{t+1}(s_{t+1}) \right] \\
&= \underbrace{\begin{bmatrix} \mathbb{E}_{\mu_{i,t}}[\phi_1(s_t)] & \cdots & \mathbb{E}_{\mu_{i,t}}[\phi_d(s_t)] \end{bmatrix}}_{\mathbb{E}_{\mu_{i,t}}[\phi(s_t)^{\mathsf{T}}]} \underbrace{\begin{bmatrix} \mathbb{E}_{\mu_{1,t+1}}[q_{t+1}(s_{t+1})] \\ \vdots \\ \mathbb{E}_{\mu_{d,t+1}}[q_{t+1}(s_{t+1})] \end{bmatrix}}_{\overline{q}_{t+1}}.
\end{aligned}
\tag{61}
$$

Next we have,

$$\mathbb{E}_{\mu_{i,t}}\left[\mathbb{E}\left[h_{t+1}^{\mathsf{T}}\left(s_{t+1}\right)|s_t\right]Y_2(t)\mathbb{E}\left[h_{t+1}\left(s_{t+1}\right)|s_t\right]\right]$$

$$=\mathbb{E}_{\mu_{i,t}}\left[\sum_{j=1}^{d}\phi_j(s_t)\mathbb{E}_{\mu_{j,t+1}}\left[h_{t+1}(s_{t+1})^{\mathsf{T}}\right]Y_2(t)\sum_{k=1}^{d}\phi_k(s_t)\mathbb{E}_{\mu_{k,t+1}}\left[h_{t+1}(s_{t+1})\right]\right]$$

$$=\mathbb{E}_{\mu_{i,t}}\left[\begin{bmatrix}\mathbb{E}_{\mu_{1,t+1}}\left[h_{t+1}(s_{t+1})\right]\\\vdots\\\mathbb{E}_{\mu_{d,t+1}}\left[h_{t+1}(s_{t+1})\right]\end{bmatrix}^{\mathsf{T}}\underbrace{\begin{bmatrix}\phi_1(s_t)I_n\\\vdots\\\phi_d(s_t)I_n\end{bmatrix}}_{\phi(s_t)\otimes I_n}Y_2(t)\begin{bmatrix}\phi_1(s_t)I_n\\\vdots\\\phi_d(s_t)I_n\end{bmatrix}^{\mathsf{T}}\underbrace{\begin{bmatrix}\mathbb{E}_{\mu_{1,t+1}}\left[h_{t+1}(s_{t+1})\right]\\\vdots\\\mathbb{E}_{\mu_{d,t+1}}\left[h_{t+1}(s_{t+1})\right]\end{bmatrix}}_{\overline{h}_{t+1}}\right] \quad (62)$$

$$=\overline{h}_{t+1}^{\mathsf{T}}\mathbb{E}_{\mu_{i,t}}\left[\left(\phi(s_t)\otimes I_n\right)Y_2(t)\left(\phi(s_t)^{\mathsf{T}}\otimes I_n\right)\right]\overline{h}_{t+1}$$

$$=\overline{h}_{t+1}^{\mathsf{T}}\mathbb{E}_{\mu_i}\left[\left(\phi(s_t)\otimes I_n\right)(1\otimes Y_2(t))(\phi(s_t)^{\mathsf{T}}\otimes I_n)\right]\overline{h}_{t+1}$$

$$=\overline{h}_{t+1}^{\mathsf{T}}\left(\mathbb{E}_{\mu_i}\left[\phi(s_t)\phi(s_t)^{\mathsf{T}}\right]\otimes Y_2(t)\right)\overline{h}_{t+1}.$$

Finally, we have

$$\mathbb{E}_{\mu_{i,t}}\left[s_t^{\mathsf{T}}Y_3(t)\mathbb{E}\left[h_{t+1}\left(s_{t+1}\right)|s_t\right]\right]=\mathbb{E}_{\mu_{i,t}}\left[s_t^{\mathsf{T}}Y_3(t)\sum_{j=1}^{d}\phi_j(s_t)\mathbb{E}_{\mu_{j,t+1}}\left[h_{t+1}(s_{t+1})\right]\right]$$

$$=\mathbb{E}_{\mu_{i,t}}\left[s_t^{\mathsf{T}}Y_3(t)\left(\phi(s_t)^{\mathsf{T}}\otimes I_n\right)\right]\overline{h}_{t+1} \quad (63)$$

$$=\mathbb{E}_{\mu_{i,t}}\left[\left(1\otimes s_t^{\mathsf{T}}Y_3(t)\right)\left(\phi(s_t)^{\mathsf{T}}\otimes I_n\right)\right]\overline{h}_{t+1}$$

$$=\mathbb{E}_{\mu_{i,t}}\left[\left(\phi(s_t)^{\mathsf{T}}\otimes s_t^{\mathsf{T}}Y_3(t)\right)\right]\overline{h}_{t+1}.$$

Substituting eq. (61), eq. (62), and eq. (63) in eq. (60), we get

$$\mathbb{E}_{\mu_{i,t}}\left[q_t(s_t)\right]=\mathbb{E}_{\mu_{i,t}}\left[\phi(s_t)^{\mathsf{T}}\right]\overline{q}_{t+1}+\mathbb{E}_{\mu_{i,t}}\left[s_t^{\mathsf{T}}Y_1(t)s_t\right]$$
$$+\overline{h}_{t+1}^{\mathsf{T}}\left(\mathbb{E}_{\mu_{i,t}}\left[\phi(s_t)\phi(s_t)^{\mathsf{T}}\right]\otimes Y_2(t)\right)\overline{h}_{t+1} \quad (64)$$
$$+2\mathbb{E}_{\mu_{i,t}}\left[\phi(s_t)^{\mathsf{T}}\otimes s_t^{\mathsf{T}}Y_3(t)\right]\overline{h}_{t+1}.$$

Then, we can write

$$\underbrace{\begin{bmatrix}\mathbb{E}_{\mu_{1,t}}\left[q_t(s_t)\right]\\\vdots\\\mathbb{E}_{\mu_{d,t}}\left[q_t(s_t)\right]\end{bmatrix}}_{\overline{q}_t}=\underbrace{\begin{bmatrix}\mathbb{E}_{\mu_{1,t}}\left[\phi(s_t)^{\mathsf{T}}\right]\\\vdots\\\mathbb{E}_{\mu_{1,t}}\left[\phi(s_t)^{\mathsf{T}}\right]\end{bmatrix}}_{\Phi_t}\underbrace{\begin{bmatrix}\mathbb{E}_{\mu_{1,t+1}}[q_{t+1}(s_{t+1})]\\\vdots\\\mathbb{E}_{\mu_{d,t+1}}[q_{t+1}(s_{t+1})]\end{bmatrix}}_{\overline{q}_{t+1}}+\begin{bmatrix}\mathbb{E}_{\mu_{1,t}}\left[s_t^{\mathsf{T}}Y_1(t)s_t\right]\\\vdots\\\mathbb{E}_{\mu_{d,t}}\left[s_t^{\mathsf{T}}Y_1(t)s_t\right]\end{bmatrix}$$

$$+\begin{bmatrix}\overline{h}_{t+1}^{\mathsf{T}}\left(\mathbb{E}_{\mu_{1,t}}\left[\phi(s_t)\phi(s_t)^{\mathsf{T}}\right]\otimes Y_2(t)\right)\overline{h}_{t+1}\\\vdots\\\overline{h}_{t+1}^{\mathsf{T}}\left(\mathbb{E}_{\mu_{d,t}}\left[\phi(s_t)\phi(s_t)^{\mathsf{T}}\right]\otimes Y_2(t)\right)\overline{h}_{t+1}\end{bmatrix}+2\begin{bmatrix}\mathbb{E}_{\mu_{1,t}}\left[\phi(s_t)^{\mathsf{T}}\otimes s_t^{\mathsf{T}}Y_3(t)\right]\\\vdots\\\mathbb{E}_{\mu_{d,t}}\left[\phi(s_t)^{\mathsf{T}}\otimes s_t^{\mathsf{T}}Y_3(t)\right]\end{bmatrix}\overline{h}_{t+1}. \quad (65)$$

Finally, from Theorem A.1, we have $\overline{h}_T=\mathbb{E}_{\mu_T}\left[h_T(s_T)\right]=F\mathbb{E}_{\mu_T}\left[s_T\right]$, and $\overline{q}_t=\mathbb{E}_{\mu_T}\left[q_T(s_T)\right]=\mathbb{E}_{\mu_T}\left[s_T^{\mathsf{T}}Ms_T\right]$.
□

## C.2 Upper bounds on $\|\overline{h}_t\|$ and $\|\overline{q}_t\|$

**Theorem C.2.** *(upper bounds on the true $\overline{h}_t$ and $\overline{q}_t$)* *Consider the dynamics in eq. (1) and the Markov Process in eq. (2). Let Assumption 2.1 and Assumption 2.2 be satisfied. Let $\overline{h}_t$ and $\overline{q}_t$ be as in Theorem C.1 for $t\in\{0,\cdots,T\}$. Then,*

$$\|\overline{h}_t\|\leq\|F\|\,\delta_s\alpha\rho^{T-t}+\frac{\overline{X}_2\delta_s\alpha\sqrt{d}}{1-\rho},$$

$$\|\overline{q}_t\|\leq\delta_s^2\|M\|+\delta_s^2\overline{Y}_1\sqrt{d}+\frac{\overline{Y}_2\|\overline{h}_{t+1}\|^2}{\sqrt{d}}+\delta_s\overline{Y}_3\|\overline{h}_{t+1}\|,$$

*for $t \in \{0, \cdots, T\}$ with $\|\overline{h}_{T+1}\| = 0$, where $\alpha > 0$, $0 < \rho < 1$ are constants, and $\overline{X}_1$, $\overline{X}_2$, $\overline{Y}_1$, $\overline{Y}_2$, and $\overline{Y}_3$ are as in eq. (55).*

*Proof.* From eq. (58), we have

$$\overline{h}_t = (\Phi_t \otimes X_1(t))\overline{h}_{t+1} + (I_d \otimes X_2(t))\overline{m}_t. \tag{66}$$

Let $\Xi(t_1, t_2) = \prod_{i=t_2-1}^{t_1} \Phi_i \otimes X_1(i)$ with $t_2 > t_1$. Then, given $\overline{h}_T$, we can write

$$\overline{h}_t = \Xi(t, T)\overline{h}_T + \sum_{j=T-1}^{t} \Xi(t, j) \left( I_d \otimes X_2(j) \right) \overline{m}_j. \tag{67}$$

Then, we can write

$$\|\overline{h}_t\| \leq \|\Xi(t, T)\|\|\overline{h}_T\| + \sum_{j=T-1}^{t} \|\Xi(t, j)\|\| \left( I_d \otimes X_2(j) \right) \|\|\overline{m}_j\|. \tag{68}$$

Now bound each term separately. For $t_2 > t_1$, we have

$$\Xi(t_1, t_2) = \prod_{i=t_2-1}^{t_1} \Phi_i \otimes X_1(i) = \left( \prod_{i=t_2-1}^{t_1} \Phi_i \right) \otimes \left( \prod_{i=t_2-1}^{t_1} X_1(i) \right). \tag{69}$$

Notice that, from eq. (36) and eq. (26), we have $X_1(i) = A^{\mathsf{T}} + K_{i,x}^{\mathsf{T}} B^{\mathsf{T}} = A_c(i)^{\mathsf{T}}$. Then, we can write

$$\prod_{i=t_2-1}^{t_1} X_1(i) = \prod_{i=t_2-1}^{t_1} A_c(i)^{\mathsf{T}} = \left( \prod_{i=t_1}^{t_2-1} A_c(i) \right)^{\mathsf{T}}. \tag{70}$$

Then, noting that $\|\cdot^{\mathsf{T}}\| = \|\cdot\|$, we can upper bound eq. (69) as

$$\|\Xi(t_1, t_2)\| \leq \left( \prod_{i=t_2-1}^{t_1} \|\Phi_i\| \right) \left( \left\| \prod_{i=t_1}^{t_2-1} A_c(i) \right\| \right). \tag{71}$$

From (Celi et al., 2022, Lemma B.1), we have $\left\| \prod_{i=t_1}^{t_2-1} A_c(i) \right\| \leq \alpha\rho^{t_2-t_1}$, where $\alpha > 0$ and $\rho \in (0, 1)$ are constants. Further, using Assumption 2.1, we have for any $i \leq 0$

$$
\begin{aligned}
\|\Phi_i\|_2 \leq \|\Phi_i\|_{\mathrm{F}} &= \sqrt{\mathrm{tr}\left( \left( \mathbb{E}_{\mu_i} [\phi^{\mathsf{T}}(s_i)] \right) \left( \mathbb{E}_{\mu_i} [\phi^{\mathsf{T}}(s_i)] \right)^{\mathsf{T}} \right)} \\
&= \sqrt{\sum_{j=1}^{d} \left( \mathbb{E}_{\mu_{j,i}} [\phi^{\mathsf{T}}(s_i)] \right) \left( \mathbb{E}_{\mu_{j,i}} [\phi^{\mathsf{T}}(s_i)] \right)^{\mathsf{T}}} \\
&= \sqrt{\sum_{j=1}^{d} \| \mathbb{E}_{\mu_{j,i}} [\phi^{\mathsf{T}}(s_i)] \|_2^2} \\
&\leq \sqrt{\sum_{j=1}^{d} \frac{1}{d}} = 1.
\end{aligned}
\tag{72}
$$

Hence, we can re-write eq. (71) as

$$\|\Xi(t_1, t_2)\| \leq \alpha\rho^{t_2-t_1}. \tag{73}$$

From Theorem A.1, we have $\overline{h}_T = \mathbb{E}_{\mu_T}[h_T(s_T)] = F\mathbb{E}_{\mu_T}[s_T]$. Then, we have

$$\left\|\overline{h}_T\right\| \leq \|F\|\,\|\mathbb{E}_{\mu_T}[s_T]\| \overset{(a)}{\leq} \|F\|\,\mathbb{E}_{\mu_T}[\|s_T\|] \overset{(b)}{\leq} \|F\|\,\delta_s, \tag{74}$$

where in step (a) we have used the Jensen's inequality, and in step (b) we have used Assumption 2.1. Next, we have

$$\|\overline{m}_t\| = \|\mathbb{E}_{\mu_t}[s_t]\| = \sqrt{\sum_{i=1}^{d}\left(\mathbb{E}_{\mu_{i,t}}[s_t]\right)^{\mathsf{T}}\left(\mathbb{E}_{\mu_{i,t}}[s_t]\right)} = \sqrt{\sum_{i=1}^{d}\|\mathbb{E}_{\mu_{i,t}}[s_t]\|^2} \leq \sqrt{d\delta_s^2} = \delta_s\sqrt{d}. \tag{75}$$

Let $\|X_2(t)\| \leq \overline{X}_2$ for $t \in \{0, \cdots, T-1\}$. Then, using eq. (73), eq. (74), and eq. (75) we can write eq. (68) as

$$\begin{aligned}
\|\overline{h}_t\| &\leq \|F\|\,\delta_s\alpha\rho^{T-t} + \overline{X}_2\delta_s\alpha\sqrt{d}\sum_{j=T-1}^{t}\rho^{j-t} \\
&\overset{(c)}{=} \|F\|\,\delta_s\alpha\rho^{T-t} + \overline{X}_2\delta_s\alpha\sqrt{d}\sum_{k=0}^{T-t-1}\rho^{k} \\
&= \|F\|\,\delta_s\alpha\rho^{T-t} + \overline{X}_2\delta_s\alpha\sqrt{d}\left(\frac{1-\rho^{T-t}}{1-\rho}\right) \\
&\leq \|F\|\,\delta_s\alpha\rho^{T-t} + \frac{\overline{X}_2\delta_s\alpha\sqrt{d}}{1-\rho}.
\end{aligned} \tag{76}$$

Now we bound $\overline{q}_t$ for $t \in \{0, \cdots, T\}$. From eq. (65), we have

$$\overline{q}_t = \Phi_t\overline{q}_{t+1} + v_t, \tag{77}$$

where,

$$v_t = \mathbb{E}_{\mu_t}\left[s_t^{\mathsf{T}}Y_1(t)s_t\right] + \begin{bmatrix} \overline{h}_{t+1}^{\mathsf{T}}\left(\mathbb{E}_{\mu_{1,t}}\left[\phi(s_t)\phi(s_t)^{\mathsf{T}}\right] \otimes Y_2(t)\right)\overline{h}_{t+1} \\ \vdots \\ \overline{h}_{t+1}^{\mathsf{T}}\left(\mathbb{E}_{\mu_{d,t}}\left[\phi(s_t)\phi(s_t)^{\mathsf{T}}\right] \otimes Y_2(t)\right)\overline{h}_{t+1} \end{bmatrix} + 2\mathbb{E}_{\mu_t}\left[\phi(s_t)^{\mathsf{T}} \otimes s_t^{\mathsf{T}}Y_3(t)\right]\overline{h}_{t+1}. \tag{78}$$

Given $\overline{q}_T$, we can write for $t \in \{0, \cdots, T-1\}$

$$\overline{q}_t = \prod_{i=T-1}^{t}\Phi_i\overline{q}_T + \sum_{i=T-1}^{t}\prod_{j=i-1}^{t}\Phi_j v_i. \tag{79}$$

Then, we can upper bound $\|\overline{q}_t\|$ as

$$\begin{aligned}
\|\overline{q}_t\| &\leq \prod_{i=T-1}^{t}\|\Phi_i\|\|\overline{q}_T\| + \sum_{i=T-1}^{t}\prod_{j=i-1}^{t}\|\Phi_j\|\|v_i\| \\
&\overset{(d)}{\leq} \|\overline{q}_T\| + \sum_{i=T-1}^{t}\|v_i\|,
\end{aligned} \tag{80}$$

where in step (d) we have used eq. (72). Now we bound each term of $v_t$ for $t \in \{0, \cdots, T-1\}$. We start with

$$\begin{aligned}
\|\mathbb{E}_{\mu_{i,t}}\left[s^{\mathsf{T}}(t)Y_1(t)s_t\right]\| &\leq \mathbb{E}_{\mu_{i,t}}\left[\|s^{\mathsf{T}}(t)Y_1(t)s_t\|\right] \\
&\leq \mathbb{E}_{\mu_{i,t}}\left[\|s_t\|^2\|Y_1(t)\|s_t\|\right] \\
&\leq \delta_s^2\|Y_1(t)\|,
\end{aligned} \tag{81}$$

for $i \in \{1, \cdots, d\}$. Then, we have

$$\|\mathbb{E}_{\mu_t}\left[s^\mathsf{T}(t)Y_1(t)s_t\right]\| = \sqrt{\sum_{i=1}^{d}\left\|\mathbb{E}_{\mu_{i,t}}\left[s^\mathsf{T}(t)Y_1(t)s_t\right]\right\|^2} \leq \delta_s^2\|Y_1(t)\|\sqrt{d}. \tag{82}$$

Next, for $i \in \{1, \cdots, d\}$, we have

$$\|\overline{h}_{t+1}^\mathsf{T}\left(\mathbb{E}_{\mu_{i,t}}\left[\phi(s_t)\phi^\mathsf{T}(s_t)\right]\otimes Y_2(t)\right)\overline{h}_{t+1}\| \leq \|\overline{h}_{t+1}\|^2\|\phi(s_t)\|^2\|Y_2(t)\| \leq \frac{\|\overline{h}_{t+1}\|^2\|Y_2(t)\|}{d}. \tag{83}$$

Then,

$$\left\|\begin{bmatrix}\overline{h}_{t+1}^\mathsf{T}\left(\mathbb{E}_{\mu_{1,t}}\left[\phi(s_t)\phi(s_t)^\mathsf{T}\right]\otimes Y_2(t)\right)\overline{h}_{t+1}\\ \vdots \\ \overline{h}_{t+1}^\mathsf{T}\left(\mathbb{E}_{\mu_{d,t}}\left[\phi(s_t)\phi(s_t)^\mathsf{T}\right]\otimes Y_2(t)\right)\overline{h}_{t+1}\end{bmatrix}\right\| = \sqrt{\sum_{i=1}^{d}\left\|\overline{h}_{t+1}^\mathsf{T}\left(\mathbb{E}_{\mu_{i,t}}\left[\phi(s_t)\phi^\mathsf{T}(s_t)\right]\otimes Y_2(t)\right)\overline{h}_{t+1}\right\|^2}$$
$$\leq \frac{\|\overline{h}_{t+1}\|^2\|Y_2(t)\|}{\sqrt{d}}. \tag{84}$$

Next, for $i \in \{1, \cdots, d\}$, we have

$$\left\|\mathbb{E}_{\mu_{i,t}}\left[\phi(s_t)^\mathsf{T}\otimes s_t^\mathsf{T}Y_3(t)\right]\right\| \overset{(e)}{\leq} \mathbb{E}_{\mu_{i,t}}\left[\left\|\phi(s_t)^\mathsf{T}\otimes s_t^\mathsf{T}Y_3(t)\right\|\right]$$
$$\leq \mathbb{E}_{\mu_{i,t}}\left[\|\phi(s_t)\|\,\|s_t\|\|Y_3(t)\|\right]$$
$$\leq \frac{\delta_s\|Y_3(t)\|}{\sqrt{d}}, \tag{85}$$

where in step (e) we have used Jensen's inequality. Then,

$$\left\|\mathbb{E}_{\mu_t}\left[\phi(s_t)^\mathsf{T}\otimes s_t^\mathsf{T}Y_3(t)\right]\right\| = \sqrt{\sum_{i=1}^{d}\left\|\mathbb{E}_{\mu_{i,t}}\left[\phi(s_t)^\mathsf{T}\otimes s_t^\mathsf{T}Y_3(t)\right]\right\|^2} \leq \delta_s\|Y_3(t)\|. \tag{86}$$

Let $\|Y_1(t)\| \leq \overline{Y}_1$, $\|Y_2(t)\| \leq \overline{Y}_2$, and $\|Y_3(t)\| \leq \overline{Y}_3$ for $t \in \{0, \cdots, T\}$. Using eq. (82), eq. (84), and eq. (86)

$$\|v_t\| \leq \delta_s^2\overline{Y}_1\sqrt{d} + \frac{\overline{Y}_2\|\overline{h}_{t+1}\|^2}{\sqrt{d}} + \delta_s\overline{Y}_3\|\overline{h}_{t+1}\|, \tag{87}$$

for $t \in \{0, \cdots, T\}$. From Theorem A.1, we have $\overline{q}_T = \mathbb{E}_\mu\left[q_T(s_T)\right] = \mathbb{E}_\mu\left[s_T^\mathsf{T}Ms_T\right]$. Then, $\|\overline{q}_T\| \leq \delta_s^2\|M\|$. Then, we can re-write eq. (80) as

$$\|\overline{q}_t\| \leq \delta_s^2\|M\| + \delta_s^2\overline{Y}_1\sqrt{d} + \frac{\overline{Y}_2\|\overline{h}_{t+1}\|^2}{\sqrt{d}} + \delta_s\overline{Y}_3\|\overline{h}_{t+1}\|. \tag{88}$$

$$\square$$

Following the same notation as eq. (9), let the true parameter be denoted by $\theta^*$, which is written as

$$\theta_t^* = \begin{bmatrix}\theta_{1,t}^*{}^\mathsf{T} & \cdots & \theta_{d,t}^*{}^\mathsf{T}\end{bmatrix}^\mathsf{T}, \quad \text{where} \quad \theta_{i,t}^* = \begin{bmatrix}\overline{h}_{i,t}\\ \overline{q}_{i,t}\end{bmatrix}. \tag{89}$$

where $\overline{h}_{i,t} \in \mathbb{R}^n$ and $\overline{q}_{i,t} \in \mathbb{R}$ are the components of $\overline{h}_t$ and $\overline{q}_t$ in Theorem C.1 for $i \in \{1, \cdots, d\}$ and $t \in \{0, \cdots, T\}$.

**Corollary C.3.** *(bound on $\theta_t^*$)* *Let $\theta_t^*$ be as in eq. (89). Then, under the same assumptions of Theorem C.1 and Theorem C.2, we have $\|\theta_t^*\| \leq c_\theta\sqrt{d}$ for $t \in \{0, \cdots, T\}$, where $c_\theta > 0$ is independent of $d$.*

*Proof.* Theorem C.2 implies that for $t \in \{0, \cdots, T\}$,

$$\|\overline{h}_t\| \leq a_h + b_h\sqrt{d}, \quad \|\|\overline{q}_t\| \leq a_q + b_q\sqrt{d}, \tag{90}$$

where $a_h > 0$, $b_h > 0$, $a_q > 0$, and $b_q > 0$ are independent of $d$. Then, we can bound $\theta_t^*$ in eq. (89) as

$$
\begin{aligned}
\|\theta_t^*\| &= \sqrt{\sum_{i=1}^{d}(\theta_{i,t}^*)^\mathsf{T}\theta_{i,t}^*} = \sqrt{\sum_{i=1}^{d}(\overline{h}_{i,t})^\mathsf{T}\overline{h}_{i,t} + (\overline{q}_{i,t})^\mathsf{T}\overline{q}_{i,t}} \\
&= \sqrt{(\overline{h}_t)^\mathsf{T}\overline{h}_t + (\overline{q}_t)^\mathsf{T}\overline{q}_t} = \sqrt{\|\overline{h}_t\|^2 + \|\overline{q}_t\|^2} \\
&\leq \|\overline{h}_t\| + \|\overline{q}_t\| \leq \underbrace{(a_h + b_h + a_q + b_q)}_{c_\theta}\sqrt{d}.
\end{aligned}
\tag{91}
$$

$\square$

Corollary C.3 implies that choosing the projection radius in Algorithm 1 as $R_\theta \geq c_\theta\sqrt{d}$ guarantees that $\theta_t^*$ belongs to the projection ball for all $t$.

## D    Proof of Theorem 3.3

Let $A_c(t) = A + BK_{x,t}$ and let $\varphi(t_2, t_1) = \prod_{i=t_1}^{t_2-1} A_c(i)$ denote the state transition matrix from $t_1$ to $t_2$.[2] Let $\pi_t^\ell(x_t, s_t) = K_{x,t}x_t^\ell + K_{s,t}s_t^\ell + K_{h,t}\left(\phi\left(s_t^\ell\right)^\mathsf{T} \otimes Z\right)\theta_{t+1}^\ell$ denote the policy learned from Algorithm 1 at episode $\ell$ and time $t$, where $Z = [I_n, 0_{n \times 1}]$. Then, the evolution of $x_t$ in system eq. (1) under the policy $\{\pi_1^\ell, \cdots, \pi_{t-1}^\ell\}$ for $t \in \{0, \cdots, T\}$ is written as

$$x_t^\ell = \varphi(t,0)x_0^\ell + \sum_{i=0}^{t-1}\varphi(t,i+1)B\overline{u}_i^\ell, \tag{92}$$

where $x_0^\ell$ is the initial state at episode $\ell$ and $\overline{u}_i^\ell = K_{i,s}s^\ell(i) + K_{i,h}\left(\phi(s_i^\ell) \otimes Z\right)\theta_{i+1}^\ell$. Then,

$$\|x_t^\ell\| \leq \|\varphi(t,0)\|\|x_0^\ell\| + \|B\|\sum_{i=0}^{t-1}\|\varphi(t,i+1)\|\left(\sup_{0 \leq j \leq t-1}\|\overline{u}_j^\ell\|\right). \tag{93}$$

From (Celi et al., 2022, Lemma B.1), we have $\|\varphi(t_2, t_1)\| \leq \alpha\rho^{t_2-t_1}$ where $\alpha > 0$ and $\rho \in (0,1)$ are constants. Then, we can write eq. (104) as

$$
\begin{aligned}
\|x_t^\ell\| &\leq \alpha\rho^t\|x_0^\ell\| + \alpha\|B\|\sum_{i=0}^{t-1}\rho^{t-i-1}\underbrace{\left(\sup_{0 \leq j \leq t-1}\|\overline{u}_j^\ell\|\right)}_{u_\infty^\ell} \\
&\overset{(a)}{=} \alpha\rho^t\|x_0^\ell\| + \alpha\|B\|\sum_{k=0}^{t-1}\rho^k u_\infty^\ell \\
&= \alpha\rho^t\|x_0^\ell\| + \alpha\|B\|\left(\frac{1-\rho^{t-1}}{1-\rho}\right)u_\infty^\ell \\
&\leq \alpha\rho^t\|x_0^\ell\| + \alpha\|B\|\left(\frac{1}{1-\rho}\right)u_\infty^\ell,
\end{aligned}
\tag{94}
$$

---

[2]The matrix multiplication is performed from the left, i.e., $A(t_1)$ appears as the rightmost matrix in the product.

where in step (a), we have changed the index in the sum to $k = t - i - 1$. Next, we bound on $u_\infty^\ell$. Let $\|\theta_t^\ell\| \leq R_\theta$, $\|K_{s,t}\| \leq \overline{K}_s$, and $\|K_{h,t}\| \leq \overline{K}_h$ for $t \in \{0, \cdots, T-1\}$ and episode $\ell$. Then we have,

$$
\begin{aligned}
\|\overline{u}_t^\ell\| &\leq \|K_{s,t}\|\|s_t^\ell\| + \|K_{h,t}\| \left(\phi(s_t^\ell) \otimes Z\right)\|\|\theta_{t+1}^\ell\| \\
&\leq \overline{K}_s \delta_s + \overline{K}_h \|\phi(s_t^\ell)\|\|Z\| R_\theta \\
&\leq \overline{K}_s \delta_s + \frac{\overline{K}_h R_\theta}{\sqrt{d}}.
\end{aligned}
\tag{95}
$$

Since the above bound is uniform for $t \in \{0, \cdots, T-1\}$, we have $u_\infty^\ell \leq \overline{K}_s \delta_s + \frac{\overline{K}_h R_\theta}{\sqrt{d}}$. The proof follows by substituting the bound of $u_\infty^\ell$ in eq. (94).

## E Proof of Theorem 3.4

We begin by presenting the following technical Lemmas.

**Lemma E.1.** *Let $X_t = \sum_{i=1}^{t-1} z_i z_i^\mathsf{T} + \gamma I_p$, where $z_i \in \mathbb{R}^p$ and $\gamma > 0$. Let $\mathbb{E}\left[zz^\mathsf{T}\right] \succeq \alpha I_p$ with $\alpha > 0$, and $\|z\| \leq \zeta$. Let $\delta \in [0,1]$ and assume $t \geq (8\zeta^2 \log(p/\delta))/\alpha$. Then, with probability at least $1 - \delta$, the minimum eigenvalue of $X_t$ satisfies*

$$
\lambda_{min}(X_t) \geq \gamma + \frac{(t-1)\alpha}{2}.
$$

*Proof.* Let $\alpha = \lambda_{\min}(\mathbb{E}\left[zz^\mathsf{T}\right])$. Define

$$
\mu_{\min} \triangleq \lambda_{\min}\left(\sum_{i=1}^{t-1} \mathbb{E}\left[zz^\mathsf{T}\right]\right) = \lambda_{\min}\left((t-1)\mathbb{E}\left[zz^\mathsf{T}\right]\right) = (t-1)\lambda_{\min}\left(\mathbb{E}\left[zz^\mathsf{T}\right]\right) = (t-1)\alpha.
$$

Further, we have $z_i z_i^\mathsf{T} \succeq 0$ and $\lambda_{\max}\left(z_i z_i^\mathsf{T}\right) = \|z_i\|^2 \leq \zeta^2$. Then, using (Tropp, 2012, Theorem 1.1), we have

$$
\begin{aligned}
\mathbb{P}\left(\lambda_{\min}\left(\sum_{i=1}^{t-1} z_i z_i^\mathsf{T}\right) \leq (1-\varepsilon)(t-1)\alpha\right) &\leq p\left(\frac{\exp(-\varepsilon)}{(1-\varepsilon)^{1-\varepsilon}}\right)^{\frac{(t-1)\alpha}{\zeta^2}} \\
&\overset{(a)}{\leq} p\exp\left(\frac{-\varepsilon^2(t-1)\alpha}{2\zeta^2}\right),
\end{aligned}
\tag{96}
$$

for $\varepsilon \in [0,1]$, where in step (a) we have used $\frac{\exp(-\varepsilon)}{(1-\varepsilon)^{1-\varepsilon}} \leq \exp\left(-\varepsilon^2/2\right)$ for $\varepsilon \in (0,1)$. Choose $\varepsilon = 0.5$, then we write eq. (96) as

$$
\mathbb{P}\left(\lambda_{\min}\left(\sum_{i=1}^{t-1} z_i z_i^\mathsf{T}\right) \leq \frac{(t-1)\alpha}{2}\right) \leq p\exp\left(\frac{-(t-1)\alpha}{8\zeta^2}\right).
\tag{97}
$$

Let $p\exp\left(\frac{-(t-1)\alpha}{8\zeta^2}\right) \leq \delta$, then we have

$$
t \geq \frac{8\zeta^2 \log(p/\delta)}{\alpha}.
$$

Then, with probability at least $1 - \delta$ we have

$$
\lambda_{\min}\left(\sum_{i=1}^{t-1} z_i z_i^\mathsf{T}\right) \geq \frac{(t-1)\alpha}{2}.
$$

Finally, we have

$$
\lambda_{\min}(X_t) \geq \lambda_{\min}\left(\sum_{i=1}^{t-1} z_i z_i^\mathsf{T}\right) + \gamma \geq \frac{(t-1)\alpha}{2} + \gamma.
$$

$\square$

**Lemma E.2.** *Consider the system eq. (1) and the Markov process eq. (2). Let Assumption 2.1 be satisfied, and let*

$$x_{t+1} = \varphi(t+1, 0)x_0 + \sum_{i=0}^{t} \varphi(t+1, i+1)B\overline{u}_i(s_i),$$

*with $x_0 \sim \mathcal{N}(0, \Sigma_0)$, and $\overline{u}_i(s_i)$ is an arbitrary input that depends on $s_i$ and is independent of $x_0$. Let*

$$\psi_t = \phi(s_t) \otimes \begin{bmatrix} 2x_{t+1} \\ 1 \end{bmatrix}.$$

*Assume $\Sigma_0 \succ 0$ and $\varphi(t+1, 0)$ is nonsingular for $t \in \{0, \cdots, T-1\}$. Then, $\mathbb{E}\left[\psi_t \psi_t^\mathsf{T}\right] \succ 0$ for $t \in \{0, \cdots, T-1\}$.*

*Proof.* We begin by writing

$$\psi_t = \phi(s_t) \otimes \begin{bmatrix} 2x_{t+1} \\ 1 \end{bmatrix} = \begin{bmatrix} 2\phi(s_t) \otimes x_{t+1} \\ \phi(s_t) \end{bmatrix}.$$

Then,

$$\psi_t \psi_t^\mathsf{T} = \begin{bmatrix} 4\phi(s_t)\phi(s_t)^\mathsf{T} \otimes x_{t+1}x_{t+1}^\mathsf{T} & 2\phi(s_t)\phi(s_t)^\mathsf{T} \otimes x_{t+1} \\ 2\phi(s_t)\phi(s_t)^\mathsf{T} \otimes x_{t+1}^\mathsf{T} & \phi(s_t)\phi(s_t)^\mathsf{T} \end{bmatrix}.$$

Taking the expectation, we get

$$\mathbb{E}\left[\psi_t \psi_t^\mathsf{T}\right] = \begin{bmatrix} 4\mathbb{E}\left[\phi(s_t)\phi(s_t)^\mathsf{T} \otimes x_{t+1}x_{t+1}^\mathsf{T}\right] & 2\mathbb{E}\left[\phi(s_t)\phi(s_t)^\mathsf{T} \otimes x_{t+1}\right] \\ 2\mathbb{E}\left[\phi(s_t)\phi(s_t)^\mathsf{T} \otimes x_{t+1}^\mathsf{T}\right] & \mathbb{E}\left[\phi(s_t)\phi(s_t)^\mathsf{T}\right] \end{bmatrix}.$$

From Assumption 2.1, we have $\mathbb{E}\left[\phi(s_t)\phi(s_t)^\mathsf{T}\right] \succ 0$ for all $t$. For notational convenience we denote $\Sigma_\phi = \mathbb{E}\left[\phi(s_t)\phi(s_t)^\mathsf{T}\right]$. We apply the Schur complement

$$S = 4\mathbb{E}\left[\phi(s_t)\phi(s_t)^\mathsf{T} \otimes x_{t+1}x_{t+1}^\mathsf{T}\right] - 4\mathbb{E}\left[\phi(s_t)\phi(s_t)^\mathsf{T} \otimes x_{t+1}\right] \Sigma_\phi^{-1} \mathbb{E}\left[\phi(s_t)\phi(s_t)^\mathsf{T} \otimes x_{t+1}^\mathsf{T}\right]. \tag{98}$$

Showing $\mathbb{E}\left[\psi_t \psi_t^\mathsf{T}\right] \succ 0$ boils down to showing that $S \succ 0$. From eq. (98), we have

$$\begin{aligned} \mathbb{E}\left[\phi(s_t)\phi(s_t)^\mathsf{T} \otimes x_{t+1}x_{t+1}^\mathsf{T}\right] &\overset{(a)}{=} \mathbb{E}\left[\mathbb{E}\left[\phi(s_t)\phi(s_t)^\mathsf{T} \otimes x_{t+1}x_{t+1}^\mathsf{T} \big| s_t\right]\right] \\ &= \mathbb{E}\left[\phi(s_t)\phi(s_t)^\mathsf{T} \otimes \mathbb{E}\left[x_{t+1}x_{t+1}^\mathsf{T} \big| s_t\right]\right] \\ &= \mathbb{E}\left[\phi(s_t)\phi(s_t)^\mathsf{T} \otimes \Sigma_{x|s}\right] + \mathbb{E}\left[\phi(s_t)\phi(s_t)^\mathsf{T} \otimes \mu_x(s_t)\mu_x(s_t)^\mathsf{T}\right], \end{aligned} \tag{99}$$

where in step (a) we used the law of total expectation, and

$$\begin{aligned} \Sigma_{x|s} &= \mathbb{E}\left[(x_{t+1} - \mathbb{E}\left[x_{t+1}|s_t\right])(x_{t+1} - \mathbb{E}\left[x_{t+1}|s_t\right])^\mathsf{T} \big| s_t\right], \\ \mu_x(s_t) &= \mathbb{E}\left[x_{t+1}|s_t\right]. \end{aligned}$$

For notational convenience, let $z = \phi(s_t) \otimes \mu_x(s_t)$. Substituting eq. (99) in eq. (98), we get

$$S = \underbrace{4\mathbb{E}\left[\phi(s_t)\phi(s_t)^\mathsf{T} \otimes \Sigma_{x|s}\right]}_{S_1} + \underbrace{4\mathbb{E}\left[zz^\mathsf{T}\right] - 4\mathbb{E}\left[z\phi(s_t)^\mathsf{T}\right]\Sigma_\phi^{-1}\mathbb{E}\left[\phi(s_t)z^\mathsf{T}\right]}_{S_2}. \tag{100}$$

We have $S_1 \succeq 0$ since $\phi(s_t)\phi(s_t)^\mathsf{T} \succeq 0$ and $\Sigma_{x|s} \succeq 0$. From eq. (100), we have

$$
\begin{aligned}
S_2 =& 4\mathbb{E}\left[zz^\mathsf{T}\right] - 4\mathbb{E}\left[z\phi(s_t)^\mathsf{T}\right]\Sigma_\phi^{-1}\mathbb{E}\left[\phi(s_t)z^\mathsf{T}\right] \\
\overset{(b)}{=}& 4\mathbb{E}\left[zz^\mathsf{T}\right] - 4\mathbb{E}\left[z\phi(s_t)^\mathsf{T}\Sigma_\phi^{-1}\mathbb{E}\left[\phi(s_t)z^\mathsf{T}\right]\right] + 4\mathbb{E}\left[\mathbb{E}\left[z\phi(s_t)^\mathsf{T}\right]\Sigma_\phi^{-1}\phi(s_t)z^\mathsf{T}\right] \\
& - 4\mathbb{E}\left[\mathbb{E}\left[z\phi(s_t)^\mathsf{T}\right]\Sigma_\phi^{-1}\phi(s_t)z^\mathsf{T}\right] \\
\overset{(c)}{=}& 4\mathbb{E}\left[zz^\mathsf{T}\right] - 4\mathbb{E}\left[z\phi(s_t)^\mathsf{T}\Sigma_\phi^{-1}\mathbb{E}\left[\phi(s_t)z^\mathsf{T}\right]\right] + 4\mathbb{E}\left[z\phi(s_t)^\mathsf{T}\right]\Sigma_\phi^{-1}\Sigma_\phi\Sigma_\phi^{-1}\mathbb{E}\left[\phi(s_t)z^\mathsf{T}\right] \\
& - 4\mathbb{E}\left[\mathbb{E}\left[z\phi(s_t)^\mathsf{T}\right]\Sigma_\phi^{-1}\phi(s_t)z^\mathsf{T}\right] \\
=& 4\mathbb{E}\Big[zz^\mathsf{T} - z\phi(s_t)^\mathsf{T}\Sigma_\phi^{-1}\mathbb{E}\left[\phi(s_t)z^\mathsf{T}\right] - \mathbb{E}\left[z\phi(s_t)^\mathsf{T}\right]\Sigma_\phi^{-1}\phi(s_t)z^\mathsf{T} \\
& + \mathbb{E}\left[z\phi(s_t)^\mathsf{T}\right]\Sigma_\phi^{-1}\phi(s_t)\phi(s_t)^\mathsf{T}\Sigma_\phi^{-1}\mathbb{E}\left[\phi(s_t)z^\mathsf{T}\right]\Big] \\
=& 4\mathbb{E}\left[\left(z - \mathbb{E}\left[z\phi(s_t)^\mathsf{T}\right]\Sigma_\phi^{-1}\phi(s_t)\right)\left(z - \mathbb{E}\left[z\phi(s_t)^\mathsf{T}\right]\Sigma_\phi^{-1}\phi(s_t)\right)^\mathsf{T}\right],
\end{aligned}
$$

where in step (b) we have added and subtracted the term $4\mathbb{E}\left[\mathbb{E}\left[z\phi(s_t)^\mathsf{T}\right]\Sigma_\phi^{-1}\phi(s_t)z^\mathsf{T}\right]$, and in step (c) we have used $I = \Sigma_\phi\Sigma_\phi^{-1}$. Then, we have $S = S_1 + S_2 \succeq 0$. From eq. (92) we have $x_{t+1} = \varphi(t+1,0)x_0 + \sum_{i=0}^t \varphi(t+1,i+1)B\overline{u}_i$, hence, $\mathbb{E}\left[x_{t+1}|s_t\right] = \mathbb{E}\left[\sum_{i=0}^t \varphi(t+1,i+1)B\overline{u}_i|s_t\right]$. For notational convenience, let $\widetilde{u}(t) = \sum_{i=0}^t \varphi(t+1,i+1)B\overline{u}_i$. Then we get

$$
\begin{aligned}
\Sigma_{x|s} =& \varphi(t+1,0)\Sigma_0\varphi(t+1,0)^\mathsf{T} + \mathbb{E}\left[\left(\widetilde{u}(t) - \mathbb{E}\left[\widetilde{u}(t)|s_t\right]\right)\left(\widetilde{u}(t) - \mathbb{E}\left[\widetilde{u}(t)|s_t\right]\right)^\mathsf{T}|s_t\right] \\
\succeq& \varphi(t+1,0)\Sigma_0\varphi(t+1,0)^\mathsf{T}.
\end{aligned}
$$

Since $\Sigma_0 \succ 0$ and $\varphi(t+1,0)$ is nonsingular for all $t$, then, $\varphi(t+1,0)\Sigma_0\varphi(t+1,0)^\mathsf{T} \succ 0$. Then, we have $S_1 = 4\mathbb{E}\left[\phi(s_t)\phi(s_t)^\mathsf{T} \otimes \Sigma_{x|s}\right] \succeq 4\mathbb{E}\left[\phi(s_t)\phi(s_t)^\mathsf{T}\right] \otimes \left(\varphi(t+1,0)\Sigma_0\varphi(t+1,0)^\mathsf{T}\right) \succ 0$ since $\mathbb{E}\left[\phi(s_t)\phi(s_t)^\mathsf{T}\right] \succ 0$, and $\varphi(t+1,0)\Sigma_0\varphi(t+1,0)^\mathsf{T}$ is independent of $s$. Therefore, $S \succ 0$, which implies $\mathbb{E}\left[\psi_t\psi_t^\mathsf{T}\right] \succ 0$ for $t \in \{0, \cdots, T-1\}$. $\qquad\square$

Now we present the proof of Theorem 3.4. For notational convenience, we denote $\phi\left(s_t^i\right)$ and $Y\left(x_t^i, u_t^i\right)$ by $\phi_t^i$ and $Y_t^i$, respectively. We have from the expression of $\theta_{t+1}^\ell$ in eq. (15)

$$
\begin{aligned}
\epsilon_{t+1}^\ell(x_{t+1}^i, s_{t+1}^i) &= 2x_{t+1}^i h_{t+1}^\ell(s_{t+1}^i) + q_{t+1}^\ell(s_{t+1}^i) \\
&= \underbrace{\begin{bmatrix} 2x_{t+1}^i & 1 \end{bmatrix}}_{(y_t^i)^\mathsf{T}} \underbrace{\begin{bmatrix} h_{t+1}^\ell(s_{t+1}^i) \\ q_{t+1}^\ell(s_{t+1}^i) \end{bmatrix}}_{v_{t+1}^\ell(s_{t+1}^i)}.
\end{aligned}
$$

We can derive an upper bound on $|\epsilon_{t+1}^\ell(x_{t+1}^i, s_{t+1}^i)|$ as

$$
|\epsilon_{t+1}^\ell(x_{t+1}^i, s_{t+1}^i)| \leq 2\|x_{t+1}^i\|\|h_{t+1}^\ell(s_{t+1}^i)\| + \|q_{t+1}^\ell(s_{t+1}^i)\|. \tag{101}
$$

Using eq. (49) we can write

$$
\begin{aligned}
\|h_{t+1}^\ell(s_{t+1}^i)\| &\leq \|X_1(t+1)\|\|\phi_{t+1}^i\|\|\theta_{t+2}^\ell\| + \|X_2(t+1)\|\|s_{t+1}^i\| \\
&\leq \frac{\overline{X}_1 R_\theta}{\sqrt{d}} + \overline{X}_2\delta_s,
\end{aligned} \tag{102}
$$

where $\|X_1(t)\| \leq \overline{X}_1$ and $\|X_2(t)\| \leq \overline{X}_2$ for all $t \in \{0, \cdots, T\}$. From eq. (50) we can write

$$
\begin{aligned}
q_{t+1}^\ell(s_{t+1}^i) \leq & \|\phi_{t+1}^i\| \|\theta_{t+2}^\ell\| + \|s_{t+1}^i\|^2 \|Y_1(t+1)\| \\
& + \|\theta_{t+2}^\ell\|^2 \|\phi_{t+1}^i\|^2 \|Y_2(t+1)\| \\
& + 2\|s_{t+1}^i\| \|Y_3(t+1)\| \|\phi_{t+1}^i\| \|\theta_{t+2}^\ell\| \\
\leq & \frac{R_\theta}{\sqrt{d}} + \delta_s^2 \overline{Y}_1 + \frac{R_\theta^2 \overline{Y}_2}{d} + 2\frac{\delta_s \overline{Y}_3 R_\theta}{\sqrt{d}},
\end{aligned}
\tag{103}
$$

where $\|Y_1(t)\| \leq \overline{Y}_1$, $\|Y_2(t)\| \leq \overline{Y}_2$, and $\|Y_3(t)\| \leq \overline{Y}_3$ for all $t \in \{0, \cdots, T\}$. From Theorem 3.3, we have

$$
\|x_t^\ell\| \leq \alpha \rho^t \|x^\ell(0)\| + \frac{\alpha \|B\|}{1-\rho} \left( \overline{K}_s \delta_s + \frac{\overline{K}_h R_\theta}{\sqrt{d}} \right),
$$

for $t \in \{0, \cdots, T\}$ and $\ell \in \{1, \cdots, L\}$, with $\alpha > 0$ and $0 < \rho < 1$. Define

$$
\overline{x} = \sup_{\substack{t \in \{0, \cdots, T\} \\ \ell \in \{1, \cdots, L\}}} \left\{ \alpha \rho^t \|x^\ell(0)\| + \frac{\alpha \|B\|}{1-\rho} \left( \overline{K}_s \delta_s + \frac{\overline{K}_h R_\theta}{\sqrt{d}} \right) \right\}.
\tag{104}
$$

Substituting eq. (102), eq. (103), and eq. (104) in eq. (101), we get

$$
|\epsilon_{t+1}^\ell(x_{t+1}^i, s_{t+1}^i)| \leq \left( \frac{2\overline{x}\,\overline{X}_1 + 1 + 2\delta_s \overline{Y}_3}{\sqrt{d}} \right) R_\theta + \frac{\overline{Y}_2}{d} R_\theta^2 + 2\overline{X}_2 \overline{x} \delta_s + \overline{Y}_1 \delta_s^2.
\tag{105}
$$

Further, we can bound

$$
\begin{aligned}
\|\psi_t^i\| = \|(Y_t^i)^\mathsf{T} \phi_t^i\| \leq & \left\| \begin{bmatrix} 2x_{t+1}^i & 1 \end{bmatrix} \right\| \|\phi_t^i\| \\
\leq & \sqrt{\frac{4\overline{x}^2 + 1}{d}} \triangleq \delta_\psi
\end{aligned}
\tag{106}
$$

Next, using the expression of $\theta_{t+1}^\ell$ in eq. (15), we write

$$
\theta_{t+1}^\ell - \theta_{t+1}^* = (\Lambda_t^\ell)^{-1} \sum_{i=1}^{\ell-1} \left( Y_t^i \right)^\mathsf{T} \phi_t^i \epsilon_{t+1}^\ell(x_{t+1}^i, s_{t+1}^i) - \theta_{t+1}^*,
\tag{107}
$$

We re-write eq. (107) as

$$
\begin{aligned}
\theta_{t+1}^\ell - \theta_{t+1}^* = & (\Lambda_t^\ell)^{-1} \left( \sum_{i=1}^{\ell-1} (Y_t^i)^\mathsf{T} (\phi_t^i)(y_t^i)^\mathsf{T} v_{t+1}^\ell(s_{t+1}^i) - \Lambda_t^\ell \theta_{t+1} \right) \\
= & (\Lambda_t^\ell)^{-1} \left( \sum_{i=1}^{\ell-1} \left( Y_t^i \right)^\mathsf{T} \left( \phi_t^i \right) \left( y_t^i \right)^\mathsf{T} v_{t+1}^\ell(s_{t+1}^i) - \sum_{i=1}^{\ell-1} (Y_t^i)^\mathsf{T} (\phi_t^i)(\phi_t^i)^\mathsf{T} Y_t^i \theta_{t+1} \right) - \lambda (\Lambda_t^\ell)^{-1} \theta_{t+1}^*.
\end{aligned}
\tag{108}
$$

From eq. (108), we expand the term as

$$
\begin{aligned}
\left( \phi_t^i \right)^\mathsf{T} Y_t^i \theta_{t+1}^* = & \sum_{j=1}^d \phi_{j,t}^i \left( y_t^i \right)^\mathsf{T} \begin{bmatrix} \overline{h}_{j,t+1}^* \\ \overline{q}_{j,t+1}^* \end{bmatrix} \\
= & \left( y_t^i \right)^\mathsf{T} \sum_{j=1}^d \phi_{j,t}^i \mathbb{E}_{\mu_j} \left[ \begin{bmatrix} h_{t+1}^*(s_{t+1}) \\ q_{t+1}^*(s_{t+1}) \end{bmatrix} \right] \\
= & \left( y_t^i \right)^\mathsf{T} \sum_{j=1}^d \phi_{j,t}^i \int_{\mathcal{S}} \underbrace{\begin{bmatrix} h_{t+1}^*(s_{t+1}) \\ q_{t+1}^*(s_{t+1}) \end{bmatrix}}_{v_{t+1}^*(s_{t+1})} \mu_j(ds_{t+1}) \\
= & \left( y_t^i \right)^\mathsf{T} \int_{\mathcal{S}} v_{t+1}^*(s_{t+1}) \sum_{j=1}^d \phi_{j,t}^i \mu_j(ds_{t+1}) \\
= & \left( y_t^i \right)^\mathsf{T} \mathbb{E} \left[ v_{t+1}^*(s_{t+1}) | s_t^i \right].
\end{aligned}
\tag{109}
$$

For notational convenience, we use $\psi_t^i = (Y_t^i)^\mathsf{T}\phi_t^i$. Substituting eq. (109) in eq. (108), we get

$$
\begin{aligned}
\theta_{t+1}^\ell - \theta_{t+1}^* =& (\Lambda_t^\ell)^{-1}\left(\sum_{i=1}^{\ell-1}\psi_t^i(y_t^i)^\mathsf{T}\left(v_{t+1}^\ell(s_{t+1}^i) - \mathbb{E}\left[v_{t+1}^\ell(s_{t+1})|s_t^i\right]\right)\right) - \lambda(\Lambda_t^\ell)^{-1}\theta_{t+1}^* \\
=& \underbrace{(\Lambda_t^\ell)^{-1}\left(\sum_{i=1}^{\ell-1}\psi_t^i(y_t^i)^\mathsf{T}\left(v_{t+1}^\ell(s_{t+1}^i) - \mathbb{E}\left[v_{t+1}^\ell(s_{t+1})|s_t^i\right]\right)\right)}_{r_1} \\
&+ \underbrace{(\Lambda_t^\ell)^{-1}\left(\sum_{i=1}^{\ell-1}\psi_t^i(y_t^i)^\mathsf{T}\left(\mathbb{E}\left[v_{t+1}^\ell(s_{t+1}) - v_{t+1}^*(s_{t+1})|s_t^i\right]\right)\right)}_{r_2} - \underbrace{\lambda(\Lambda_t^\ell)^{-1}\theta_{t+1}^*}_{r_3}.
\end{aligned}
\tag{110}
$$

Let $\xi_{i,t+1}^\ell = (y_t^i)^\mathsf{T}v_{t+1}^\ell(s_{t+1}^i) - \mathbb{E}\left[(y_t^i)^\mathsf{T}v_{t+1}^\ell(s_{t+1})|s_t^i\right]$. We have $\mathbb{E}\left[\xi_{i,t+1}^\ell|\mathcal{F}_t^{\ell-1}\right] = 0$. Since $(y_t^i)^\mathsf{T}v_{t+1}^\ell(s_{t+1}^i)$ is bounded (see eq. (105)), then $\xi$ is $\sigma$-subGaussian with

$$
\sigma = \left(\frac{2\overline{x}\,\overline{X}_1 + 1 + 2\delta_s\overline{Y}_3}{\sqrt{d}}\right)R_\theta + \frac{\overline{Y}_2}{d}R_\theta^2 + 2\overline{X}_2\overline{x}\delta_s + \overline{Y}_1\delta_s^2.
\tag{111}
$$

Then, using (Abbasi-Yadkori et al., 2011, Theorem 1), we have with probability at least $1 - \delta$, we have

$$
\left\|\sum_{i=1}^{\ell-1}(\psi_t^i(y_t^i)^\mathsf{T}\left(v_{t+1}^\ell(s_{t+1}^i) - \mathbb{E}\left[v_{t+1}^\ell(s_{t+1})|s_t^i\right]\right)\right\|_{(\Lambda_t^\ell)^{-1}}^2 \leq 2\sigma^2\left(\log\left(\sqrt{\frac{\det\left(\Lambda_t^\ell\right)}{\det\left(\Lambda_t^1\right)}}\right) + \log\left(\frac{1}{\delta}\right)\right).
\tag{112}
$$

Recall from Alg. 1, we have

$$
\begin{aligned}
\Lambda_t^1 &= \lambda I_{d(n+1)}, \\
\Lambda_t^\ell &= \sum_{i=1}^{\ell-1}\left(Y_t^i\right)^\mathsf{T}\phi_t^i\left(\phi_t^i\right)^\mathsf{T}Y_t^i + \lambda I_{d(n+1)} = \lambda\underbrace{\left(\frac{1}{\lambda}\sum_{i=1}^{\ell-1}\left(Y_t^i\right)^\mathsf{T}\phi_t^i\left(\phi_t^i\right)^\mathsf{T}Y_t^i + I_{d(n+1)}\right)}_{\overline{\Lambda}_t^\ell}.
\end{aligned}
$$

Then,

$$
\frac{\det\left(\Lambda_t^\ell\right)}{\det\left(\Lambda_t^1\right)} = \frac{\det\left(\lambda\overline{\Lambda}_t^\ell\right)}{\det\left(\lambda I_{d(n+1)}\right)} = \det\left(\overline{\Lambda}_t^\ell\right) = \det\left(\frac{1}{\lambda}\sum_{i=1}^{\ell-1}\left(Y_t^i\right)^\mathsf{T}\phi_t^i\left(\phi_t^i\right)^\mathsf{T}Y_t^i + I_{d(n+1)}\right) = \prod_{i=1}^{d(n+1)}(1 + \gamma_i),
\tag{113}
$$

where $\gamma_i$ is the $i$-th eigenvalue of $\frac{1}{\lambda}\sum_{i=1}^{\ell-1}\left(Y_t^i\right)^\mathsf{T}\phi_t^i\left(\phi_t^i\right)^\mathsf{T}Y_t^i$. Let $\left\|\left(Y_t^i\right)^\mathsf{T}\phi_t^i\right\| \leq \delta_\psi$ for $t \in \{0, \cdots, T\}$ and $i \in \{1, \cdots, L\}$ (see eq. (106)). Then,

$$
\gamma_i \leq \frac{1}{\lambda}\sum_{i=1}^{\ell-1}\mathrm{Tr}\left(Y_t^i\right)^\mathsf{T}\phi_t^i\left(\phi_t^i\right)^\mathsf{T}Y_t^i \leq \frac{1}{\lambda}\sum_{i=1}^{\ell-1}\left\|\left(Y_t^i\right)^\mathsf{T}\phi_t^i\right\|^2 \leq \frac{\ell\delta_\psi^2}{\lambda}.
\tag{114}
$$

Then

$$
\log\left(\sqrt{\det\left(\overline{\Lambda}_t^\ell\right)}\right) = \frac{1}{2}\log\left(\prod_{i=1}^{d(n+1)}(1 + \gamma_i)\right) = \frac{1}{2}\sum_{i=1}^{d(n+1)}\log(1 + \gamma_i) \leq \frac{d(n+1)}{2}\log\left(1 + \frac{\ell\delta_\psi^2}{\lambda}\right).
\tag{115}
$$

Substituting eq. (115) in eq. (112), we get with probability at least $1 - \delta$

$$
\left\|\sum_{i=1}^{\ell-1}(\psi_t^i(y_t^i)^\mathsf{T}\left(v_{t+1}^\ell(s_{t+1}^i) - \mathbb{E}\left[v_{t+1}^\ell(s_{t+1})|s_t^i\right]\right)\right\|_{(\Lambda_t^\ell)^{-1}}^2 \leq \sigma^2\left(d(n+1)\log\left(1 + \frac{\ell\delta_\psi^2}{\lambda}\right) + 2\log\left(\frac{1}{\delta}\right)\right).
\tag{116}
$$

Then,

$$\left\| \sum_{i=1}^{\ell-1} (\psi_t^i(y_t^i)^\mathsf{T} \left( v_{t+1}^\ell(s_{t+1}^i) - \mathbb{E}\left[ v_{t+1}^\ell(s_{t+1})|s_t^i \right] \right) \right\|_{(\Lambda_t^\ell)^{-1}} \le \sigma \sqrt{d(n+1)\log\left(1 + \frac{\ell\delta_\psi^2}{\lambda}\right) + 2\log\left(\frac{1}{\delta}\right)}. \quad (117)$$

Then, we have

$$\begin{aligned}
|\phi_t^\mathsf{T} Y_t r_1| &\le \left\| \phi_t^\mathsf{T} Y_t (\Lambda_t^\ell)^{-\frac{1}{2}} \right\| \left\| \sum_{i=1}^{\ell-1} (\psi_t^i(y_t^i)^\mathsf{T} \left( v_{t+1}^\ell(s_{t+1}^i) - \mathbb{E}\left[ v_{t+1}^\ell(s_{t+1})|s_t^i \right] \right) \right\|_{(\Lambda_t^\ell)^{-1}} \\
&\le \sigma \sqrt{\left( d(n+1)\log\left(1 + \frac{\ell\delta_\psi^2}{\lambda}\right) + 2\log\left(\frac{1}{\delta}\right) \right)} \sqrt{\phi_t^\mathsf{T} Y_t \left(\Lambda_t^\ell\right)^{-1} Y_t^\mathsf{T} \phi_t}.
\end{aligned} \quad (118)$$

Next, we have

$$\begin{aligned}
|\phi_t^\mathsf{T} Y_t r_3| &\le \lambda \left\| \phi_t^\mathsf{T} Y_t \left(\Lambda_t^\ell\right)^{-\frac{1}{2}} \right\| \left\| \left(\Lambda_t^\ell\right)^{-\frac{1}{2}} \right\| \|\theta_{t+1}^*\| \\
&\le \sqrt{\lambda} \|\theta_{t+1}^*\| \sqrt{\phi_t^\mathsf{T} Y_t \left(\Lambda_t^\ell\right)^{-1} Y_t^\mathsf{T} \phi_t}.
\end{aligned} \quad (119)$$

Next, we have

$$\begin{aligned}
r_2 &= (\Lambda_t^\ell)^{-1} \left( \sum_{i=1}^{\ell-1} (Y_t^i)^\mathsf{T} \phi_t^i(y_t^i)^\mathsf{T} \left( \mathbb{E}\left[ v_{t+1}^\ell(s_{t+1})|s_t^i \right] - \mathbb{E}\left[ v_{t+1}^*(s_{t+1})|s_t^i \right] \right) \right) \\
&= (\Lambda_t^\ell)^{-1} \left( \sum_{i=1}^{\ell-1} (Y_t^i)^\mathsf{T} \phi_t^i(y_t^i)^\mathsf{T} \int_{\mathcal{S}} \left( v_{t+1}^\ell(s_{t+1}) - v_{t+1}^*(s_{t+1}) \right) \sum_{j=1}^d \phi_{j,t}^i \mu_{j,t}(ds_{t+1}) \right) \\
&= (\Lambda_t^\ell)^{-1} \left( \sum_{i=1}^{\ell-1} (Y_t^i)^\mathsf{T} \phi_t^i(y_t^i)^\mathsf{T} \sum_{j=1}^d \phi_{j,t}^i \int_{\mathcal{S}} \left( v_{t+1}^\ell(s_{t+1}) - v_{t+1}^*(s_{t+1}) \right) \mu_{j,t}(ds_{t+1}) \right) \\
&= (\Lambda_t^\ell)^{-1} \underbrace{\left( \sum_{i=1}^{\ell-1} \psi_t^i(\psi_t^i)^\mathsf{T} \right)}_{\Lambda_t^\ell - \lambda I_{d(n+1)}} \mathbb{E}_{\mu_{t+1}}[v_{t+1}^\ell(s_{t+1}) - v_{t+1}^*(s_{t+1})] \\
&= \mathbb{E}_{\mu_{t+1}}\left[ v_{t+1}^\ell(s_{t+1}) - v_{t+1}^*(s_{t+1}) \right] - \lambda(\Lambda_t^\ell)^{-1} \mathbb{E}_{\mu_{t+1}}\left[ v_{t+1}^\ell(s_{t+1}) - v_{t+1}^*(s_{t+1}) \right].
\end{aligned} \quad (120)$$

Then, we have

$$\begin{aligned}
\phi_t^\mathsf{T} Y_t r_2 &= \phi_t^\mathsf{T} Y_t \mathbb{E}_{\mu_{t+1}}\left[ v_{t+1}^\ell(s_{t+1}) - v_{t+1}^*(s_{t+1}) \right] - \lambda\phi_t^\mathsf{T} Y_t(\Lambda_t^\ell)^{-1}\mathbb{E}_{\mu_{t+1}}\left[ v_{t+1}^\ell(s_{t+1}) - v_{t+1}^*(s_{t+1}) \right] \\
&= y_t^\mathsf{T} \mathbb{E}\left[ v_{t+1}^\ell(s_{t+1}) - v_{t+1}^*(s_{t+1})|s_t \right] - \lambda\phi_t^\mathsf{T} Y_t(\Lambda_t^\ell)^{-1}\mathbb{E}_{\mu_{t+1}}\left[ v_{t+1}^\ell(s_{t+1}) - v_{t+1}^*(s_{t+1}) \right] \\
&= \mathbb{E}\left[ V_{t+1}^\ell(x_{t+1}, s_{t+1}) - V_{t+1}(x_{t+1}, s_{t+1})|s_t \right] - \underbrace{\lambda\phi_t^\mathsf{T} Y_t(\Lambda_t^\ell)^{-1}\mathbb{E}_{\mu_{t+1}}\left[ v_{t+1}^\ell(s_{t+1}) - v_{t+1}^*(s_{t+1}) \right]}_{r_4}.
\end{aligned} \quad (121)$$

Since we choose $R_\theta$ in Algorithm 1 to be the upper bound on $\|\theta_{t+1}^*\|$ (which we derive in Appendix C.2) for $t \in \{0, \cdots, T-1\}$, and we have $\|\theta_{t+1}^\ell\| \le R_\theta$ (from Algorithm 1) for $t \in \{0, \cdots, T-1\}$ and $\ell \in \{1, \cdots, L\}$, it can be seen from eq. (102) and eq. (103) that $\|v_{t+1}^\ell(s_{t+1})\| \le \delta_v$ and $\|v_{t+1}^*(s_{t+1})\| \le \delta_v$. We can derive an expression for $\delta_v$ using eq. (102) and eq. (103) as

$$\|v_{t+1}^\ell(s_{t+1})\| = \left\| \begin{bmatrix} h_{t+1}^\ell(s_{t+1}) \\ q_{t+1}^\ell(s_{t+1}) \end{bmatrix} \right\| \le \underbrace{\sqrt{\|h_{t+1}^\ell(s_{t+1})\|^2 + \|q_{t+1}^\ell(s_{t+1})\|^2}}_{\delta_v}. \quad (122)$$

Then, we can bound $|r_4|$ in eq. (121) as

$$\left|\lambda\phi_t^\mathsf{T}Y_t(\Lambda_t^\ell)^{-1}\mathbb{E}_{\mu_{t+1}}\left[v_{t+1}^\ell(s_{t+1})-v_{t+1}^*(s_{t+1})\right]\right| \leq \lambda\left\|\phi_t^\mathsf{T}Y_t(\Lambda_t^\ell)^{-\frac{1}{2}}\right\|\left\|(\Lambda_t^\ell)^{-\frac{1}{2}}\right\|\left\|\mathbb{E}_{\mu_{t+1}}\left[v_{t+1}^\ell-v_{t+1}^*\right]\right\|$$
$$\leq 2\sqrt{\lambda}\delta_v\sqrt{\phi_t^\mathsf{T}Y_t\left(\Lambda_t^\ell\right)^{-1}Y_t^\mathsf{T}\phi_t}.$$

(123)

Using the parametrized form of the $Q$-function from Theorem 3.1, we have

$$Q_t^\ell(x_t,s_t,u_t)-Q_t^*(x_t,s_t,u_t)=\phi_t^\mathsf{T}Y_t\left(\theta_{t+1}^\ell-\theta_{t+1}^*\right)=\phi_t^\mathsf{T}Y_t\left(r_1+r_2+r_3\right)$$
$$\implies Q_t^\ell(x_t,s_t,u_t)-Q_t^*(x_t,s_t,u_t)-\mathbb{E}\left[V_{t+1}^\ell(x_{t+1},s_{t+1})-V_{t+1}^*(x_{t+1},s_{t+1})|s_t\right]=\underbrace{\phi_t^\mathsf{T}Y_t\left(r_1+r_3\right)-r_4}_{\Delta_t^\ell(x_t,s_t,u_t)}.$$

(124)

Then, using eq. (118), eq. (119), and eq. (123), we can bound

$$\left|\Delta_t^\ell(x_t,s_t,u_t)\right| \leq \left(\sigma\sqrt{\left(d(n+1)\log\left(1+\frac{\ell\delta_\psi^2}{\lambda}\right)+2\log\left(\frac{1}{\delta}\right)\right)}+\left\|\theta_{t+1}^*\right\|\sqrt{\lambda}+2\sqrt{\lambda}\delta_v\right)\sqrt{\phi_t^\mathsf{T}Y_t\left(\Lambda_t^\ell\right)^{-1}Y_t^\mathsf{T}\phi_t}$$
$$=\chi(\ell)\sqrt{\phi_t^\mathsf{T}Y_t\left(\Lambda_t^\ell\right)^{-1}Y_t^\mathsf{T}\phi_t}.$$

(125)

Let $\delta_t^\ell=V_t^\ell(x_t^{*\ell},s_t^\ell)-V_t^*(x_t^{*\ell},s_t^\ell)$ and $\zeta_{t+1}^\ell=\mathbb{E}\left[\delta_{t+1}^\ell|s_t^\ell\right]-\delta_{t+1}^\ell$, where $x_t^{*\ell}$ is the state under the optimal policy $\pi_t^*$ starting from $x_0^\ell$ for $t\in\{0,\cdots,T\}$ and $\ell\in\{1,\cdots,L\}$. From the definition of the value function, we have $V_t^*(x,s)=\min_{u\in\mathcal{U}}Q_t^*(x,s,u)$. Further, since Algorithm 1 selects a greedy policy with respect to $Q_t^\ell(x,s,u)$, we have $V_t^\ell(x,s)=\min_{u\in\mathcal{U}}Q_t^\ell(x,s,u)$. Let $u_t^{*\ell}$ be the optimal control input at that generates $x_t^{*\ell}$ for $t\in\{0,\cdots,T-1\}$ and $\ell\in\{1,\cdots,L\}$. Then, we can write

$$\delta_t^\ell=V_t^\ell(x_t^{*\ell},s_t^\ell)-V_t^*(x_t^{*\ell},s_t^\ell)$$
$$=Q_t^\ell(x_t^{*\ell},s_t^\ell,u_t^\ell)-Q_t^*(x_t^{*\ell},s_t^\ell,u_t^{*\ell})$$
$$\leq Q_t^\ell(x_t^{*\ell},s_t^\ell,u_t^{*\ell})-Q_t^*(x_t^{*\ell},s_t^\ell,u_t^{*\ell}).$$

(126)

Then, from eq. (124) we can write

$$Q_t^\ell(x_t^{*\ell},s_t^\ell,u^{*\ell})-Q_t^*(x_t^{*\ell},s_t^\ell,u^{*\ell})=\mathbb{E}\left[\delta_{t+1}^\ell|s_t^\ell\right]+\Delta_t^\ell(x_t^{*\ell},s_t^\ell,u_t^{*\ell})=\delta_{t+1}^\ell+\zeta_{t+1}^\ell+\Delta_t^\ell(x_t^{*\ell},s_t^\ell,u_t^{*\ell})$$

(127)

Substituting eq. (126) in eq. (127), we get

$$\delta_t^\ell\leq\delta_{t+1}^\ell+\zeta_{t+1}^\ell+\Delta_t^\ell(x_t^{*\ell},s_t^\ell,u_t^{*\ell}).$$

Note that $x_0^{*\ell}=x_0^\ell$. Next, from eq. (17) we write

$$\mathcal{R}(L)=\sum_{\ell=1}^L\delta_0^\ell\leq\sum_{\ell=1}^L\sum_{t=1}^T\zeta_t^\ell+\sum_{\ell=1}^L\sum_{t=0}^{T-1}\Delta_t^\ell(x_t^{*\ell},s_t^\ell,u_t^{*\ell}).$$

(128)

We have $\mathbb{E}\left[\zeta_t^\ell|\mathcal{F}_{t-1}^\ell\right]=0$. Also, since we have $|\delta_t^\ell|\leq\sigma$ (see eq. (105)) with $\sigma$ as in eq. (111), then we have $|\zeta_t^\ell|\leq\sigma$ for $t\in\{0,\cdots,T\}$ and $\ell\in\{0,\cdots,L\}$. Hence $\zeta_t^\ell$ is a martingale difference sequence. Then, using the Azuma-Hoeffding inequality, for any $\varepsilon>0$, we get

$$\mathbb{P}\left(\sum_{\ell=1}^L\sum_{t=1}^T\zeta_t^\ell\geq\varepsilon\right)\leq\exp\left(\frac{-\varepsilon^2}{2\sum_{i=1}^{LT}\sigma^2}\right)=\delta.$$

Hence, we get with probability at least $1-\delta$

$$\sum_{\ell=1}^L\sum_{t=1}^T\zeta_t^\ell\leq\sigma\sqrt{2LT\log(1/\delta)}.$$

(129)

Next, using eq. (125) we can write

$$
\begin{aligned}
\sum_{\ell=1}^{L}\sum_{t=0}^{T-1}\Delta_t^\ell(x_t^{*\ell}, s_t^\ell, u_t^{*\ell}) &\leq \sum_{\ell=1}^{L}\sum_{t=0}^{T-1}\chi(\ell)\sqrt{(\phi_t^\ell)^\mathsf{T}Y_t^{*\ell}\left(\Lambda_t^\ell\right)^{-1}(Y_t^{*\ell})^\mathsf{T}\phi_t(s_t^\ell)} \\
&\leq \chi(L)\sum_{\ell=1}^{L}\sum_{t=0}^{T-1}\|\left(\Lambda_t^\ell\right)^{-1/2}(Y_t^{*\ell})^\mathsf{T}\phi_t(s_t^\ell)\| \\
&\overset{(a)}{\leq} \chi(L)\delta_\psi\sum_{\ell=1}^{L}\sum_{t=0}^{T-1}\|\left(\Lambda_t^\ell\right)^{-1/2}\|,
\end{aligned}
$$

where in step (a) we have used eq. (106). Assume the state transition matrix, $\varphi(t, 0)$, in eq. (92) is nonsingular for $t \in \{0, \cdots, T\}$.[3] Then, from Lemma E.2 we have $\mathbb{E}\left[\psi_t\psi_t^\mathsf{T}\right] \succeq \gamma I_{d(n+1)}$ with $\gamma > 0$. Further, from eq. (106), we have $\|\psi_t^\ell\| \leq \delta_\psi$ for $t \in \{0, \cdots, T\}$ and $\ell \in \{1, \cdots, L\}$. Let $\delta \in [0, 1]$ and assume $\ell \geq (8\delta_\psi^2\log(d(n+1)/\delta))/\gamma$. Then, using Lemma E.1 we have with probability at least $1 - \delta$

$$
\lambda_{\min}(\Lambda_t^\ell) \geq \lambda + \frac{(\ell - 1)\gamma}{2}.
$$

Then, we can write

$$
\begin{aligned}
\sum_{\ell=1}^{L}\sum_{t=0}^{T-1}\|(\Lambda_t^\ell)^{-1/2}\| &\leq \sum_{\ell=1}^{L}\sum_{t=0}^{T-1}\frac{1}{\sqrt{\lambda_{\min}(\Lambda_t^\ell)}} \\
&\leq \sum_{\ell=1}^{L}\frac{T}{\sqrt{\lambda + \frac{(\ell-1)\gamma}{2}}} \\
&\leq \frac{T}{\sqrt{\lambda}} + \sum_{\ell=2}^{L}\frac{T}{\sqrt{\lambda + \frac{(\ell-1)\gamma}{2}}} \\
&\leq \frac{T}{\sqrt{\lambda}} + \sum_{\ell=2}^{L}\frac{T\sqrt{2}}{\sqrt{(\ell-1)\gamma}} \\
&\leq \frac{T}{\sqrt{\lambda}} + \frac{2\sqrt{2}T}{\sqrt{\gamma}}\sum_{\ell=2}^{L}(\sqrt{(\ell-1)} - \sqrt{(\ell-2)}) \\
&= \frac{T}{\sqrt{\lambda}} + \frac{2\sqrt{2}T}{\sqrt{\gamma}}\sqrt{(L-1)} \\
&\leq \frac{T}{\sqrt{\lambda}} + \frac{4T\sqrt{L}}{\sqrt{\gamma}}.
\end{aligned}
$$

Then, we have

$$
\sum_{\ell=1}^{L}\sum_{t=0}^{T-1}\Delta_t^\ell(x_t^{*\ell}, s_t^\ell, u_t^{*\ell}) \leq \left(\frac{\delta_\psi T}{\sqrt{\lambda}} + \frac{4\delta_\psi T\sqrt{L}}{\sqrt{\gamma}}\right)\chi(L), \tag{130}
$$

Substituting eq. (129) and eq. (130) in eq. (128), we get

$$
\mathcal{R}(L) \leq \sigma\sqrt{2LT\log(1/\delta)} + \left(\frac{\delta_\psi T}{\sqrt{\lambda}} + \frac{4\delta_\psi T\sqrt{L}}{\sqrt{\gamma}}\right)\chi(L). \tag{131}
$$

the proof follows by substituting $\chi(L)$ defined in eq. (125) in eq. (131), and the probability follows from the union bound.

---

[3]Since the system and the weight matrices are known, this assumption can be satisfied by an appropriate choice of the weight matrices.

## F  Least-squares value iteration for the formulation with process noise in Section 4

In this Appendix, we formulate the regularized least squares regression for the formulation presented in Section 4. Following similar steps as in Appendix B, we define the Bellman target at time $t$ as

$$g_t(x_t, s_t, u_t) = c(x_t, s_t, u_t) + \min_v \widehat{Q}_{t+1}(x_{t+1}, s_{t+1}, v), \tag{132}$$

where $x_{t+1}$ and $s_{t+1}$ denote the states resulting from taking action $u_t$ in states $x_t$ and $s_t$, and $\widehat{Q}_{t+1}(x_{t+1}, s_{t+1}, v)$ is the estimate of the state-action value function at time $t+1$. We re-write eq. (132) as

$$
\begin{aligned}
g_t(x_t, s_t, u_t) =& c(x_t, s_t, u_t) + \widehat{V}^*_{t+1}(x_{t+1}, s_{t+1}) \\
\overset{(b)}{=}& c(x_t, s_t, u_t) + x^{\mathsf{T}}_{t+1} G_{t+1} x_{t+1} + 2\widehat{h}^{\mathsf{T}}_{t+1}(s_{t+1})x_{t+1} + \widehat{q}_{t+1}(s_{t+1}) + \sum_{i=t+2}^{T} \operatorname{tr}[G_i \Sigma_w],
\end{aligned}
\tag{133}
$$

where in step (b), we have used Theorem 4.1, and $\widehat{h}^\ell_{t+1}(\cdot)$ and $\widehat{q}^\ell_{t+1}(\cdot)$ are as in eq. (49) and eq. (50), respectively. The only modification induced by process noise is the appearance of the constant term $\operatorname{tr}(G_{t+1}\Sigma_w)$. The temporal difference (TD) error is written as

$$
\begin{aligned}
\varepsilon_t(x_t, s_t, u_t, x_{t+1}, s_{t+1}) =& g_t(x_t, s_t, u_t, x_{t+1}, s_{t+1}) - \widehat{Q}_t(x_t, s_t, u_t) \\
=& c(x_t, s_t, u_t) + \min_v \widehat{Q}_{t+1}(x_{t+1}, s_{t+1}, v) - \widehat{Q}_t(x_t, s_t, u_t) \\
=& c(x_t, s_t, u_t) + x^{\mathsf{T}}_{t+1} G_{t+1} x_{t+1} + 2\widehat{h}^{\mathsf{T}}_{t+1}(s_{t+1})x_{t+1} + \widehat{q}_{t+1}(s_{t+1}) + \sum_{i=t+2}^{T} \operatorname{tr}[G_i \Sigma_w] \\
& - c(x_t, s_t, u_t) - (Ax_t + Bu_t)^{\mathsf{T}} G_{t+1}(Ax_t + Bu_t) - \phi(s_t)^{\mathsf{T}} Y(x_t, u_t)\widehat{\theta}_{t+1} - \sum_{i=t+1}^{T} \operatorname{tr}[G_i \Sigma_w] \\
=& x^{\mathsf{T}}_{t+1} G_{t+1} x_{t+1} + 2\widehat{h}^{\mathsf{T}}_{t+1}(s_{t+1})x_{t+1} + \widehat{q}_{t+1}(s_{t+1}) - \phi(s_t)^{\mathsf{T}} Y(x_t, u_t)\widehat{\theta}_{t+1} \\
& - (Ax_t + Bu_t)^{\mathsf{T}} G_{t+1}(Ax_t + Bu_t) - \operatorname{tr}(G_{t+1}\Sigma_w),
\end{aligned}
\tag{134}
$$

The TD error $\varepsilon_t(x_t, s_t, u_t, x_{t+1}, s_{t+1})$ in eq. (134) captures the discrepancy between the Bellman target and the current estimate of the $Q$-function. In Least-Squares Value Iteration (LSVI) in Algorithm 1, we minimize the squared TD error over the dataset collected up to episode $\ell - 1$, to obtain an updated estimate of the $Q$-function. Specifically, the parameters $\widehat{\theta}_t$ at episode $\ell$ denoted by $\theta^\ell_t$ is obtained by solving the following regularized least-squares problem

$$
\begin{aligned}
\theta^\ell_{t+1} =& \arg\min_\theta \underbrace{\sum_{j=1}^{\ell-1} \varepsilon_t(x^j_t, s^j_t, u^j_t, x^j_{t+1}, s^j_{t+1})^2 + \lambda\|\theta\|^2_2}_{J} \\
=& \arg\min_\theta \sum_{j=1}^{\ell-1} \Big( (Ax^j_t + Bu^j_t)^{\mathsf{T}} G_{t+1}(Ax^j_t + Bu^j_t) + \phi(s^j_t)^{\mathsf{T}} Y(x^j_t, u^j_t)\theta + \operatorname{tr}(G_{t+1}\Sigma_w) \\
& \qquad - (x^j_{t+1})^{\mathsf{T}} G_{t+1} x^j_{t+1} - 2\Big( h^\ell_{t+1}(s^j_{t+1}) \Big)^{\mathsf{T}} x^j_{t+1} - q^\ell_{t+1}(s^j_{t+1}) \Big)^2 + \lambda\|\theta\|^2_2.
\end{aligned}
\tag{135}
$$

Taking the derivative of eq. (135) with respect to $\theta$, we get

$$
\begin{aligned}
\frac{\partial J}{\partial \theta} =& 2\sum_{j=1}^{\ell-1} Y^{\mathsf{T}}(x^j, u^j)\phi(s^j)\Big( (Ax^j_t + Bu^j_t)^{\mathsf{T}} G_{t+1}(Ax^j_t + Bu^j_t) + \phi(s^j_t)^{\mathsf{T}} Y(x^j_t, u^j_t)\theta + \operatorname{tr}(G_{t+1}\Sigma_w) \\
& \qquad - (x^j_{t+1})^{\mathsf{T}} G_{t+1} x^j_{t+1} - 2\Big( h^\ell_{t+1}(s^j_{t+1}) \Big)^{\mathsf{T}} x^j_{t+1} - q^\ell_{t+1}(s^j_{t+1}) \Big) + 2\lambda\theta.
\end{aligned}
$$

Setting the above derivative to zero and solving for $\theta$, we get

$$\theta_{t+1}^\ell = \Lambda_t^{-1} \sum_{j=1}^{\ell-1} Y^\mathsf{T}(x^j, u^j)\phi(s^j)\epsilon^\ell(x_t^i, x_{t+1}^i, s_{t+1}^i, u_t^i),$$

$$\Lambda_t = \sum_{j=1}^{\ell-1} Y(x^j, u^j)^\mathsf{T}\phi(s^j)\phi(s^j)^\mathsf{T} Y(x^j, u^j) + \lambda I_{d(n+1)}, \tag{136}$$

$$\epsilon^\ell(x_t^i, x_{t+1}^i, s_{t+1}^i, u_t^i) = (x_{t+1}^i)^\mathsf{T} G_{t+1} x_{t+1}^i - 2\big(h_{t+1}^\ell(s_{t+1}^i)\big)^\mathsf{T} x_{t+1}^i + q_{t+1}^\ell(s_{t+1}^i)$$
$$- (Ax_t^i + Bu_t^i)^\mathsf{T} G_{t+1}(Ax_t^i + Bu_t^i) - \mathrm{tr}(G_{t+1}\Sigma_w).$$

## G Least-squares value iteration for the setting with unknown dynamics in Section 5

In this appendix, we derive the least-squares value iteration updates for the unknown-dynamics setting in Section 5 and present Algorithm 2, which extends Algorithm 1 to the case where $A$ and $B$ are unknown. Using the expression of $Q_t$ in eq. (24), we can write

$$Q_t^*(x, s, u) = c(x, s, u) + z^\mathsf{T} P_{t+1} z + 2\sum_{i=1}^d \phi_i(s)z^\mathsf{T}\widetilde{h}_{i,t+1} + \sum_{i=1}^d \phi_i(s)\overline{q}_i(t+1)$$

$$= c(x, s, u) + \big(z^\mathsf{T} \otimes z^\mathsf{T}\big)\mathrm{vec}(P_{t+1}) + \sum_{i=1}^d \phi_i(s)\begin{bmatrix} 2z^\mathsf{T} & 1 \end{bmatrix}\begin{bmatrix} \widetilde{h}_{i,t+1} \\ \overline{q}_{i,t+1} \end{bmatrix}$$

$$= c(x, s, u) + \big(z^\mathsf{T} \otimes z^\mathsf{T}\big)\Gamma\underbrace{\mathrm{vech}(P_{t+1})}_{P_{v,t+1}} + \sum_{i=1}^d \phi_i(s)\begin{bmatrix} 2z^\mathsf{T} & 1 \end{bmatrix}\underbrace{\begin{bmatrix} \widetilde{h}_{i,t+1} \\ \overline{q}_{i,t+1} \end{bmatrix}}_{\overline{\theta}_{i,t+1}}$$

$$= c(x, s, u) + \big(z^\mathsf{T} \otimes z^\mathsf{T}\big)\Gamma P_{v,t+1} + \begin{bmatrix} \phi_1(s) & \cdots & \phi_d(s) \end{bmatrix}\underbrace{\begin{bmatrix} [2z^\mathsf{T}\ 1] & 0 & \cdots & 0 \\ 0 & [2z^\mathsf{T}\ 1] & \cdots & 0 \\ \vdots & \ddots & \ddots & \vdots \\ 0 & 0 & \cdots & [2z^\mathsf{T}\ 1] \end{bmatrix}}_{I_d \otimes [2z^\mathsf{T}\ 1]}\underbrace{\begin{bmatrix} \overline{\theta}_{1,t+1} \\ \vdots \\ \overline{\theta}_{d,t+1} \end{bmatrix}}_{\overline{\theta}_{t+1}}$$

$$= c(x, s, u) + \big(z^\mathsf{T} \otimes z^\mathsf{T}\big)\Gamma P_{v,t+1} + \phi(s)^\mathsf{T}\big(I_d \otimes [2z^\mathsf{T}\ 1]\big)\overline{\theta}_{t+1}$$

$$= c(x, s, u) + \underbrace{\begin{bmatrix} z^\mathsf{T} \otimes z^\mathsf{T} & \phi(s)^\mathsf{T} \end{bmatrix}\begin{bmatrix} \Gamma & 0 \\ 0 & I_d \otimes [2z^\mathsf{T}\ 1] \end{bmatrix}}_{\widetilde{\Psi}(x,s,u)^\mathsf{T}}\underbrace{\begin{bmatrix} P_{v,t+1} \\ \overline{\theta}_{t+1} \end{bmatrix}}_{\theta_{t+1}}$$

$$= c(x, s, u) + \widetilde{\Psi}(x, s, u)^\mathsf{T}\theta_{t+1}. \tag{137}$$

Following similar steps as in Appendix B, we define the Bellman target at time $t$ as

$$g_t = c(x_t, s_t, u_t) + \underbrace{\min_v \widehat{Q}_{t+1}(x_{t+1}, s_{t+1}, v)}_{y_t}, \tag{138}$$

where $x_{t+1}$ and $s_{t+1}$ denote the states resulting from taking action $u_t$ in states $x_{t+1}$ and $s_{t+1}$, and $\widehat{Q}_{t+1}(x_{t+1}, s_{t+1}, v)$ is the estimate of the state-action value function at time $t + 1$. At time $t$, the weight parameter $\theta_{t+1}$ is obtained by solving the regularized least-squares problem

$$\widehat{\theta}_{t+1} = \arg\min_\theta \sum_{i=1}^{\ell-1} \left(\widetilde{\Psi}(x_t^i, s_t^i, u_t^i)^\mathsf{T}\theta - y_t^i\right)^2 + \lambda\|\theta\|_2^2.$$

Following similar steps as in Appendix B, this problem admits the closed-form solution

$$\Lambda_t^\ell = \sum_{i=1}^{\ell-1} \widetilde{\Psi}(x_t^i, s_t^i, u_t^i) \widetilde{\Psi}(x_t^i, s_t^i, u_t^i)^\top + \lambda I_{n_\theta}, \quad \text{with} \quad n_\theta = \frac{(n+m+2d)(n+m+1)}{2}, \tag{139}$$

$$\widehat{\theta}_{t+1}^\ell = (\Lambda_t^\ell)^{-1} \sum_{i=1}^{\ell-1} \widetilde{\Psi}(x_t^i, s_t^i, u_t^i) y_t^i. \tag{140}$$

$$y_t^i = {x_{t+1}^i}^\top G_{t+1}^\ell x_{t+1}^i + 2\big(h_{t+1}^\ell(s_{t+1}^i)\big)^\top x_{t+1}^i + q_{t+1}^\ell(s_{t+1}^i), \tag{141}$$

where

$$G_{t+1}^\ell = P_{11,t+2}^\ell + W - (P_{12,t+2}^\ell + D)(R + P_{22,t+2}^\ell)^{-1}(P_{12,t+2}^\ell + D)^\top, \tag{142}$$

$$h_{t+1}^\ell(s_{t+1}^i) = \begin{bmatrix} I & {K_{x,t+1}^\ell}^\top \end{bmatrix} \big(\phi(s_{t+1}^i)^\top \otimes I_{n+m}\big) \widetilde{h}_{t+2}^\ell + (F + {K_{x,t+1}^\ell}^\top H^\top) s_{t+1}^i, \tag{143}$$

$$\begin{aligned} q_{t+1}^\ell(s_{t+1}^i) = {}& \phi(s_{t+1}^i)^\top \overline{q}_{t+2}^\ell + {s_{t+1}^i}^\top (M + H K_{s,t+1}^\ell) s_{t+1}^i \\ & - 2{s_{t+1}^i}^\top H(R + P_{22,t+2}^\ell)^{-1} \big(\phi(s_{t+1}^i)^\top \otimes \begin{bmatrix} 0_{m\times n} & I_m \end{bmatrix}\big) \widetilde{h}_{t+2}^\ell \\ & - \big(\widetilde{h}_{t+2}^\ell\big)^\top \left(\phi(s_{t+1}^i) \otimes \begin{bmatrix} 0_{n\times m} \\ I_m \end{bmatrix}\right) \big(R + P_{22,t+2}^\ell\big)^{-1} \big(\phi(s_{t+1}^i)^\top \otimes \begin{bmatrix} 0_{m\times n} & I_m \end{bmatrix}\big) \widetilde{h}_{t+2}^\ell, \end{aligned} \tag{144}$$

$$K_{x,t+1}^\ell = -(R + P_{22,t+2}^\ell)^{-1}(P_{12,t+2}^\ell + D^\top), \tag{145}$$

$$K_{s,t+1}^\ell = -(R + P_{22,t+2}^\ell)^{-1} H^\top. \tag{146}$$

Note that the half vectorization ensures that $P_{t+1}$ in step 17 of Algorithm 2 is symmetric. However, to enforce the constraint $P_{t+1} \succeq 0$, we project the estimate onto the positive semidefinite cone. Let $P_{t+1} = U\Sigma U^\top$ be the eigenvalue decomposition of the updated matrix. We then replace $\Sigma$ by $\Sigma_+ = \text{diag}(\max\{\sigma_1, 0\}, \ldots, \max\{\sigma_{n+m}, 0\})$, and set $P_{t+1} = U\Sigma_+ U^\top$, as shown in steps 18-20, where $\{\sigma_1, \cdots, \sigma_{n+m}\}$ are the eigenvalues of $P_{t+1}$. In step 22, we apply the control policy in eq. (25) with an added exploration term. Specifically, the input is chosen as

$$u_t^\ell = \widehat{K}_{x,t} x_t^\ell + \widehat{K}_{s,t} s_t^\ell + \widehat{\widetilde{K}}_{h,t} \big(\phi(s_{t+1}^i)^\top \otimes \begin{bmatrix} 0_{m\times n} & I_m \end{bmatrix}\big) \widehat{\widetilde{h}}_{t+1}^\ell + \eta_t^\ell, \tag{147}$$

where $\eta_t^\ell$ is an exploration signal used to ensure sufficient excitation for learning the unknown system parameters.

## H  Half vectorization

For a matrix $A \in \mathbb{R}^{n\times m}$, let $\text{vec}(A) \in \mathbb{R}^{nm}$ denotes the vectorization of $A$, which is obtained by stacking the columns of $A$ on top of one another. For a symmetric matrix $A \in \mathbb{R}^{n\times n}$,

$$A = \begin{bmatrix} a_{11} & a_{12} & \cdots & a_{1n} \\ a_{12} & a_{22} & \cdots & a_{2n} \\ \vdots & \vdots & \ddots & \vdots \\ a_{1n} & a_{2n} & \cdots & a_{nn} \end{bmatrix},$$

we use $\text{vech}(A) \in \mathbb{R}^{n(n+1)/2}$ to denote the half-vectorization of $A$, which is obtained by vectorizing the lower triangular part of $A$, i.e.,

$$\text{vech}(A) = [a_{11}, a_{12}, \cdots, a_{1n}, a_{22}, \cdots, a_{2n}, \cdots, a_{nn}]^\top.$$

For a symmetric matrix $A \in \mathbb{R}^{n\times n}$, let $P \in \mathbb{R}^{n^2 \times n(n+1)/2}$ be the unique matrix such that $\text{vec}(A) = P\,\text{vech}(A)$, where

$$P = \sum_{i\geq j}^{n} \text{vec}(Q_{ij})\, u_{ij}^\top, \tag{148}$$

where $Q_{ij} \in \mathbb{R}^{n\times n}$ is a matrix with 1 in the $(i,j)$ and $(j,i)$ positions and 0 elsewhere, and $u_{ij} \in \mathbb{R}^{n(n+1)/2}$ is a unit vector with 1 in the position $(j-1)n + i - \frac{1}{2}j(j-1)$ and 0 elsewhere.

---

**Algorithm 2** Least-Squares Value Iteration with Unknown System Dynamics

---

1: **Given:** episodes $L$, horizon $T$, regularizer $\lambda > 0$, projection radius $R_\theta$

2: Initialize $\widehat{\theta}_{t+1}^1 \leftarrow 0$ and $\Lambda_t^1 \leftarrow \lambda I_{n_\theta}$, with $n_\theta = \frac{(n+m+2d)(n+m+1)}{2}$, for all $t = 0, \ldots, T-1$      (for $\ell = 1$)

3: **for** episode $\ell = 1, \ldots, L$ **do**

4:     Sample $x_0^\ell \overset{\text{i.i.d.}}{\sim} \mathcal{N}(0, \Sigma_x)$ and $s_0^\ell \overset{\text{i.i.d.}}{\sim} \mu_0$

5:     **Backward pass: parameter updates using data from episodes** $1, \ldots, \ell-1$

6:     **for** $t = T-1, \ldots, 0$ **do**

7:         Define dataset $\mathcal{D}^{\ell-1} := \{(x_t^i, s_t^i, u_t^i, x_{t+1}^i, s_{t+1}^i) : i < \ell\}$      (cf. eq. (11))

8:         For each sample in $\mathcal{D}^{\ell-1}$, compute the Bellman target

$$y_t^i \leftarrow \min_v \widehat{Q}_{t+1}\left(x_{t+1}^i, s_{t+1}^i, v; \widehat{\theta}_{t+2}^\ell\right)$$

9:         Form features $\widetilde{\Psi}_t^i \leftarrow \widetilde{\Psi}(x_t^i, s_t^i, u_t^i)$ (from eq. (137))

10:         Update Gram matrix and least-squares solution:

$$\Lambda_t^\ell \leftarrow \sum_{i=1}^{\ell-1} \widetilde{\Psi}_t^i(\widetilde{\Psi}_t^i)^\top + \lambda I_{n_\theta}, \quad \text{where} \quad n_\theta = \frac{(n+m+2d)(n+m+1)}{2},$$

$$\widehat{\theta}_{t+1}^\ell \leftarrow (\Lambda_t^\ell)^{-1} \sum_{i=1}^{\ell-1} \widetilde{\Psi}_t^i y_t^i,$$

11:         **if** $\|\widehat{\theta}_{t+1}^\ell\|_2 > R_\theta$ **then**

12:             $\widehat{\theta}_{t+1}^\ell \leftarrow \dfrac{R_\theta}{\|\widehat{\theta}_{t+1}^\ell\|_2} \widehat{\theta}_{t+1}^\ell$

13:         **end if**

14:     **end for**

15:     **Forward pass: greedy roll-out and data collection**

16:     **for** $t = 0, \ldots, T-1$ **do**

17:         Extract $\widehat{P}_{t+1}^\ell$ and $\widehat{\widetilde{h}}_{t+1}^\ell$ from $\widehat{\theta}_{t+1}^\ell$

18:         $P_{t+1} = U\Sigma U^\top$

19:         $\Sigma_+ = \text{diag}(\max\{\sigma_1, 0\}, \ldots, \max\{\sigma_{n+m}, 0\})$

20:         $P_{t+1} \leftarrow U\Sigma_+ U^\top$

21:         Form gains using eq. (26) with blocks $\widehat{P}_{21,t+1}^\ell, \widehat{P}_{22,t+1}^\ell$:

$$\widehat{K}_{x,t} = -(R + \widehat{P}_{22,t+1})^{-1}(D^\top + \widehat{P}_{21,t+1}), \quad \widehat{K}_{s,t} = -(R + \widehat{P}_{22,t+1})^{-1}H^\top, \quad \widehat{\widetilde{K}}_{h,t} = -(R + \widehat{P}_{22,t+1})^{-1}$$

22:         Apply optimal policy with exploration term $\eta_t^\ell$ (cf. eq. (25)):

$$u_t^\ell \leftarrow \widehat{K}_{x,t} x_t^\ell + \widehat{K}_{s,t} s_t^\ell + \widehat{\widetilde{K}}_{h,t}\left(\phi(s_{t+1}^i)^\top \otimes \begin{bmatrix} 0_{m \times n} & I_m \end{bmatrix}\right)\widehat{\widetilde{h}}_{t+1}^\ell + \eta_t^\ell$$

23:         Take action $u_t^\ell$, observe $x_{t+1}^\ell$ and $s_{t+1}^\ell$

24:     **end for**

25: **end for**

---

