# OpenReview forum: "Linear Dynamics meets Linear MDPs: Closed-Form Optimal Policies via Reinforcement Learning"
_TMLR — Under review for TMLR_

### Review · Reviewer_TWTf · 2026-05-08

**Summary Of Contributions:**

This paper proposes a reinforcement learning framework that couples a deterministic linear time-invariant (LTI) dynamical system with a feature-based linear Markov process modeling an exogenous stochastic environment. The assumption is that the environment evolves independently of the agent's control actions. By exploiting this decoupling and the quadratic cost structure, the authors derive a parametric closed-form expression for the optimal Q-function and policy, then propose a least-squares value iteration (LSVI) algorithm that learns the value function parameters without requiring exploration or explicit transition estimation. Extensions to Gaussian process noise and unknown system dynamics are presented numerically.

Key strengths:
- Clean derivation of a closed-form policy that avoids inner optimization at each step, a genuine simplification over standard LSVI approaches.
- ISS stability guarantees under the learned policy are a meaningful contribution that bridges RL regret analysis with control-theoretic safety requirements.

Key weaknesses:
- The exogenous environment assumption is highly restrictive and limits the practical relevance of the framework to a narrow class of problems.
- Only toy 2-dimensional numerical examples are provided, despite motivating the work with power systems, robotics, and economics.
- The extensions to process noise (Section 4) and unknown dynamics (Section 5) lack theoretical guarantees, and the unknown dynamics setting is restricted to stable systems.

**Audience:**

Yes

**Audience Explanation:**

The paper addresses a legitimate gap at the intersection of control theory and RL. Researchers working on structured RL problems would find the closed-form policy structure and the ISS stability analysis of interest. The observation that exogenous environment structure eliminates the need for exploration and improves regret scaling is a useful insight.

The presentation is generally clear and well-organized, with the main results accessible to readers from both control and RL backgrounds.

**Broader Impact Concerns:**

None. The paper includes a substantive Broader Impact statement (page 16) that appropriately identifies both positive potential (adaptive control under uncertainty) and risks (deployment without sufficient validation in safety-critical systems). The theoretical nature of the work does not raise immediate ethical concerns.

**Claims And Evidence:**

Yes

**Claims Explanation:**

Partially.

The theoretical results for the nominal setting (Theorems 3.1, 3.3, 3.4) are the strongest part of the paper. The Q-function decomposition in Theorem 3.1 is derived correctly via a clean induction argument (Appendix A), and the decoupling of the LTI state from the environment state through the feature map is well-established. The closed-form policy in Corollary 3.2 follows naturally. The regret bound (Theorem 3.4, Appendix E) follows standard techniques from the linear bandit/MDP literature, adapted to the specific structure of this problem. The improvement to linear dependence on T and on d stems from the exogenous nature of the environment, which removes the need for exploration bonuses and simplifies the error propagation.

However, several claims are not adequately supported:

The paper's framing as "the first to integrate LQR and linear MDPs" (Section 1.2, first bullet) requires qualification. Recent work on linear Bellman completeness (e.g., "Computationally Efficient RL under Linear Bellman Completeness for Deterministic Dynamics
", Wu et. al. 2025) explicitly establishes that the linear Bellman complete setting unifies linear MDPs and LQR under a common framework. While the present paper's specific formulation with an exogenous environment is novel, the broader claim of first integration is overstated, and this line of work is not cited.

The practical motivation is not matched by the experiments. All three numerical experiments use small 2D systems with a single scalar control input and a univariate Markov environment with d=2 features. The gap between the motivating applications and the illustrative examples is substantial. There is no demonstration on a system of even moderate dimension, no comparison with any baseline method, and no computational timing results.

The extensions in Sections 4 and 5 are incomplete. The process noise extension (Section 4) shows that the Q-function structure is preserved up to an additive constant, which is a useful observation, but the stability and regret analyses are not extended. The unknown dynamics extension (Section 5) introduces Algorithm 2 but, as the authors acknowledge in Remark 5, neither stability nor convergence is guaranteed. The numerical results for the unknown dynamics setting use a stable open-loop system specifically because the method fails for unstable systems (noted at the end of Section 6.3). This is a strong limitation for a control framework, since many systems of practical interest are open-loop unstable.

**Requested Changes:**

1. Cite and discuss the linear Bellman completeness literature, which establishes that linear Bellman completeness unifies linear MDPs and LQR. Revise the "first to integrate" claims accordingly. The paper's contribution is the specific exogenous-environment formulation and the resulting algorithmic/theoretical benefits, not the general idea of combining LQR and linear MDPs.

2. Provide at least one experiment of meaningful scale (e.g., a system with dimension n >= 10, multiple control inputs, and a higher-dimensional environment state). The current 2D examples do not demonstrate that the approach scales or that it offers practical benefits over simpler methods (e.g., standard LQR ignoring the environment structure, or certainty-equivalent control with estimated environment parameters).

3. Include baseline comparisons in the experiments. At minimum, compare against: (a) a standard LQR controller that ignores the environment, (b) the LSVI-UCB algorithm applied to the full state space. Without these, it is impossible to assess the practical value of the proposed framework.

4. Discuss the restrictiveness of the exogenous environment assumption more thoroughly. Provide concrete examples of systems that do and do not satisfy this assumption among the motivating applications.

5. For the process noise extension (Section 4), extend the stability analysis or at least provide a formal statement of what additional conditions would be needed for ISS to hold. The current treatment stops at showing the Q-function structure is preserved.

6. Clarify the relationship between the projection radius R_theta and practical algorithm performance. Remark 3 states it should be "large enough" but the bound depends on unknown quantities (the true parameters). Discuss how a practitioner would choose this in the absence of oracle knowledge.

---

### Review · Reviewer_qKwr · 2026-05-11

**Summary Of Contributions:**

The paper studies the policy optimization problem in a linear dynamical system under the influence of an exogenous stochastic environment whose transition kernel is unknown but admits a linear feature representation. The authors first consider the setting where the dynamical system evolves in a deterministic fashion with known dynamics, and designs a least-squares value iteration algorithm that enjoys a sublinear regret bound. The authors then discuss and make the generalization to problems with known dynamics and Gaussian noise, and those with unknown dynamics. Small scale numerical simulations are carried out, which verifies the algorithm performance and certain theoretical results.

**Audience:**

Yes

**Audience Explanation:**

The paper studies a problem at the intersection of linear control and policy optimization in linear Markov (decision) processes. There are large research communities for both topics. However, I am not quite sure whether the particular problem formulation introduced in this paper as a hybrid of the two worlds will be of interest to the communities, as it is unclear what concrete applications can be modeled by this framework.

**Broader Impact Concerns:**

No concerns

**Claims And Evidence:**

Yes

**Claims Explanation:**

The paper presents mathematical derivations and supporting empirical evidence in an overall clear and sound manner. The assumptions required for the analysis in the nominal setting are clearly stated.

**Requested Changes:**

1) The paper is technically sound and delivers a suite of solid results on the proposed problem formulation. However, my main concern, as stated above, is whether the problem formulation meaningfully models real-world problems. I strongly believe that the paper needs to discuss motivating application more clearly. What problems are naturally formulated with the state being a mixture of a continuous component evolving according to a linear dynamical system and a discrete one evolving according to a Markov chain? I do not see what problems satisfy the structure in practice.

2) Is $u^\star_t$ in Corollary 3.2 defined?

3) Regarding the paragraph after Theorem 3.4 -- It is unclear from the discussion whether the better dependency of the regret on $T$ and $d$, compared to Jin et al., is due to the simplified problem setting or improved analytical techniques. Is there a technical innovation that the authors come up with here?

4) The authors discussed the challenge of analyzing the algorithm regret in systems with unknown dynamics and the need of new tools beyond the ones used in this paper. What about the case of known systems with process noise? Is there a technical challenge that prevents the authors from deriving a regret guarantee?

---

### Review · Reviewer_AeAF · 2026-05-17

**Summary Of Contributions:**

This paper introduces a new framework of modeling agents with LQR and environments with linear MDP. The problem is formulated to minimize a cumulative quadratic cost where the policy must account for both the LQR state and the linear MDP state, under the assumption that the exogenous environmental state is unaffected by the agent's policy. While the agent’s internal state information is known, the transition probability in the linear MDP is unknown. The authors propose finding the optimal policy by approximating the Q-function via least-squares value iteration. The paper provides an Input-to-State Stability (ISS) result and a regret bound over $L$ episodes. The authors also provide extensions for noisy LQR and unknown LQR dynamics, accompanied by empirical simulations.

**Strength**
1. The problem is well motivated in the sense that common frameworks do not distinguish between the dynamics of the agent and the external environments.
2. It is both clear and conceptually interesting to see how this new combined problem can be solved by combining classical tools from LQR with linear MDP frameworks.

**Weakness**
1. While the assumption that the environment states are unaffected by the actions simplifies the analysis, it is a bit restrictive. In real-world setting, it is also natural to assume that the agent’s actions alter the underlying state of environment.
2. The presentation of the experimental results is highly redundant. The textual descriptions in Sections 6.1, 6.2, and 6.3 use nearly identical boilerplate phrasing to describe the graphs. Rather than repeating that the algorithm converges in each setting, the authors should use this space to discuss the differences between the settings.
3. At the end of Section 6.3, the authors state, “We have also observed that the regret may not converge if the open loop system is unstable." However, this empirical observation is completely unsupported by any figure, table, or supplementary data in the manuscript.
4. In Remark 4, the authors introduce an excitation term ($\eta_t$) to learn the unknown dynamics. This is a conventional way to learn the parameters in control, but they didn't provide theoretical guarantees for their new framework. Moreover, a discussion about degradation from injecting random excitation should be included.

**Audience:**

Yes

**Audience Explanation:**

Even though LQR and linear MDPs are well-studied problems, modeling agents and environments as distinct coupled systems is an interesting approach that will appeal to researchers at the intersection of control theory and reinforcement learning.

**Claims And Evidence:**

No

**Claims Explanation:**

1. Theorem 3.4 assumes a uniform upper bound $\delta_\psi$ for $||Y(x_t^\ell, u_t^\ell)^\top \phi(s_t^\ell)||$. However, based on the ISS bound, the bound of $||x_t^\ell||$ explicitly depends on $||x_0^\ell||$, which is drawn from a Gaussian distribution. Because Gaussian random variables have infinite support, a strict uniform bound is mathematically unfeasible. The authors must treat this using a high-probability bound.
2. Lemma E.1 applies Tropp’s (2012) Matrix Chernoff bound, which strictly requires the matrices {$z_iz_i^\top$} to be statistically independent. When applied in the regret bound, the vectors $z_i$ are replaced with $\psi_t^i$. However, since $\psi_t^i$ consists of $s_t$ and $x_{t+1}$, the vectors $\psi_t^i$ are not sequentially independent. Therefore, a standard Chernoff bound cannot be applied.
3. In the proof of Theorem 3.4, when they apply Self-Normalized Bound for Vector-Valued Martingales (Theorem 1 by Abbasi-Yadkori et al., 2011), the definition of the filtration $\mathcal{F}_t^{\ell-1}$ is totally missing. This is an important condition to apply the theorem, so this should be dealt carefully.
4. The experimental setting violates the theoretical assumptions. The feature function $\phi$ and the measure $\mu_t$ defined in Eq. (28) do not satisfy the strict norm bounds required by Assumption 2.1. The authors do not acknowledge or justify this relaxation anywhere in the text.

**Requested Changes:**

1. Address the sequential dependence in Lemma E.1. A proper martingale-based matrix concentration inequality must be used in place of the standard Matrix Chernoff bound.
2. Define a filtration and justify the application self-normalized bound exactly.
3. Revise the derivation of $\delta_\psi$ in Theorem 3.4 to properly reflect a high-probability bound over the Gaussian initial states, rather than treating it as a uniform constant.
4. Explicitly clarify and justify the mismatch between the theoretical bounds in Assumption 2.1 and the empirical values used in Eq. (28).
5. Condense the repetitive text in Sections 6.1, 6.2, and 6.3. Expand the discussion to highlight how the extensions (noise, unknown dynamics) uniquely challenge the algorithm compared to the noiseless/known dynamics setting.
6. Provide a supplementary figure or appendix data to substantiate the claim in Section 6.3 regarding non-convergence under open-loop instability, or revise the text to frame this purely as a theoretical limitation.
7. In Remark 4, provide proper discussions and citations for the excitation term.
8. In Section 3.3, explicitly state how the constant $\rho$ is chosen. The $\rho$ comes from the spectral radius of the closed-loop system, but it is not described in the section.

---

### Review · Reviewer_B5ih · 2026-05-18

**Summary Of Contributions:**

The paper studies a control problem where a known deterministic linear system is coupled with an markovian process with transition kernel being linear with known features. The objective is a finite-horizon quadratic-cost control problem over the system state, environment state, and action. The main contribution of this paper is a closed-form parametric representation of the optimal Q-function and greedy policy, followed by an LSVI algorithm that estimates the unknown value-function parameters without explicitly estimating the transition kernel. The paper proves an ISS-style stability bound and a high-probability regret bound for the nominal known-dynamics setting and provides an extension to the unknown system dynamics setting.

The paper has an interesting modeling direction. Combining LQR structure with feature-based linear Markov models is a natural and useful way to separate known physical dynamics from unknown exogenous stochastic dynamics. The Riccati recursion handles the quadratic part in the controllable state, while the unknown Markov process enters through feature-linear terms.

The theoretical results in nominal settings are interesting. The paper gives an ISS-type bound for the learned policy and a regret bound, which brings new useful insights for this structure.

The paper compares mainly to linear MDP work such as Jin et al., but the comparison is not precise. The model here is substantially easier in some respects because the unknown transition is exogenous and action-independent, while the physical dynamics and quadratic cost are known in the nominal setting. Therefore, the claimed improvement in horizon and feature dimension over standard linear MDP bounds is not a fair comparison.

The action-independent Markov process is reasonable for some exogenous signals, such as weather or demand, but it rules out many robotics/autonomous-system settings where the environment state responds to the agent’s actions. The paper’s claim that no exploration is needed depends critically on this assumption. This, in my opinion, extremely limit the application of this framework.

The controllability assumption is also strong, which assumes an input sequence can drive the system from any initial state to any final state in finite time. This is a strong assumption for real systems, especially because the paper uses unconstrained inputs.

Theorem 3.4 assumes a uniform bound, but the papers samples $x_0$ from a gaussian, which has unbounded support. Can the authors explain more clearly on the setup or am I missing something?

The Gaussian process-noise extension is mathematically straightforward, but this is more like an observation than a substantial new contribution, and the paper provides no stability or regret analysis for this case. The unknown-dynamics extension is even less satisfactory. The paper explicitly states that Algorithm 2 has no closed-loop stability or convergence guarantee in general, especially for unstable systems, and later reports that regret may not converge when the open-loop system is unstable.

The experiments are all based on low-dimensional synthetic tracking examples with essentially the same exogenous process, lacking baselines, ablations over feature dimensions or horizon, etc.

**Audience:**

Yes

**Audience Explanation:**

Integrating control results back into RL is interesting.

**Broader Impact Concerns:**

There's no ethical concern that I am aware of.

**Claims And Evidence:**

Yes

**Claims Explanation:**

Partially yes.  As mentioned above, the assumption on the gaussian distribution is lacking rigor.

**Requested Changes:**

Can the authors provide more comprehensive literature review over the related work? For convergence guarantee, I'd like to see comparison against LQR and linear MDP results, assumptions, key algorithm design choices. There should also be discussion on extension of LQR in the noisy setting, or unknown dynamics setting.

---

### Review · Reviewer_2Qb7 · 2026-06-19

**Summary Of Contributions:**

This paper introduces an RL framework for systems with known physical dynamics, but unknown exogenous environments. The latter are modeled using Linear MDPs. The authors propose an episodic algorithm using value iteration to estimate the environment dynamics-dependent parameters. Key theoretical contributions include guaranteeing input to state stability and a sublinear regret bound.

**Additional Comments:**

#### Section 2
- How easy is it generalize the assumption that environment does not get affected by control actions?
#### Generic
- The paper will benefit from another round of careful proofreading. I have pointed out some potential typos above.
- In some references, the publication venue is absent, e.g., Celi et al. 2022.

**Audience:**

Yes

**Audience Explanation:**

The paper bridges control theory (with perfectly known system dynamics) and RL (with stochastic environments). This will be of interest to both communities.

**Claims And Evidence:**

Yes

**Claims Explanation:**

This is primarily a theoretical paper. So, the theoretical results stated in the paper are supported by rigorous proofs. I verified the proofs in Appendix A, B, and D, and gave comments on minor issues below.

**Requested Changes:**

I sincerely apologize to the authors for the delay in submitting my review due to several personal reasons. I liked this paper, even though I don't exactly work in the field.

#### Section 3
- Theorem 3.1
    - In (38), it should be $P_{T-1}$ instead of $P_t$, right? Similarly in (41), it should be $P_k$.
    - Check the typo in $h$ term in (42)
    - Also, $s'$ and $s_{t+1}$ are used at different places in Theorem A.1 and the proof. Can you make it consistent?
    - In the final step of the proof, we see $\bar q_{i,t+1}, \bar h_{i,t+1}$ defined as "expected values." However, $\mu$ is a signed measure in Assumption 2.1. How are these quantities "expectation" then?
- In (9), isn't $q$ a scalar?
- Section 3.2 and Appendix B
    - In (49)-(50), on the right-hand side of equations, the $h,q$ terms should be written with hats: $\hat h_{t+2}$ and $\hat q_{t+2}$, right?
    - This might be a very basic question, but in LSVI, why don't we sum over $t$, and only over the episodes $\ell$?
    - Isn't the ridge-regression penalty sufficient to ensure that $\theta$ stays bounded? Why do we need the additional projection? I suppose there should be a correspondence between $R_\theta, \lambda$ which eliminates the need for projection.
- Section 3.3
    - Please mention explicitly the norm used for $K_{s,t}, K_{h,t}$?
    - Again, maybe a basic question, but the bound in theorem 3.3 is independent of the no. of past episodes $\ell$ and $t$. I understand the independent in $t$ - we want stability at all $t$. But, should we not get some improvement with increasing $\ell$? Having seen multiple episodes, should I not get better stability?
- Section 3.4
    - What is $\beta$?
    - $\delta$ should be in $(0,1/3)$, not closed interval.
    - In the regret bound, why does $\sqrt{n}$ appear? Given that we have perfect knowledge of the control dynamics.
    - Also, can you explain the regret bound in terms of what the expected regret is, and the blow-up due to concentration?

#### Section 6
- Fig. 1 - What is the additional information in 1(b) compared to 1(a). Doesn't sublinear regret also imply that avg regret will converge?
- Fig. 2(d) - i don't see the trajectories tracking the mean. What am I missing?